# Nanoplastic concentrations across the North Atlantic

Sophie ten Hietbrink[1,5,6], Dušan Materić[1,2,6 ✉], Rupert Holzinger[1], Sjoerd Groeskamp[3] & Helge Niemann[3,4]

Plastic pollution of the marine realm is widespread, with most scientific attention given to macroplastics and microplastics[1,2]. By contrast, ocean nanoplastics (<1 μm) remain largely unquantified, leaving gaps in our understanding of the mass budget of this plastic size class[3–5]. Here we measure nanoplastic concentrations on an ocean-basin scale along a transect crossing the North Atlantic from the subtropical gyre to the northern European shelf. We find approximately 1.5–32.0 mg m$^{-3}$ of polyethylene terephthalate (PET), polystyrene (PS) and polyvinyl chloride (PVC) nanoplastics throughout the entire water column. On average, we observe a 1.4-fold higher concentration of nanoplastics in the mixed layer when compared with intermediate water depth, with highest mixed-layer nanoplastic concentrations near the European continent. Nanoplastic concentrations at intermediate water depth are 1.8-fold higher in the subtropical gyre compared with the open North Atlantic outside the gyre. The lowest nanoplastic concentrations, with about 5.5 mg m$^{-3}$ on average and predominantly composed of PET, are present in bottom waters. For the mixed layer of the temperate to subtropical North Atlantic, we estimate that the mass of nanoplastic may amount to 27 million tonnes (Mt). This is in the same range or exceeding previous budget estimates of macroplastics/microplastics for the entire Atlantic[6,7] or the global ocean[1,8]. Our findings suggest that nanoplastics comprise the dominant fraction of marine plastic pollution.

Concerns about plastic in the environment had already been raised in the 1960s (ref. 9). By now, it has become one of the largest contemporary environmental hazards[10], with plastic accumulating in every known natural habitat[11–14]. A substantial fraction of the global annual plastic production ends up in the ocean[15], for example, through riverine transport[16,17], atmospheric deposition[18] and direct coastal or ship-based littering[19]. The further fate of plastic debris in the ocean depends on several factors, including the density of the plastic items and their transport at the ocean surface[3]. Accumulation hotspots of floating plastics include bays and convergence zones, such as the subtropical ocean gyres[1,8], and a considerable fraction of marine plastic litter is redeposited along shorelines[1,19,20]. Plastic may also degrade: wave action exerts shear stress, solar ultraviolet radiation induces photooxidation and microbes can further weaken the structural integrity of the polymer so that macroplastic items (size: >5 mm) fragment into microplastics (size: 1 μm to 5 mm) and nanoplastics (size: <1 μm)[3,21–23]. In particular, photodegradation has been discussed as a key process in the breakdown of floating plastic litter at the sea surface it probably provides a constant source of nanoplastic particles to the ocean[3,23,24], with potentially negative effects on marine life[10,25,26]. In contrast to macroplastics and microplastics, the dispersion of nanoplastics is not governed by buoyancy properties. With decreasing particle size, dispersion is more dominantly controlled by the collision of nanoplastics with water molecules and Brownian motion[27].

Polythene (PE), PS, PVC and PET particles are indeed found as nanoplastics in the ocean[4,5,28], but the distribution and concentrations of nanoplastics, both geographically and over depth, are virtually unknown. This knowledge gap exists because it is challenging to sample and analyse nanoplastics at environmentally relevant concentrations[29,30]. Hence, nanoplastics are not included in any ocean plastic budget estimates[1,6,8]. This hinders our comprehensive understanding of the potential environmental impact and health hazards of marine plastic pollution. A skewed ocean plastic size distribution towards smaller particle diameters[31,32], however, suggests that nanoplastics could be a globally important contaminant[6].

During a research cruise with RV Pelagia in 2020, we sampled the water column from the sea surface to the bottom across the North Atlantic Ocean from the subtropical gyre to the northern European shelf (Fig. 1) and measured nanoplastics with thermal-desorption proton-transfer-reaction mass spectrometry (TD-PTR-MS). This method allows identification of the polymer backbone as well as quantification of nanoplastic particles in seawater using fingerprinting algorithms[4,33].

[1]Institute for Marine and Atmospheric Research Utrecht (IMAU), Utrecht University, Utrecht, The Netherlands. [2]Department of Environmental Analytical Chemistry (EAC), Helmholtz Centre for Environmental Research – UFZ, Leipzig, Germany. [3]NIOZ Royal Netherlands Institute for Sea Research, 't Horntje (Texel), The Netherlands. [4]Department of Earth Sciences, Utrecht University, Utrecht, The Netherlands. [5]Present address: Department of Geological Sciences, Stockholm University, Stockholm, Sweden. [6]These authors contributed equally: Sophie ten Hietbrink, Dušan Materić. ✉e-mail: dusan.materic@ufz.de

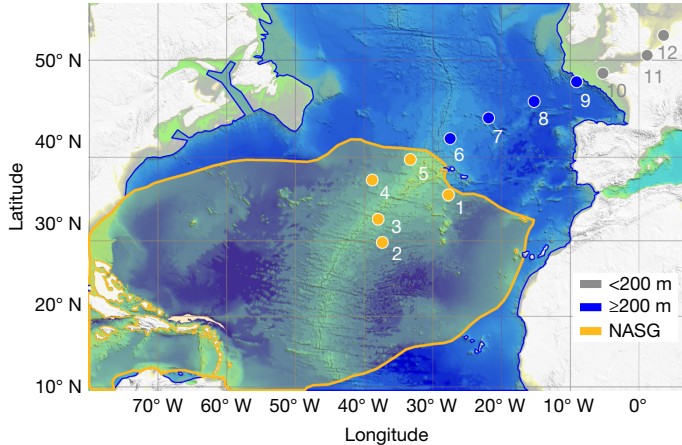

**Fig. 1 | Map of the 12 hydrocast stations along a transect crossing the North Atlantic from the subtropical gyre to the northern European shelf.** Stations 1–5 are located in the NASG ('gyre'), stations 6–9 are in the open ocean (that is, water depth ≥ 200 m; 'outside gyre') between the shelf and the NASG and stations 10–12 are on the European shelf (water depth below 200 m; 'coastal'). The extent of the NASG (Extended Data Figs. 5 and 6) is highlighted in orange and the remaining part of the open subtropical to temperate North Atlantic (8° N to 55° N) is highlighted in blue. Bathymetry data were compiled from the freely available databases of GEBCO (https://www.gebco.net/) and EMODnet (https://emodnet.ec.europa.eu/en) and the map was created with the Global Mapper software package.

## Ubiquitous presence of nanoplastics

Samples for nanoplastic analysis were recovered from 12 hydrocast stations, of which stations 1–5 were located in the North Atlantic subtropical gyre (NASG; 'gyre'), stations 6–9 were in the open ocean but outside the gyre ('outside gyre') and stations 10–12 were on the European shelf ('coastal') (Fig. 1).

The mixed layer of the ocean was sampled at 10 m water depth (see Extended Data Fig. 5c for mixed-layer depth ranges of the stations). Nanoplastics in this layer comprise PVC, PET and PS in the mg m$^{-3}$ range at all 12 hydrocast stations (Fig. 2a), amounting to a total nanoplastic concentration (PVC + PET + PS) of about 18.1 ± 2.1 mg m$^{-3}$ (average ± standard error). In one sample (station 8; mixed layer), polypropylene (PP) and polypropylene carbonate (PPC) were also detected (24.27 and 21.25 mg m$^{-3}$, respectively; data not shown). Because this sample was anomalous compared with all of the other results, we cannot rule out the possibility that the PP and PPC are a result of contamination and, hence, we excluded these results from successive analyses. We found that total nanoplastic concentrations were ≥1.5-fold higher at the 'coastal' stations (25.0 ± 4.2 mg m$^{-3}$) when compared with the open-ocean regions (Fig. 2d). Differences in nanoplastic concentrations were mainly caused by ≥1.7-fold higher PS and ≥1.7-fold higher PET concentrations when comparing the 'coastal' with the open-ocean stations (Extended Data Fig. 1). PVC concentrations were, on the other hand, only slightly higher (≲1.3-fold). The 'gyre' stations showed a lower average concentration of total nanoplastics (15.1 ± 3.3 mg m$^{-3}$) when compared with the 'outside gyre' stations (16.7 ± 3.5 mg m$^{-3}$), but this was not significant (Fig. 2d). No notable differences were found for single polymers when comparing 'gyre' and 'outside gyre' stations.

Similar to the mixed layer, we found PVC, PET and PS nanoplastics in the intermediate layer at 1,000 m water depth (stations 1–9; Fig. 2b) amounting to an average nanoplastic concentration of 10.9 ± 1.6 mg m$^{-3}$. The water depth at all 'coastal' stations was <1,000 m, restricting comparison of the intermediate water layer to the 'gyre' and 'outside gyre' stations. The intermediate depth at the 'gyre' stations showed a 1.8-fold higher average concentration of total nanoplastics (13.5 ± 2.0 mg m$^{-3}$) compared with the 'outside gyre' stations (7.5 ± 2.2 mg m$^{-3}$; Fig. 2e).

Unlike the ubiquitous presence of all polymer types in the mixed layer, we could not observe PS, PVC and PET across stations consistently. PET nanoplastic concentrations were 2.5-fold higher in the 'gyre' compared with the 'outside gyre' stations. PVC and PS concentrations in the 'gyre' and 'outside gyre' stations were similar (Extended Data Fig. 1).

Ocean-bottom waters (sampled 30 m above the seafloor) contained considerable amounts of PET, whereas PVC and PS were, with the exception of one station, below detection limit (Fig. 2c). The average total nanoplastic bottom-water concentration was 5.5 ± 0.6 mg m$^{-3}$ along the transect from stations 1 to 9. Because of the shallow water depth at stations 10–12, bottom waters at these stations were sampled at approximately 5–10 m above the seafloor (and not 30 m above seafloor) and thus excluded from statistical comparison. The highest total nanoplastic concentration was observed at station 8, exclusively consisting of PET (Fig. 2c). No significant differences in total nanoplastic concentrations were found when comparing bottom waters from the 'gyre' and 'outside gyre' stations (Fig. 2f).

We assessed the vertical distribution of nanoplastics in the North Atlantic water column by averaging total nanoplastic concentrations along the open-ocean section of the transect (stations 1–9) for every depth interval (Fig. 2g). Average total nanoplastic concentrations decreased 1.4-fold, from mixed-layer to intermediate waters, and foremost by 2.0-fold from intermediate to bottom waters (Fig. 2h). The decrease in PVC and PS, 2.6-fold and 2.0-fold, respectively, from mixed-layer to intermediate waters and 12.1-fold and 13.3-fold from intermediate to bottom waters seemed comparably steady (Extended Data Fig. 1). PET concentrations, on the other hand, remained relatively high throughout the water column.

## Controls on nanoplastic distribution

The hotspot concentrations in the mixed layer close to the European continent (Fig. 2d) and, to a lesser extent, in intermediate waters in the NASG (Fig. 2e) indicate two sources of nanoplastics. At the shelf, nanoplastics may enter the ocean through the same routes as macroplastics and microplastics, that is, by means of rivers and surface water runoff[4,16,17,34] (Fig. 2i). Also, nanoplastic from land can become airborne and transported as nanoplastic aerosols, eventually entering the ocean through wet and dry deposition[35,36]. Shelf mixed-layer waters with comparably high nanoplastic concentrations[4] are then entrained with less polluted offshore waters (Fig. 2d), which explains our finding of decreased nanoplastic concentrations further away from the coast. Although atmospheric deposition of microplastics and nanoplastics to the surface ocean is not constrained in our study, it seems likely that this decreases offshore just as for other land-based aerosol sources[37]. However, floating macroplastics and microplastics generally accumulate in the subtropical gyres[1,7,8,38] and probably release secondary nanoplastics, originating from continuing fragmentation of the floating plastic through shear stress (waves) and photodegradation (solar ultraviolet light)[23,24,39,40]. The moderate difference in nanoplastic concentrations between 'gyre' and 'outside gyre' stations (Fig. 2d) thus indicates that nanoplastic concentrations in the mixed layer might be horizontally homogenized as a result of shear dispersion and wind-induced turbulent mixing[41,42]. Also, nanoplastics might be redistributed through air–sea interactions. Particles <1 μm can be released to the atmosphere by means of bubble burst ejection and aerolization of spray[36,43], after which they can be transported over long distances of hundreds of kilometres in the atmosphere before being redeposited into the ocean[44].

## Vertical distribution of nanoplastics

Compared with the mixed layer, a different nanoplastic distribution pattern emerges at 1,000 m water depth, with a more distinct maximum in nanoplastic concentrations at 'gyre' stations (Fig. 2d,e). Here, differences in nanoplastic concentrations reflect relative differences

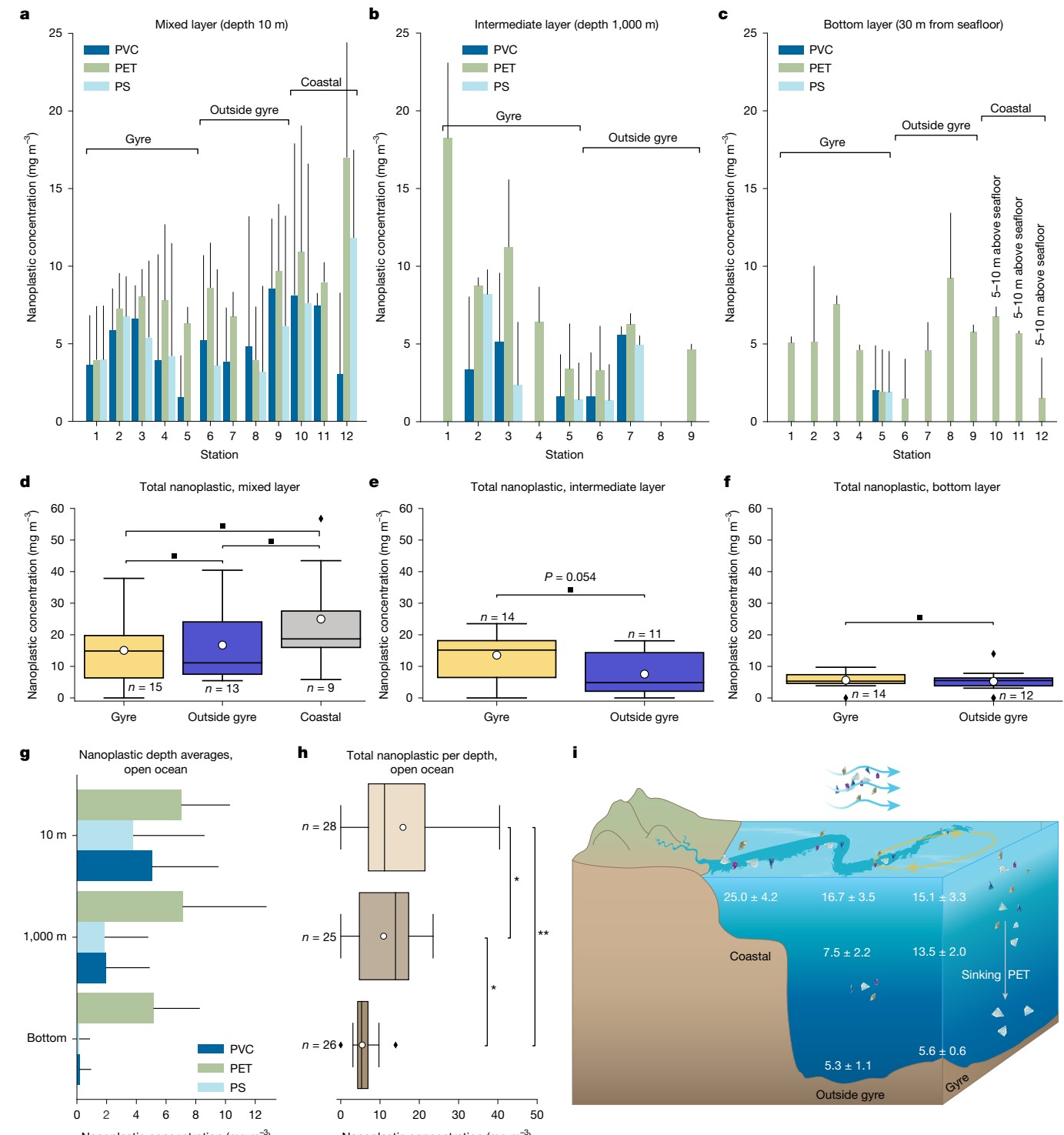

**Fig. 2 | Horizontal and vertical distribution of nanoplastics in the water column of the North Atlantic. a–c**, Average nanoplastic concentrations of PVC, PET and PS at 12 stations along a transect from the NASG ('Gyre'; stations 1–5), the open ocean between the shelf and the gyre ('Outside gyre'; stations 6–9) and at the shelf break or on the European shelf ('Coastal'; stations 10–12). Nanoplastic concentrations were measured at three water depths from the mixed layer (10 m below sea level, mbsl) (**a**), intermediate layer (1,000 mbsl; only offshore stations 1–9) (**b**) and bottom layer (30 m above the seafloor at the offshore stations 1–9 and 5–10 m above the seafloor at coastal stations 10–12) (**c**). The error bars represent the s.d. of the measurements taken at each station. **d–f**, Total (PS + PET + PVC) nanoplastic concentrations for the three groups 'Gyre', 'Outside gyre' and 'Coastal' in the mixed layer (**d**), intermediate layer (**e**) and bottom layer (**f**) shown as box plots. **g,h**, For the open ocean (stations 1–9),

average concentrations over depth are shown for individual (**g**) or total (**h**) nanoplastic concentrations. In **g**, the error bars represent the s.d. of the nanoplastic concentrations in each depth category. All box plots indicate the ±25 percentiles of the median, with the whiskers extending to the data points that fall within the 1.5 interquartiles. Data points that fall outside this range are indicated by a diamond. The mean value is indicated with the white dot. Differences between groups were analysed using a one-way ANOVA test and a *t*-test for means comparison. Significance levels with $P < 0.01$ (**), $0.01 < P < 0.05$ (*) and $P > 0.05$ (■) are indicated. **i**, Overview of the average nanoplastic concentrations and standard error (in mg m$^{-3}$) in the 'Gyre', 'Outside gyre' and 'Coastal' regions. Putative origins of nanoplastics and transport processes are highlighted.

in floating and submerged macroplastic and microplastic concentrations[1,6–8]. This suggests a decoupling of processes determining the horizontal distribution of nanoplastics in the mixed layer versus deeper-water layers. Indeed, stratification separates these water layers (Extended Data Fig. 2) and thus strongly reduces solute exchange between the two water masses. However, sinking particles and aggregates (for example, marine snow) can cross the pycnocline[45]. Hence, as well as varying circulation patterns and stratification, differences in productivity across ocean provinces may also influence the distribution of nanoplastics. However, the 1-μm filtration threshold excludes marine snow, preventing us from accounting for most aggregated nanoplastics. PVC, PS and, most importantly, PET were found to largely contribute to the submerged macroplastics and microplastics pool just below the mixed layer (approximately 100–300 m water depth) at both 'gyre' and 'outside gyre' regions of the North Atlantic[7]. Moreover, the presence of PET nanoplastic at water depths of >300 m was recently demonstrated[5]. Sinking of macroplastics and microplastics and continuing fragmentation of the submerged and sinking particles are hence a seemingly important factor in determining nanoplastic concentration and distribution in the intermediate water layer. An abundance of plastic particles, more dominantly composed of polyesters, was found on and in deep-sea sediments[46,47]. Nanoplastic production from sinking microparticles and macroparticles is hence the least parsimonious explanation for the presence of nanoplastics in bottom waters, as well as sinking of nanoplastic aggregates. At these depths below the epipelagic zone, continuing photooxidation will have diminished, although continuing fragmentation can be a result of antecedent photodegradation[48]. Other possible mechanisms contributing to nanoplastic production could be mechanical stress[49], although to a lesser extent than for the mixed layer, and biodegradation, including microbial degradation of macroplastics and microplastics[48,50], as well as ingestion or digestion of microplastics by macrofauna[51,52]. Accumulation of nanoplastics in a nepheloid layer—which, in some areas in the North Atlantic, can extend up to 800 m above the seabed[53]—as well as resuspension of sediments and the remobilization of potentially deposited nanoplastics may further contribute to elevated nanoplastic concentrations in bottom waters. Plastic mass production began in the 1950s, but the age of subtropical Atlantic bottom waters is >400 years (ref. 54). Deep-water-mass formation and thermohaline convection thus seem unlikely to account for the presence of nanoplastics in bottom waters.

## A mixed-layer nanoplastic mass budget

In the mixed layer within the 'gyre' (stations 1–5), we measured average nanoplastic concentrations of 15.1 mg m$^{-3}$ (6.67 ± 1.12 mg m$^{-3}$ PET, 4.06 ± 1.44 mg m$^{-3}$ PS, 4.32 ± 1.27 mg m$^{-3}$ PVC). These data contrast with previous reports of directly measured macroplastic and microplastic concentrations. At the same stations as measured here, the mass of macroplastic and microplastic (>500 μm; consisting primarily of PE and PP) was found to amount to about 0.11 mg m$^{-3}$ at the sea surface and to <0.02 mg m$^{-3}$ (consisting primarily of PET) at depth >5 m in the mixed layer[7]. Higher microplastic (32–651 μm) mass concentrations of about 1.25 mg m$^{-3}$ (consisting primarily of PP and PE) at the sea surface and 0.62 mg m$^{-3}$ (consisting primarily of PE, PP and PS) at depth >10 m were found at two other stations in the mixed layer of the NASG[6]. Also, recently modelled concentrations of up to 3.4 mg m$^{-3}$ of buoyant macroplastics and microplastics (0.1–1,600.0 mm, primarily PE, PP and PS) at the sea surface of the NASG[1] are lower than our measured nanoplastic concentrations.

To estimate a mixed-layer nanoplastic mass budget, we considered an average climatological mixed-layer depth for November (indicated by the contours in Extended Data Fig. 5c) and the region of the temperate to subtropical North Atlantic. This is bounded by the subpolar gyre north of 55° N and by the southern extent of the NASG at 8.5° N

(Extended Data Figs. 5 and 6). The volume of the climatological mixed layer was 10.1 × 10$^{14}$ m$^3$ for the NASG and 7.01 × 10$^{14}$ m$^3$ for the remaining temperate to subtropical North Atlantic (Extended Data Fig. 5c). As bulk plastic concentration measurements are inherently prone to methodological bias[6,16], the following provides a polymer-specific budget assessment. With respect to our measurements in the mixed layer in the 'gyre' (stations 1–5), the total nanoplastic mass amounts to 15.20 Mt (6.74 ± 1.13 Mt PET, 4.10 ± 1.46 Mt PS, 4.37 ± 1.28 Mt PVC). For the mixed layer in the 'outside gyre' region (stations 6–9), our extrapolation yielded a total nanoplastic mass of 11.73 Mt (5.21 ± 0.84 Mt PET, 2.42 ± 1.09 Mt PS, 4.10 ± 0.96 Mt PVC). This is substantially higher than the recently modelled macroplastic and microplastic mass of buoyant plastic in the mixed layer, amounting to 0.31 Mt for the 'gyre' and to 0.05 Mt for the remaining temperate to subtropical North Atlantic[1].

Owing to the ability of nanoplastic to traverse biological barriers[55], translocate[56], bioaccumulate[25] and interact chemically at rapid rates[57], nanoplastics may represent the most problematic plastic size fraction for ocean life. Notably, most studies assessing the impacts and toxicity of nanoplastics use baseline nanoplastic concentrations that are unsupported by robust environmental measurements. Although mechanisms that contribute to the creation of secondary nanoplastics from parent ocean macroplastics and microplastics have been shown[23,24,39,40], only three studies were able to detect these compounds in the ocean water column[4,5,28]. This study provides, to our knowledge, the first quantitative evidence of the ubiquitous presence of PET, PVC and PS nanoplastics from the mixed-layer to deep-sea bottom waters across the temperate to subtropical North Atlantic. Spatially extrapolated, our measurements strongly suggest that nanoplastics are the largest fraction of the marine plastic mass budget. This implies that the total mass of plastic in the ocean is higher than previously thought, because nanoplastics were not accounted for in marine plastic budget assessments[1,6,8]. Our finding underscores the need to determine the origin, formation and transport of nanoplastics, as well as their further fate in the ocean.

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

## Methods

### Sampling

The samples were collected aboard RV Pelagia during cruise 64PE480 in November 2020. Samples were taken at nine stations along a transect through the temperate to subtropical North Atlantic and at three stations positioned on the European continental shelf (Fig. 1). To enable cross-comparison between different stations, three depths (10 m and 1,000 m water depths and 30 m above the seafloor) were sampled at every deep-ocean station (stations 1–9). Consequently, the actual depth below the sea surface of the deepest sampling point varied as a function of the local water depth. A conductivity, temperature and depth (CTD) sensor phalanx with a rosette sampler comprising an array of 24 polytetrafluoroethylene (PTFE)-lined, PVC Niskin bottles with a volume of 12 l was used for profiling water properties and recovering discrete water samples. During the hydrocast, the Niskin bottles were kept open so that they were flushed with local water during descent and ascent until closure at the desired water depth. Once the CTD sensor was placed on deck, the bottle faucet and tubing used for tapping seawater were thoroughly flushed with sample water before sampling. Then, 2-l glass bottles (Fisherbrand, FB8002000) with PTFE stoppers were rinsed three times with water from the clean deionized water system of the ship and subsequently pre-rinsed (three times) with sample water from the Niskin bottle. Finally, a 2-l aliquot was tapped from the Niskin bottle into the glass bottle and immediately sealed with the stopper. The samples were stored in a dark and cool environment until further analysis in our home laboratories. To safeguard against contamination concerns, we performed a series of field blanks (see the 'Quality assurance and control' section).

### TD-PTR-MS analysis

The water samples were processed in the PTR-MS lab at the Institute for Marine and Atmospheric Research Utrecht. During the time of analysis, the lab was thoroughly cleaned and dedusted on a weekly basis. Typically, only one person was present in the lab during analysis to minimize potential contamination. Blanks were included with every sample batch to account for the risk of airborne contamination. For future work, processing samples in a cleanroom should be considered, although the effectiveness of clean labs in eliminating plastic contamination at the nanoscale is at present uncertain. The 2-l samples were homogenized by shaking the bottle before subsampling. Immediately afterwards, an aliquot of 10 ml was taken from the 2-l glass bottle and stored in a pre-combusted glass chromatography vial (VWR). To separate nanoplastics from microplastics, the 10-ml aliquot was filtered through a 1.0-μm PTFE syringe filter. For further analysis, subsamples were prepared in triplicate, for which 1.5 ml of sample was pipetted into a new pre-combusted glass chromatography vial. The water matrix was removed using an evaporation/sublimation system[58]. The dried samples were introduced to the PTR-MS unit through a thermal desorption system, using a heating protocol defined as follows: starting temperature of 50 °C, followed by a quick increase at 1 °C s$^{-1}$ to 100 °C, then a temperature increase to 200 °C at a rate of 0.19 °C s$^{-1}$ and, finally, the temperature was increased to 360 °C at a rate of 0.44 °C s$^{-1}$. The final dwell time was 1 min at 360 °C. The thermally desorbed compounds were carried by a constant stream of zero air at 50 SCCM to the PTR-ToF-MS instrument (PTR-TOF 8000, Ionicon Analytik). The inlet temperature was set to 180 °C and the drift tube operation parameters were set to 2.90 mbar, 477 V and 120 °C, resulting in an $E/N$ of approximately 120 Td.

### Nanoplastic quantification

The software PTRwid was used to extract the mass spectra[59]. For data reduction, the mass spectra were averaged over a time period of 5 min once the thermal desorption unit reached a temperature of 200 °C, that is, we only considered the time window from 200 °C to 360 °C, during which most of the plastic thermally desorbs. Hence, much of the organic matter matrix was excluded from analysis, as many monomers and most volatile compounds typically desorb at temperatures below 200 °C (refs. 4,33,58). Data integration for oven temperatures from 200 °C to 360 °C not only excludes volatile compounds but also avoids pyrolysis and extensive thermolysis of the sample matrix. Consequently, our method measures collectively free nanoplastics and nanoplastics that are loosely associated to organic matter or that are aggregated, provided that the aggregates pass filter pores (≤1 μm) during prefiltration. To account for background contamination, the mass-specific average of the lab blanks from the corresponding sample batch was subtracted from the averaged nanoplastic masses in the samples. After subtraction, a 3$\sigma$ limit of detection filter was applied, for which the mass-specific signal was set to zero when it did not exceed three times the standard deviation of the lab blanks. The lab blanks consisting of HPLC water (VWR, filtered with 0.2-μm filter, CAS number 7732-18-5) were subjected to similar preparation and analysis as performed for the normal samples. In this manner, we corrected for background noise and possible procedural contamination in the samples. The pre-processed data were subsequently used for nanoplastic fingerprinting against chemically unaltered plastics (the library mass spectra) as described in detail in previous works[4,33]. The fingerprint algorithm compares the spectra against a library comprising the seven most prevalent polymers: PE, PET, PS, PP, PPC, PVC and tyre wear. A matching score of 2$\sigma$ ($z$-score = 2, $P < 0.02275$, one-tail distribution) was considered a positive fingerprint. Algal organic matter may slightly increase false-positive PS detection (see the 'Quality assurance and control' section and *Sargassum* experiment in Extended Data Table 1). To minimize this risk of false-positive annotations, we only considered a $z$-score of 4 or higher as a positive fingerprint match for PS. Matching scores are indicated with * ($z$-score > 2), ** ($z$-score > 3) and *** ($z$-score > 4), for which a higher matching score indicates a better fit with the library mass spectra. We conducted a Monte Carlo analysis to assess the potential interference of organic matter with plastic fingerprinting. The analysis showed that plastic overestimation did not exceed 31% before the match fails (Extended Data Fig. 7). Ion counts were converted to mole fraction using:

$$\text{Mole fraction} = \frac{1}{kt} \times \frac{[\text{MH}^+]}{[\text{H}_3\text{O}^+]} \times \frac{\text{tr}(\text{mH}_3\text{O}^+)}{\text{tr}(\text{mMH}^+)} \tag{1}$$

in which $k$ is the reaction rate coefficient, $t$ the residence time of the primary ions in the drift tube, [MH$^+$] the protonated analyte and [H$_3$O$^+$] the proton donor, hydronium. tr(mH$_3$O$^+$) and tr(mMH$^+$) represent the transmission functions of the hydronium and protonated analyte. The mole fractions were then converted to plastic concentrations (mg m$^{-3}$) by correcting for the sample load and dilution factor. Duplicate measurements instead of triplicate are available for station 9 in the mixed layer, stations 5 and 8 at 1,000 m water depth and station 5 in the bottom-water layer owing to file-corruption issues. Presented nanoplastic concentrations are semiquantitative as not all of the plastic material is eventually converted into detectable ions. This is because of (1) thermal desorption not being perfectly efficient and (2) fractions of the analyte ending up as non-analysable ions. Hence, the reported concentrations represent the lower limit of nanoplastic concentrations. Spike-and-recovery experiments were carried out for PS. Homogenized suspensions of 100 or 200 ng of PS was loaded into a vial along with 1.5 ml of seawater sample. Fingerprinting these spiked samples consistently yielded positive matches for PS with $z$-scores of 4 or higher. By contrast, only 29.4% of the unspiked mixed-layer samples with PS showed $z$-scores of 4 or above. Spiking experiments were performed in triplicate to obtain a reliable recovery rate (Extended Data Table 2). The spiking experiment revealed a recovery/ionization efficiency rate of roughly 7% ± 2.2, which agrees with our previous works[4,33,35]. This entails that the actual PS concentrations in the samples might be 14 times higher. Because of the difficulties in loading precise amounts of plastic

in the nanogram range, spike-and-recovery experiments have not yet been performed for PVC or PET. In a previous study, a linear correction factor of 5.28 ± 1.48 for PS and a nonlinear correction factor between 15.05 ± 0.9 for 59 ng PET load and 26.06 ± 6.8 for 177 ng PET load have been reported[4]. A cross-library correction was applied for PS and PVC concentrations, as these polymer mass spectra partially overlap, resulting in artificially higher PS concentrations when PVC is present and vice versa. These cross-library corrections were calculated on the basis of a 1:1 mixture of 1,000 ng PS and 1,000 ng PVC constructed from library mass spectra which were subsequently fingerprinted.

Moreover, high PS contents were found to lower the PVC matching score, potentially leading to false negatives in PVC detection. This probably affected the surface samples at station 12, at which high amounts of PS but low amounts of PVC were observed. Concentrations of PET were found to be unaffected by the presence of other polymers, owing to its very distinctive mass spectrum.

## Quality assurance and control

Several field blanks were carried out to monitor potential plastic contamination during sampling. We performed field blanks in triplicate at the beginning, middle and end of the cruise, amounting to nine field blanks in total. The Niskin bottles were flushed twice using Milli-Q water and rinsed once more with HPLC water. Then, 2.5 l of HPLC water was poured into the Niskin bottles and left for 1 h in the Niskin bottle to simulate the time that is needed for the CTD sensor to reach the surface of the ocean after closing a Niskin bottle at depth. The Niskin bottle with HPLC water was then sampled in a similar manner as for the normal seawater samples. Field blanks were analysed in the same batches as normal samples. Although we found a low background signal of nanoplastics in the lab blanks ($0.90 ± 1.45$ mg m$^{-3}$ averaged over all polymers and all lab blanks), the field blanks did not contain substantial further nanoplastic contamination (Extended Data Figs. 3 and 4); hence, we concluded that the low concentrations of background nanoplastics originated from the preparation and procedures in our laboratory and not from the sampling procedure. The average nanoplastic background concentration of $0.90 ± 1.45$ mg m$^{-3}$ is low compared with the transect averages of $18.1 ± 2.1$ mg m$^{-3}$ for the mixed layer, $10.9 ± 1.6$ mg m$^{-3}$ for 1,000 m depth and $5.5 ± 0.6$ mg m$^{-3}$ for the bottom layer.

To assess potential false positives from organic matter, we analysed *Sargassum* biomass samples as a proxy for complex organic material. *Sargassum* is abundant in the Sargasso Sea and disperses to other parts of the Atlantic, including the northeast[60]. Approximately 0.5 mm$^3$ of *Sargassum* biomass−collected during our previous campaign and stored frozen−was dried in an oven at 50 °C for 2 h before TD-PTR-MS analysis. The *Sargassum* biomass samples (no digestion applied) showed no positive matches for PE, PP, PET, PVC, or tyre wear particles and only a negligible match for PS, characterized by a low final PS quantity and a low algorithm matching score (see Extended Data Table 1). To maintain a conservative approach, we considered this PS match as a potential false positive in our water samples and, accordingly, increased the PS matching threshold to eliminate such false positives across all samples.

## The missing PE and PP nanoplastic paradox

We could not detect PE and PP nanoplastics in this study (Extended Data Fig. 8). The only other study investigating nanoplastics in surface waters of the NASG (using pyrolysis−gas chromatography−mass spectrometry)[28] could also not find a clear PE signal matching the pyrolytic fingerprint of their PE standard. Neither PE nor PP nanoplastics were reported along Atlantic or Pacific coastlines[5]. This is surprising considering that PE and PP account for about half of the global plastic production[61] and have been found as the most abundant floating polymer types in the ocean, including the NASG[6,7,46]. We cannot fully explain this at present as our method has proved suitable to measure PE and PP−provided the chemical composition remains unaltered−in freshwater, air and marine biota samples[33,35,62,63], in which it was the dominant polymer.

Consequently, possible explanations are the following: (1) the nanoplastics are chemically modified in seawater compared with unaltered polymers so that mass spectrometric fingerprinting cannot detect the modified PE/PP; (2) the concentration of PE and PP nanoplastics were below our detection limit; or (3) the chemical composition of PE or PP is masked by the organic background in ocean water. We cannot rule out any of these explanations. However, through a Monte Carlo analysis (Extended Data Fig. 7), we could indeed show that PE identification was most sensitive to the effect of randomly added organic matter. It also seems very likely that photodegradation not only leads to the production of secondary nanoplastics from parent macroplastics/microplastics[3,24] but that the secondary PE and PP nanoplastics have also undergone some chemical alteration[23,28] (for example, photooxidation introduces carbonyl groups[3]). This might result in a disparity with the diagnostic fingerprint and would explain why the ions typically associated with PE or PP were not detected.

## Calculation of the mixed-layer volume

The dynamic height anomaly (DHA) contours of $\Psi$ (m$^2$ s$^{-2}$) as defined in Section 3.27 of ref. 64 were used to define the NASG:

$$k \times \nabla_P \Psi = fv - fv_{ref} \tag{2}$$

Here $k = (0, 0, 1)$, $f$ is the Coriolis parameter (s$^{-1}$), $v$ is the geostrophic velocity (m s$^{-1}$) with respect to some reference pressure $P_{ref}$ and $v_{ref}$ is the reference velocity at $P_{ref}$. The gradient of the DHA was taken at constant pressure as $\nabla_P \Psi = \left( \frac{\partial \Psi}{\partial x}, \frac{\partial \Psi}{\partial y}, 0 \right)$. For this study we choose $P_{ref} = 1,000$ dbar. This was combined with flow velocities derived from Argo floats at parking level[65]. $\Psi_{ref}$ was defined as the relative DHA, set relative to 1,000 dbar. $\Psi_{ref}$ was defined as the reference DHA, such that the sum

$$\Psi = \Psi_{rel} + \Psi_{ref} \tag{3}$$

equals the DHA. Here $\Psi_{rel}$ can be directly obtained from the thermal wind balance.

To calculate $\Psi_{rel}$, we used the annual mean World Ocean Atlas 2018 1° gridded climatology[66] as input for in situ temperature and practical salinity. This was then converted into conservative temperature (CT) and absolute salinity (SA) using the Gibbs Seawater software toolbox[67]. Both CT and SA were used as input for the gsw_toolbox function 'gsw_geo_strf_dyn_height' to calculate $\Psi_{rel}$ with respect to 1,000 m (Extended Data Fig. 5b). To obtain $\Psi_{ref}$, we constructed an inverse estimate (Extended Data Fig. 6) equated as follows:

$$\Psi_{i+1,j}^{ref} - \Psi_{i,j}^{ref} = \Delta x f v_{i+0.5,j}^{ref} \tag{4}$$

$$\Psi_{i,j+1}^{ref} - \Psi_{i,j}^{ref} = -\Delta y f u_{i,j+0.5}^{ref} \tag{5}$$

Here $i$ represent longitudes and $j$ represents latitudes, both limited to the North Atlantic basin. $\Delta x$ and $\Delta y$ are the related distances and $u$ and $v$ are the eastward and northward velocities, respectively. Each $\Psi_{ref}$ can be included in up to four equations, which can be written as $Ax = b$. Here $x$ are the unknown stream functions, $b$ is the known right-hand side values of equations (4) and (5) and $A$ is a matrix containing −1 or 1 that multiplies the unknown $x$ ($\Psi$) values. This set of equations is solved using MATLAB least-squares minimization machinery given by $x = A \backslash b$, giving the reference DHA $\Psi_{ref}$ (Extended Data Fig. 5a).

To define the NASG, we first considered that the gyre is mostly concentrated in the upper 400 m (Fig. 1 in ref. 68). On the basis of the World Ocean Atlas vertical grid sizes, we averaged over the upper 410 m. The resulting streamlines of the DHA (Extended Data Fig. 6) correspond well to model-based Lagrangian trajectories (Figs. 1d and 3 in ref. 68) and stream function (Fig. 1 in ref. 69). This supports that

the observation-based DHA streamlines calculated here are an accurate indication of the flow field.

To further define the gyre, we selected the last streamline (8 m² s⁻²) that loops from the northern part of the NASG to the southern part without crossing the coast (Extended Data Fig. 6). We used a lower bound latitude cut-off of 8.5° N, as this corresponds with the most western extent of the 8 m² s⁻² contour line. The northern bound of our study region was set at 55° N, as that separates the subpolar area from the temperate to subtropical region in which we sampled. The NASG is then bounded by the 8 m² s⁻² contour (black dots in Extended Data Fig. 5c), whereas the residual area bounded landwards by a 200-m iso-bath is defined as 'outside gyre' (red plusses in Extended Data Fig. 5c).

The climatological mixed-layer depth was calculated[70] using World Ocean Atlas November mean data (Extended Data Fig. 5c). The station mixed-layer depths were calculated from the CTD sensor measurements from this study (Extended Data Fig. 5c). Although the CTD sensor occasionally measured deeper instantaneous mixed-layer depths than the climatological mean, they are within expectations. Therefore, we used the World Ocean Atlas climatological mixed-layer depth values as a first-order estimate to determine the mixed-layer volume both inside and outside the gyre. For the calculation of the macroplastic/microplastic mass inside and outside the NASG, we extracted the modelled concentration values from ref. 1 and overlaid these onto the World Ocean Atlas grid points. This allowed us to make a direct comparison with our nanoplastic data.

### Sensitivity analysis of the fingerprinting algorithm
To evaluate the uncertainty in potential overestimation of our plastic identification approach (for example, owing to the presence of natural organic matter), we performed a Monte Carlo assessment[71]. We simulated the addition of organic matter to the mass spectra of our plastic library and assessed identification and quantification performance. We systematically added 50–350% (increment of 50%) of signal randomly spread over up to 5, 10 and 40 ions of our library used for the identification of nanoplastics. Each sequence of the run was done in 1,000 replicas.

Our Monte Carlo analysis showed that the identification of PET and PS was least affected by the simulated addition of organic matter. We could add 200% of the organic matter in relation to the polymer signal without compromising identification of these two plastics. PVC plastic identification was affected more strongly; addition of more than 100% progressively reduced the plastic identification of the fingerprinting algorithms. PE identification was mostly affected by organic matter presence, for which the recognition of the polymer was greatly affected already when about 50% organic matter was added.

On the other hand, the Monte Carlo analysis showed that the overestimation in all scenarios (different levels of organic matter impurity spread over different numbers of ions) for all plastic polymers did not exceed 31%. For PET, for example, increasing the organic matter background by 100%, 150%, 200% or 250% of the polymer signal, the overestimation was only about 20%, 27%, about 31% (peak) and about 10%, respectively (Extended Data Fig. 7). In other words, if a sample contains a high amount of natural organic matter, the plastic recognition (fingerprint match) is likely to fail before the nanoplastic amount is overestimated by >31%. Thus, we consider our results conservative, with a possible overestimation of roughly 30% owing to the organic matrix effects.

### Data availability
All data (including all stages of data processing) can be downloaded from DAS permanent repository: https://doi.org/10.25850/nioz/7b.b.kj. This study used the YoMaHa'07 (ref. 57) dataset of velocities derived from Argo float trajectories provided by APDRC/IPRC.

The observation-based velocity fields were downloaded from http://apdrc.soest.hawaii.edu/projects/yomaha/. The World Ocean Atlas annual mean data and monthly mean data can be found on the NOAA website (https://www.nodc.noaa.gov/OC5/woa18/). Source data are provided with this paper.

### Code availability
The code used for the gyre mixed-layer volume can be found at https://doi.org/10.25850/nioz/7b.b.kj. The fingerprint codes are published and available at https://doi.org/10.24416/UU01-HKNCGC. The GSW toolbox is available at http://www.teos-10.org/software.htm.

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

**Acknowledgements** We thank the crew and captain of RV Pelagia and the shipboard scientific party for their help at sea. This work has been supported by the Dutch Research Council (grant nos. OCENW.XS2.018 and OCENW.XS21.2.042) and the European Research Council (ERC-CoG grant no. 772923, project VORTEX).

**Author contributions** H.N. and D.M. devised the project, with funding acquired by H.N. and D.M. The principal methodology was developed by D.M. Fieldwork and sampling was done by S.t.H. and H.N. Formal analysis and lab work was performed by S.t.H. in the laboratory of R.H., under the supervision of D.M. and R.H. Data analysis and interpretation was done by S.t.H. and D.M. Determination of the gyre extent and mixed-layer volume was done by S.G. All authors contributed to writing of the paper.

**Funding** Open access funding provided by Helmholtz-Zentrum für Umweltforschung GmbH - UFZ.

**Competing interests** The authors declare no competing interests.

**Additional information**
**Correspondence and requests for materials** should be addressed to Dušan Materić.

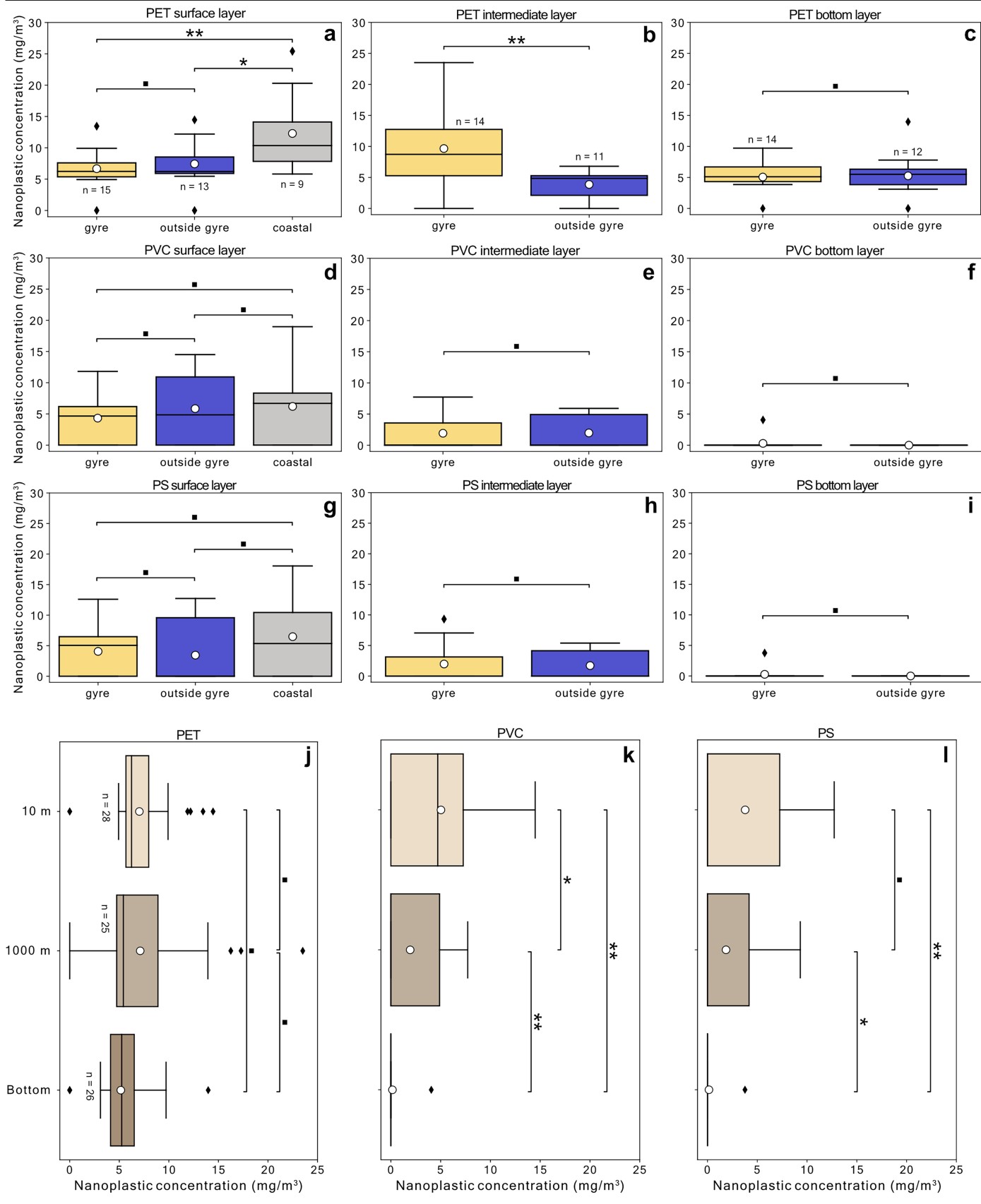

**Extended Data Fig. 1** | See next page for caption.

**Extended Data Fig. 1 | Box plots of nanoplastic polymer distribution in the water column of the North Atlantic.** Average nanoplastic concentrations of PVC, PET and PS for the groups ('gyre'; stations 1–5), the open ocean between the shelf and the gyre ('outside gyre'; stations 6–9) and at the shelf break or on the European shelf ('coastal'; stations 10–12) (**a**–**i**) and for the mixed layer (10 mbsl), intermediate layer (1,000 mbsl) and bottom layer (30 m above the seafloor) for the offshore stations (stations 1–9) (**j**–**l**). The boxes indicate the ±25 percentiles of the median, with the whiskers extending to the data points that fall within the 1.5 interquartiles. Data points that fall outside this range are indicated by a diamond. The mean value is indicated with the white dot. Differences between groups were analysed using a one-way ANOVA test and a $t$-test for means comparison. Significance levels with $P$-values < 0.01 (**), 0.01 < $P$-value < 0.05 (*) and $P$-value > 0.05 (■) are indicated.

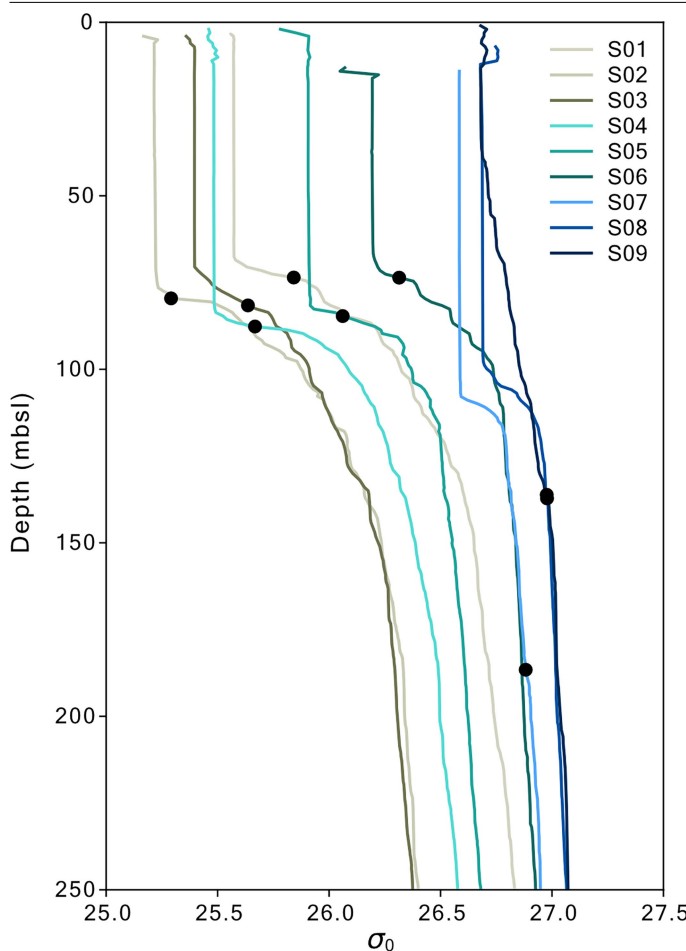

**Extended Data Fig. 2 | Surface-referenced potential density profiles of the upper 250 m at the offshore stations (stations 1–9).** Calculated mixed-layer depth for each offshore station is indicated with a black dot. Data were obtained with a CTD sensor.

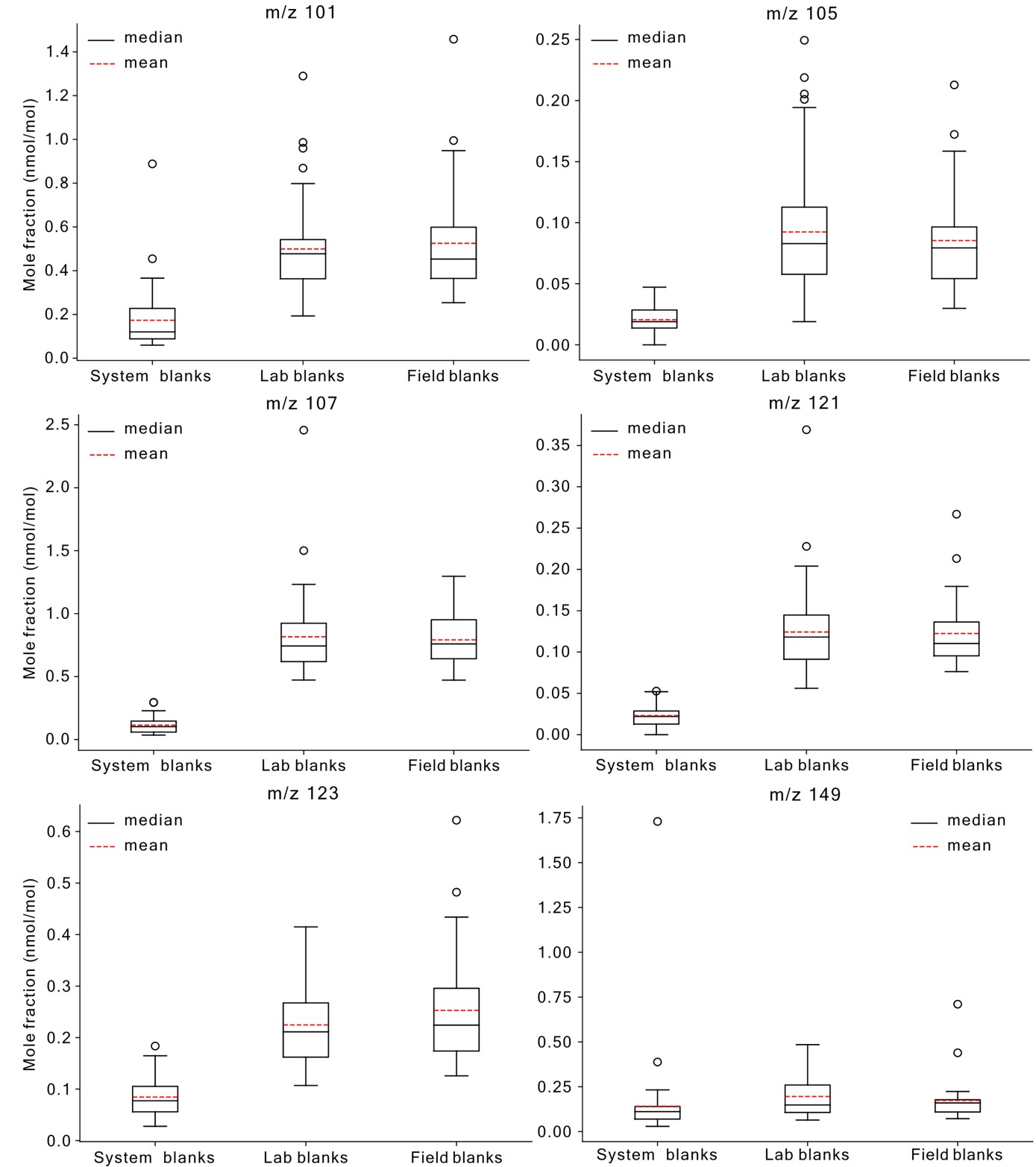

**Extended Data Fig. 3 | Mole fractions of the system, lab and field blanks for six masses that are associated with the presence of plastics.** The boxes indicate the ±25 percentiles of the median, with the whiskers extending to the data points that fall within the 1.5 interquartiles. Data points that fall outside this range are indicated by a circle. The mean value is indicated with the red dashed line. Elevated counts on $m/z$ 101 are associated with the presence of PE, PP and PPC, $m/z$ 105 with PS and PET, $m/z$ 107 with PS and PVC, $m/z$ 121 with PVC and PE, $m/z$ 123 with PET, PP and PPC and $m/z$ 149 with PET and PVC. No more nanoplastic in the field blanks could be detected compared with the lab blanks, ruling out contamination originating from the storage bottles and Niskin bottles.

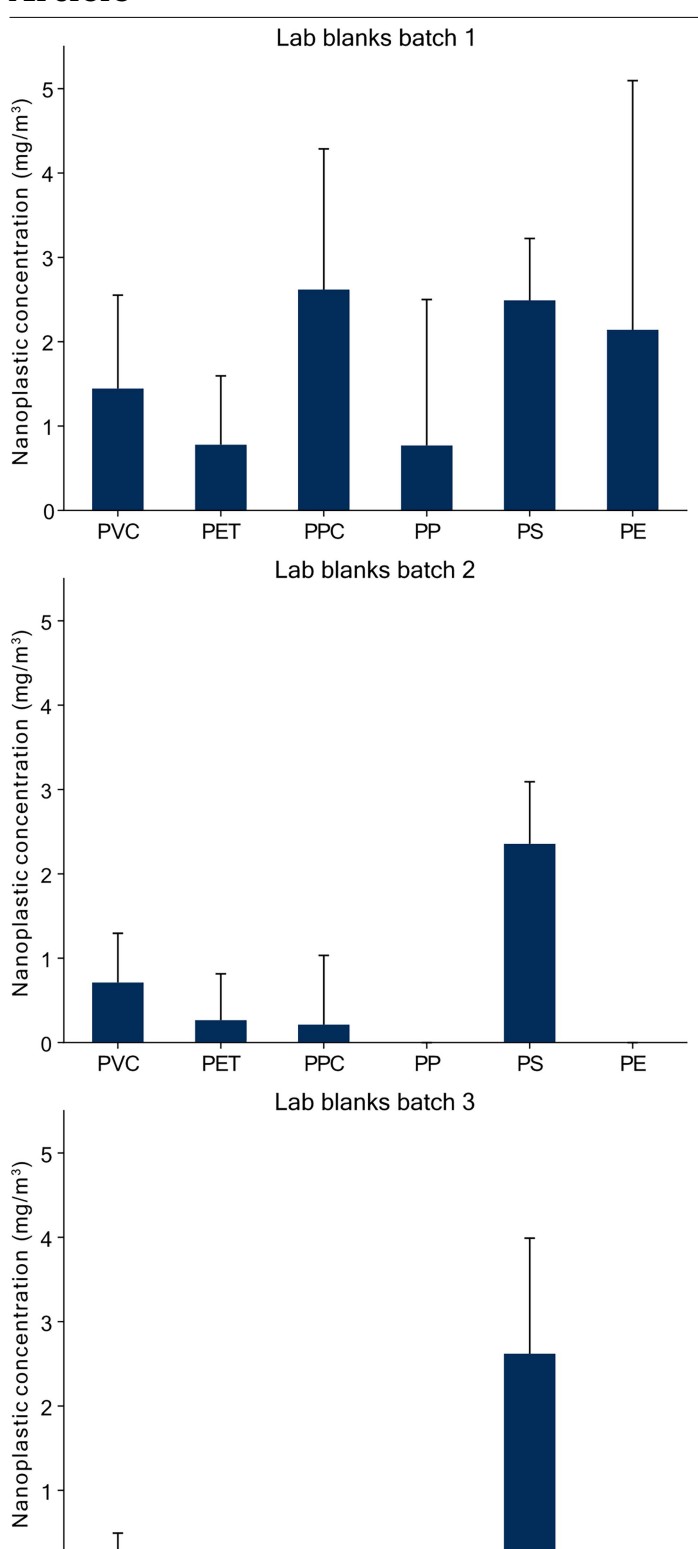

**Extended Data Fig. 4 | Averaged plastic contamination detected in the lab blanks.** The error bars represent the s.d. of the blanks in each batch. All lab blank batches were found to have consistently low average PE, PET, PPC, PP, PS and PE nanoplastic concentrations <3 mg m$^{-3}$. After background subtraction, composed of the mean of the lab blanks of the corresponding batch, still considerable amounts of nanoplastic could be detected in the ocean-water samples. We acknowledge, nonetheless, that the presence of background nanoplastic, although in low amounts, results in further uncertainty of nanoplastic concentrations. Negligible amounts of PET were detected in the lab blanks performed during the measurements of the bottom-water samples (see 'batch 3'), implying that the considerable amounts of PET nanoplastic detected at several kilometres depth are not a result of procedural contamination. However, up to 4 mg m$^{-3}$ of PS has been observed in some of the lab blanks.

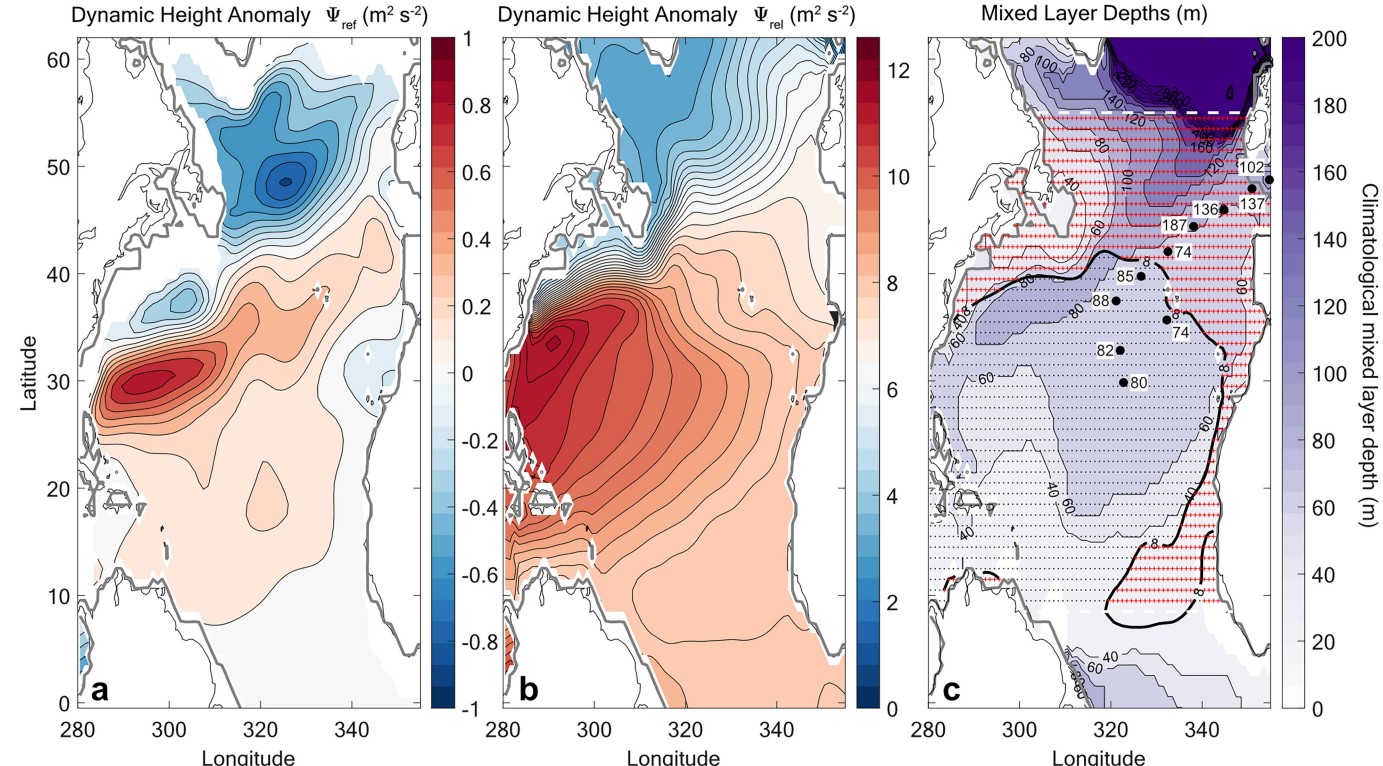

**Extended Data Fig. 5 | DHA contours (m² s⁻²) of $\Psi_{ref}$ at 1,000 m depth (a), the depth-weighted average $\Psi_{rel}$ over 410 m depth (b) and the November climatological and station mixed-layer depths (c).** Note the different scales of the colour maps for panels **a** and **b**. Panel **c** shows the climatological mixed-layer depth from November gridded climatology (purple colours) that was used for the North Atlantic mass-budget calculations, which are in good agreement with the station mixed-layer depths derived from CTD sensor measurements (white boxes) at the stations (large black dots). The small black dots indicate the grid points 'inside the gyre', whereas the red plusses indicate the grid points 'outside the gyre', both bounded by the latitude domain. The thin black contour is the coastline and the grey contour marks the 200-m isobath.

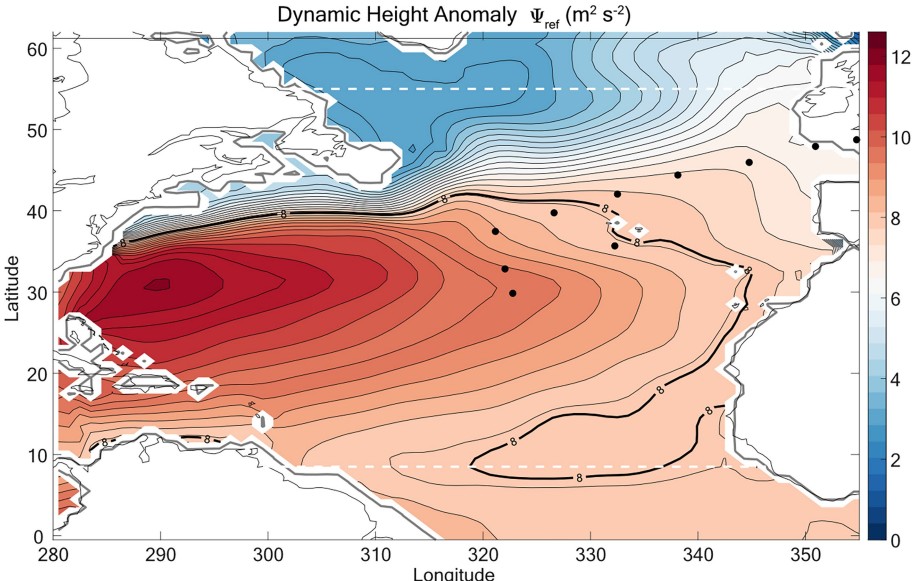

**Extended Data Fig. 6 | DHA contours ($m^2 s^{-2}$, equation (2)).** The thick black contour ($8 m^2 s^{-2}$) marks the outer edge of the gyre. The dashed white lines are the upper (55° N) and lower (8.5° N) latitude bounds of the domain we analysed. The black dots are the cruise stations. The thin black contour is the coastline and the grey contour marks the 200-m isobath. The DHA is a result of averaging over the upper 410 m depth.

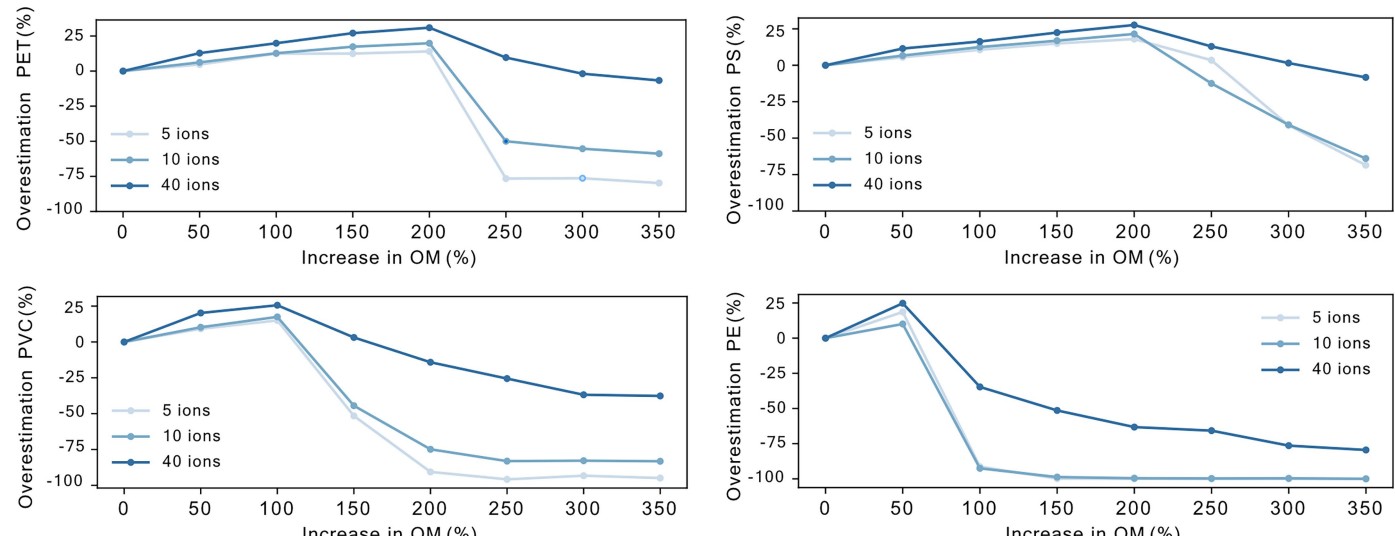

**Extended Data Fig. 7 | Monte Carlo analysis of the simulation of the addition of organic matter and its subsequent influence on the fingerprinting of PET, PS, PVC and PE.** The randomized artificial addition of organic matter (OM) was spread out over 5, 10 or 40 ions that are used for nanoplastic fingerprinting.

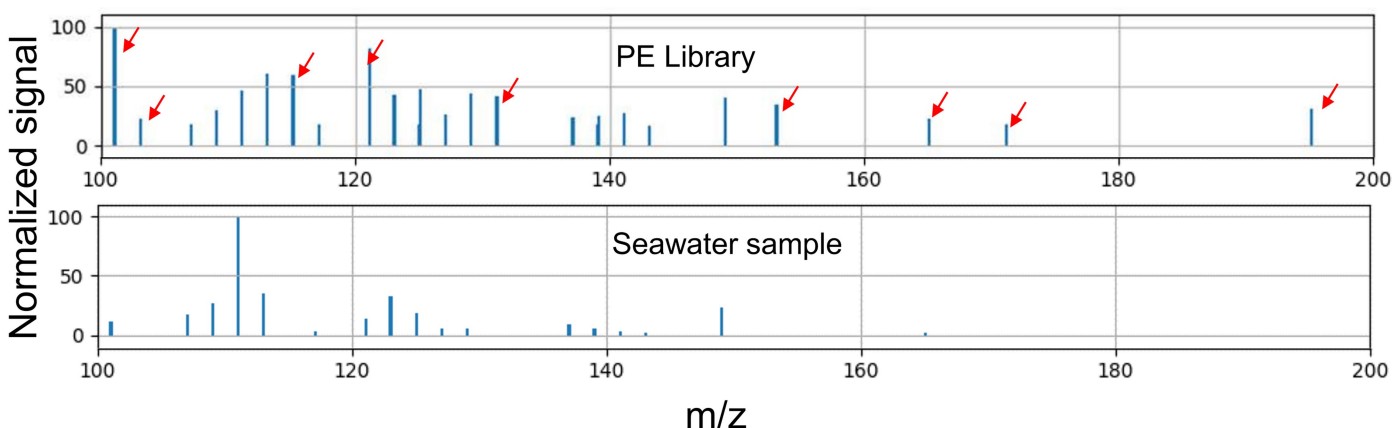

**Extended Data Fig. 8 | The absence of certain ions typically associated with PE in the seawater samples.** Many ion markers typically observed in the mass spectra of PE are completely absent from our samples (indicated by the red arrows). As a result, we cannot definitively determine whether background organic matter is obscuring the PE signal or whether the absence of diagnostic ions indicates that the original PE matrix has been altered (for example, through photooxidation). Regardless, we must conclude that chemically unaltered PE, if present, remains below our detection limit in seawater samples.

**Extended Data Table 1 | Results of the *Sargassum* fingerprinting experiment**

| Sample | ALG1 | ALG3 | Mass |
|---|---|---|---|
| PS_Spike_1-4 | *** | *** | 93.99 ng |
| PS_Spike_2-5 | ** | *** | 28.01 ng |
| PS_Spike_3-6 | ** | *** | 89.83 ng |
| Sargassum Tissue_1-7 | - | ** | 2.41 ng |
| Sargassum Tissue_2-11 | - | * | 8.84 ng |
| Sargassum Tissue_3-15 | - | * | 2.82 ng |

Only for PS, *Sargassum* induced a false-positive fingerprint exclusively with algorithm (ALG) 3. For all other polymers, no positive fingerprint could be generated using *Sargassum* biomass alone.

**Extended Data Table 2 | Results of the PS spike-and-recovery experiments**

| Sample | PS load (ng) | Retrieved (ng) | Recovery/ionization efficiency rate (%) |
|--------|--------------|----------------|------------------------------------------|
| S4 D11 | 100 | 7.2 ± 2.2 | 7.2 ± 2.2 |
| S4 D11 | 200 | 14.0 ± 2.8 | 7.0 ± 1.4 |
| S12 D1 | 200 | 13.0 ± 2.3 | 6.5 ± 1.2 |

Retrieved amount is the calculated average of the triplicates, shown with their standard deviation.