## [Peer Review File · Nature]

High nanoplastic concentrations across the North Atlantic

Corresponding Author: Dr Dušan Materić

Version 0:

Reviewer comments:

Referee #1

(Remarks to the Author)

Summary of the key results:

Materić et al. quantify and identify chemical composition of nanoplastic contaminants in the marine samples collected at 2-3 depths (surface, 1000 m and 5-30 m above the seabed) on a transect crossing the North Atlantic (12 stations in total) from the European shelf to the centre of the North Atlantic gyre. The authors report presence of nanoplastics (<1 µm) at all sampled locations and depths, composed of mainly PS, PVC and PET. They measured elevated concentrations of these polymers in the surface water of the European shelf and inside the gyre. The authors report an overall decreasing vertical gradient of the combined load of these three polymers, with significantly smaller abundance observed close to the seabed. PET dominated the nanoplastics load at all the deep stations. Assuming average concentrations measured in the surface layer, the authors estimated that 33-50 Mt of PET+PVC+PS nanoplastics are present in the top 80 of the North Atlantic basin. Their estimates significantly exceed the load of larger plastic contaminants reported in the ocean. The authors conclude that nanoplastics are the largest contributor to the marine plastic contamination load.

The presence and fate of plastic litter in the marine environment is of major concern with high societal relevance. The measurements of very small, nano-sized plastics (nanoplastics) in the ocean are extremely difficult and thus rare. To date, the studies of nanoplastics in the ocean were focused on measurements in the surface water (e.g. Ter Halle et al. 2017). The work by Materić et al. is the first assessment of nanoplastic contamination on a basin-wide transect covering different geographic and oceanographic regions, and at distinct depth horizons. This study is extremely timely and sheds new light on the extent and fate of plastic contaminants in the marine environment all the way to nano-scale. Although limited to three polymer types and 12 stations, the observations are very important and useful for constraining further the marine plastic budgets and improving the models of dispersion and fate of plastic contaminants in the environment.

Overall, this study might rock field of plastics research, since it puts the attention to the importance of very small (nano-sized) plastic contaminants, which are not captured common with methods as manta trawls, and which are currently recommended to use for monitoring of surface sea water (e.g. GESAMP (2019) Guidelines or the monitoring and assessment of plastic litter and microplastics in the ocean) and are still used to predict the global mass of marine plastics (e.g. Kaandorp et al. 2023). The authors also show that nanoplastics can be of a major importance in the overall context of plastic weight and fate in the ocean compared to small microplastics (e.g. Pabortsava and Lampitt 2020) and larger microplastics captured with nets (> 200-300 microns in size; Kaandorp et al. 2023). These find can have significant implications plastic contaminants in smaller size range (<1 micron) are predicted to be the most pervasive and harmful because of their ability to pass through biological cells and tissues.

The dataset that the authors produce is first-of-its-kind and in that sense very valuable. Especially the depth-resolved stations are unique. However, I have several major points for consideration which preclude me from recommending this manuscript for publication in current form. This study requires significant revisions before being accepted for publication in Nature or Nature family journals.

The authors report three polymer types sampled at 12 stations, they should discuss further how representative their samples/observations are for the chosen region and also discuss them in broader context (including ocean transport and physical properties of the water column which will be different in the gyral, non-gyral and shelf regions).

The extrapolations of the mass-budgets of the three polymer types should be clarified and justified. It is currently unclear, for example, why the authors picked 80 m as a depth interval for the mass budget calculations. The findings of this study should also be evaluated in the context of plastic sources and current understanding of fate of plastics pollutants in general. I am

surprised not to see a reference to the conclusions of Kaandorp et al. 2023 which are so contrasting to the results of this study.

The authors compare their observations with the studies reporting bulk concentrations of plastics. They need to refine their discussion towards polymer-specific data comparison with earlier studies to strengthen their evidence/conclusion. This will also provide a better context for the 'missing PE and PP nanoplastics' reported in this study.

The overall description of sample collection, handling and analysis seem to be appropriate, thus other scientists in the field should be able to replicate this type of nanoplastics measurements. The authors should elaborate on the laboratory conditions where the sample processing was performed, e.g. were the samples handled in the clean laboratory?

I am however not an expert in Thermal Desorption – Proton Transfer Reaction – Mass Spectrometry technique, and specifically in the fingerprinting algorithm used to differentiate synthetic polymer types from natural material. This is interesting given that the authors refer to a potential masking of PE by organic background (line 183). Overall, the description of data processing needs to be more detailed and some essential aspects of it should be included in the methodology. For example, it is unclear whether a higher matching score means a better hit and if a matching score has an upper value. Number of polymers included in the mass spectra library should be included. Based on reference 29, the mass spectral library used in this study consists of 8 polymers (3 of which are variations of PE). I do not see it as a problem, but a limited number of spectra in the library should be acknowledged.

The authors report their values for PS and PVC as underestimates (14 times and higher) based on the recovery of polystyrene (PS), and partly overlapping spectra of PVC and PS. They also report relatively high values of PS in the lab blanks, which can particularly be an issue for samples where PS concentrations were relatively low (< 5 ng/L). The authors should clarify how the blank corrections were performed and include it into quality assurance and control section. The legend of the Extended Data Figure 3 does refer to blank correction but it is unclear whether the correction was polymer-specific or an average value of 0.90 ng/L was used.

The group that station 9 belongs to is inconsistent throughout the manuscript. In lines 71, 94-95, station 9 is a 'shelf break/coastal' station; in lines 113-114, station 9 is pulled into 'open ocean' station, but not in line 193, when 'open ocean' stations are 1-8. This is also inconsistent throughout figure 2, where 'outside gyre' are stations 1-9 and 'shelf break' stations are 9-12.

The authors need to clarify what criteria were used to determine the boundary/extent of the gyre? The choice of the sampling depth for the deep stations need clarification. Why 30 m above the seabed was chosen for the open station and not 5-10 m as for the shelf break stations to make them consistent throughout? How comparable are the data given these differences? For example, the effect of resuspension on the amount of nanoplastics observed at seabed might be more pronounced at shelf stations.

In the current form and until the point of concern are clarified, the data needs to be treated with caution. In addition to clarifications suggested in the sections above, the authors should rework the presentation of their Figures. In Figure 1, grouping of stations will be more obvious if the dots on the map are numbered instead of having a colour code. References to Figure 2 should be improved throughout the text and indicate the subfigure ID.

Figure 2 layout and presentation can also be improved by indicating a group name (inside gyre /outside gyre/shelf) in subfigures a, b, c. It will make a comparison with subfigures d-f easier. All the box plots in Fig. 2 should indicate number of samples pulled per group.

I suggest to present the mass concentration data in units of mass per m³ to be consistent with the units used for budget calculations.

Although appropriate credit to previous work has been given, on number of occasions, references to previous studies are required (see the specific comments).

Statistical analyses have been performed adequately and my only question is whether the authors tested their data for being normally distributed before applying ANOVA test.

The abstract needs significant improvements and be free of references and abbreviations. In the current form it is too detailed, and with some additional information given in parentheses (lines 23-24, and 26). In line 15, 'globally' needs to be removed; the use of 'this domain' is unclear (line 17). Reference to analytical method is not essential (line 20). The article does not have a summary paragraph, and it needs to be added to place the results and their significance into wide context.

Introduction highlights major knowledge gaps but the language needs to be improved. There is a contradiction between lines 36 ('accumulating' plastics) and lines 38-39 where the fate of plastic is referred to as 'manifold' (also confusing word). In line 40, 'accumulating hotspots' need to be rephrased. Subtropical gyres are also zones of convergence (line 41). References are required in lines 49-51. The authors also need to include sentence about why measuring the nanoplastic concentrations is important.

Line(s) 72 What does well-mixed surface layer mean as opposed to surface layer? I suggest to drop 'well-mixed' to avoid

confusion.

Line(s) 72 Start a new paragraph for clarity

Line(s) 131-133 Provide a reference to the 'less polluted' offshore waters – it is unclear whether the authors refer to their nanoplastic observations or earlier studies reporting decreasing gradient in microplastics and macroplastics abundance away from the shore.

Line(s) 142 What is ocean surface boundary level and how is it related to 10 m depth where the samples were collected?

Line(s) 136 Large variations in the inter-gyre abundances of macro and microplastics reported in previous studies should be acknowledged.

Line(s) 154 Where is the depth of winter mixing at all the stations/across the transect? Exchange can happen when mixed layer depth is the highest.

Line(s) 157 Reference needed

Line(s) 161-167 References needed

170-171 Degradation, fragmentation and resuspension of plastics at the seabed to be discussed/acknowledged

Line(s) 185 Reference missing

Line(s) 279 Is there an estimate of thermal desorption efficiency for the polymers analysed?

Line(s) 301 Uncertainty/error/standard deviation missing here

Line(s) 301 and 309 Provide standard deviation

Referee #2

(Remarks to the Author)

The article presents the distribution and quantification of plastic nanoparticles in the North Atlantic Ocean. For this, the authors rely on a technique called Thermal Desorption – Proton Transfer Reaction – Mass Spectrometry. For several reasons mentioned below, I recommend rejecting this paper and validating the analytical method using other complementary techniques, taking into account the colloidal marine organic matter present in the oceans, which is strongly associated with nanoplastics.

To my knowledge, the technique of Thermal Desorption coupled with Proton Transfer Reaction Mass Spectrometry (PTR-MS) is primarily employed for the analysis of organic volatile compounds rather than for the specific detection of nanoplastics. However, the application of this technique for nanoplastics can be considered in certain cases, which are empty such as samples without natural organic matrices. Thermal desorption is a method used to release volatile compounds from a material by heating the sample. PTR-MS, on the other hand, is a mass spectrometry technique that utilizes protonated ions to analyze volatile compounds in real-time. By combining these, it is possible to potentially detect and quantify volatile compounds emitted by nanoplastics when they are heated.

Several points raise questions for me: (i) the significant quantities that may exceed those of organic matter; (ii) the vertical distribution, closely resembling the distribution of organic matter (CDOM/CPOM) found in major oceans; (iii) possible interferences in plastic detection.

However, the authors do not address the potential interferences of colloidal organic matter here. In the literature, it is now proven that nanoplastics strongly form hetero-aggregates with various natural macromolecules. The authors also do not propose hypotheses regarding the potential presence of these nanoparticles. Plastic nanoparticles dispersed alone in the marine environment will quickly aggregate due to the high ionic strength and the presence of strongly complexing divalent cations. The only possibility of having them dispersed is if these particles are associated with marine organic matter capable of maintaining anthropogenic nanoparticles stably in solution.

When using thermal desorption, heating the sample releases volatile compounds present in the material. However, if volatile organic compounds from organic matter are present in the sample, they can also be released and detected by PTR-MS. This situation can lead to signal overlap and make it challenging to distinguish between compounds originating from nanoplastics and those from organic matter. To minimize these interferences, it may be necessary to selectively adjust the ions to specifically target molecular masses associated with nanoplastics in presence of not of organic matter. Does the authors have applied selective digestion protocol to remove natural organic matter? Appropriate sample preparation methods can be used to eliminate as much organic matter as possible that could interfere with the detection of nanoplastics.

Regarding the third point, the authors claim that certain plastics interfere, and the quantification is semi-quantitative and likely significantly underestimated. Given the high quantities of polystyrene (which are underestimated) and the absence of polyethylene/polypropylene, the possible interference of organic matter is also mentioned. The stability of plastics is not addressed either. For example, PET is rapidly hydrolyzed into micro and nanoparticulate forms under marine conditions. The high measured quantities imply particle concentrations far from those produced on a global scale.

Although the authors addressed the validation of their method with laboratory blanks, demonstrating the presence of nanoparticles requires providing details about size, size distribution, or colloidal behavior. Environmental nanometrology involves determining several physico-chemical parameters specific to nanoparticles to attribute the colloidal nature of the material. Given the concentrations stated in the article, the authors could employ electron microscopy or other techniques to directly or indirectly detect and characterize the presence of plastic nanoparticles. There is a crucial lack of information on sample preparation, preventing the validation of the authors' method. From what I understand, the authors filtered the 2 liters of water through a 1-micron PTFE filter, which is also a plastic. Unfortunately, there are no specific details provided.

Even though detecting nanoplastics at different stations is crucial for documenting the distribution of these emerging contaminants, the authors take considerable shortcuts in estimating the total mass of nanoplastics in the global ocean (33-350 million tons). This approximation, while appealing, is purely hypothetical and requires validation through additional

sampling and studies. This is especially true since the authors themselves admit that the method is semi-quantitative.

In summary, while Thermal Desorption coupled with PTR-MS can be useful for studying volatile compounds associated with nanoplastics, it may not provide the necessary specificity for the direct identification of nanoplastics themselves. For publication in Nature, it is crucial to validate the results obtained with other analytical techniques to confirm the presence of nanoplastics. For the specific analysis of nanoplastics, other techniques such as infrared spectroscopy, scanning electron microscopy (SEM), tandem mass spectrometry (MS/MS), or gas chromatography-mass spectrometry (GC-MS) may be useful to cross-validate the results, especially at these levels. These methods allow for a more precise characterization of nanoplastics in terms of chemical composition, size, shape, and other properties.

Version 1:

Reviewer comments:

Referee #1

(Remarks to the Author)

I am still of an opinion that the manuscript by Materic et al. presents the work of high scientific importance and is worthy of publication in a Nature/Nature group journal provided that the methodology is solid and the issues raised in the second review are addressed. The former requires an expert opinion on PTR-MS for measuring nanoplastics, as I have a limited understanding of this technique. My concerns with this technique are i) low recovery rate for PS, ii) unexplained inability to detect PE and PP in marine samples (also in Materic et al. 2022) but not in other environmental media, iii) potential interference of organic matter (e.g. CDOM, phytoplankton) also raised by Reviewer 2. My new set of comments are below and are a point by point reply to the authors labelled as RR x.xx for clarity (also see separate document attached).

R.1.1. We greatly appreciate the reviewer's time and effort in suggesting improvements to our manuscript, as well as their positive assessment of the novelty and impact of our work.

RR. 1.1. Thank you.

The authors report three polymer types sampled at 12 stations, they should discuss further how representative their samples/observations are for the chosen region and also discuss them in broader context (including ocean transport and physical properties of the water column which will be different in the gyral, non-gyral and shelf regions).

R.1.2. Water in the North Atlantic subtropical gyre circles several times around the center before it moves downward and/or northward (Berglund et al 2022). There is a convergence of water towards the middle and subsequent downwelling. Hence, water parcels "trapped" in the gyre will temporarily be subject to different dynamical pathways than water outside of the gyre. Because of this circulatory movement of the upper water mass and our transect crossing these flowlines, our 'gyre' stations are representative for the 'gyre' region. We have further elaborated this in the revised manuscript (see Methods: "Calculation of the North Atlantic surface layer volume"). In the original manuscript, the extrapolation of the 'outside gyre' region was comprising the remainder of the North Atlantic. Upon reviewing oceanographic circulation patterns, we agree that this was not fully justified because of the circulation pattern in the subarctic/arctic North Atlantic featuring its own subpolar gyre. In the revised version of the manuscript, we have now confined the northern limit for extrapolation to 55 °N, as this separates the sub-polar area from the temperate to subtropical region in which we sampled. We did not perform an extrapolation for the 'coastal' region, as nanoplastic concentrations are likely much more heterogeneous there due to the vicinity of contamination (point) sources. Considering the differences in oceanographical properties as well as coastal dynamics, we are confident that the three categories 'gyre', 'outside gyre' and 'coastal' most representative to be used in our statistical analysis.

RR. 1.2.: The authors provided a good description of the physical properties of the gyres (here, the circulation patterns). The gyres also have distinct biogeochemical conditions (e.g. strongly and/or permanently stratified water column, low productivity, etc.) compared to the non-gyral regions, which will impact on the vertical distribution of nanoplastics. This needs to be acknowledged.

The extrapolations of the mass-budgets of the three polymer types should be clarified and justified. It is currently unclear, for example, why the authors picked 80 m as a depth interval for the mass budget calculations.

R.1.3. The motivation and explanation for the extrapolation volume has been improved by the addition of section "Calculation of the North Atlantic surface layer volume" in the new method section. Previously, we based the surface layer depth on our CTD measurements. Now, we improved our extrapolation by including multi-year Argo Float data to better constrain gyre boundaries. Instead of only using our own CTD measurements, we have now calculated the surface layer depth based on the World Ocean Atlas November mean data. Nevertheless, our CTD measurements agree generally well with the multi-year Ocean Atlas data set, so we are confident that we sampled a representative surface layer. This provides a good estimate for the nanoplastic mass budget calculation in the subtropical to temperate North Atlantic Ocean.

RR. 1.3.: It still remains confusing to understand the meaning of the 'surface layer' in this manuscript. In oceanographic context, surface layer could have a number of definitions (e.g. top 5-10 m, photosynthetically active layer, etc.). The authors

define the vertical boundary of the surface layer to the mixed layer depth. To avoid confusion and misinterpretation, I suggest they refer to 'the mixed layer' instead of the 'surface layer'. Also, provide the 'surface'/'mixed layer' depth range for clarity (in Ex. Fig. 5c red numbers are difficult to read).

The findings of this study should also be evaluated in the context of plastic sources and current understanding of fate of plastics pollutants in general. I am surprised not to see a reference to the conclusions of Kaandorp et al. 2023 which are so contrasting to the results of this study.

R.1.4. The authors fully agree that the paper of Kaandorp et al. 2023 should be discussed. Our nanoplastic mass budget findings are now explicitly and in a quantitative matter discussed and compared to the buoyant microplastic concentrations reported in Kaandorp et al., 2023 (see R.1.5. last paragraph of "A nanoplastic mass budget for the surface layer of the North Atlantic"). The Kaandorp data was downloaded and overlaid with the same regions ('gyre' and 'outside gyre') as used in our nanoplastic mass budget calculations. In this manner, we were able to compare the buoyant microplastic budget for the North Atlantic with that of nanoplastic, which supports our claim that the nanoplastic budget likely exceeds that of microplastic. Additionally, we would like to emphasize that the paper of Kaandorp et al. disregards nanoplastics and fitted their model to buoyant microplastic observations, not taking into account for example PVC and PET. The conclusion in Kaandorp et al. that their modeling approach provides a well-constrained ocean plastic budget may thus be valid for microplastic, but not for nanoplastics. Nevertheless, Kaandorp et al. suggested that about ~15% of the annual input of plastic to the ocean is "lost" due to fragmentation (see Fig. 2 in Kaandorp et al.). Fragmentation is basically the grinding of larger plastic pieces to fine microplastics and ultimately nanoplastics. The formation of nanoplastics due to fragmentation has been shown in several studies, with pioneering works by for example in Lambert & Wagner (Chemosphere, 2016), which we have cited already in the original manuscript.

RR. 1.4.: I agree. Yet, it is not only the case of not considering PVC or PET in the work by Kaandorp et al. 2023. Their model only considers plastics >100 microns in size and accounts neither for small microplastics nor nanoplastics, which are more abundant than larger plastics – this need be emphasized further and will add more value to the data presented in this manuscript. In lines of 201-212 it is important to include the lowest size category macro/microplastics reported in studies referenced for the comparison, which will accentuate the importance of the very small nanosized category of plastic contaminants reported in this study.

The authors compare their observations with the studies reporting bulk concentrations of plastics. They need to refine their discussion towards polymer-specific data comparison with earlier studies to strengthen their evidence/conclusion. This will also provide a better context for the 'missing PE and PP nanoplastics' reported in this study.

R.1.5. We appreciate this valuable comment. The authors fully agree that it is more sensible to include a polymer-specific discussion, especially since we could not find PE and PP nanoplastics. The new discussion was updated as follows:

"In the surface layer within the 'gyre' (stations 1-5), we measured average nanoplastic concentrations of 16.3 mg m⁻³ (6.67 ± 1.12 mg m⁻³ PET, 5.32 ± 1.21 mg m⁻³ PS, 4.32 ± 1.27 mg m⁻³ PVC). These contrast previous reports of directly measured macro/microplastic concentrations. At the same stations as measured here, the microplastic mass (consisting primarily of PE and PP) was found to amount to ~0.11 mg m⁻³ at the sea surface and to <0.02 mg m⁻³ (consisting primarily of PET) at depth >5 m in the surface layer². Higher macro/microplastic mass concentrations of ~1.25 mg m⁻³ (consisting primarily of PP and PE) at the sea surface and 0.62 mg m⁻³ (consisting primarily of PE, PP and PS) at depth >10 m in the surface layer were found at two other stations in the NASG¹. Also recently modelled concentrations of up to 3.4 mg m⁻³ buoyant (primarily PE, PP and PS) macro/microplastics at the sea surface of the NASG³ are lower than our measured nanoplastic concentrations.

To estimate a surface layer nanoplastic mass budget, we considered an average surface layer depth for November and the region from the temperate to subtropical North Atlantic. This is bounded by the subpolar gyre north of 55 °N and by the southern extend of the NASG at 8.5 °N (Extended Data Fig. 5 and 7). The volume of the surface layer in the NASG was determined to be 10.1 × 10¹⁴ m³, and 7.01 × 10¹⁴ m³ for the remaining temperate to subtropical North Atlantic (Extended Data Fig. 5c). With respect to our measurements in the surface layer in the 'gyre' (stations 1-5), the total nanoplastic mass amounts to 16.47 million tonnes (Mt) (6.74 ± 1.13 Mt PET, 5.37 ± 1.22 Mt PS, 4.36 ± 1.28 Mt PVC). For the surface layer of the 'outside gyre' region (stations 6-9), our extrapolation yielded a total nanoplastic mass of 13.22 Mt (5.21 ± 0.84 Mt PET, 4.00 ± 0.91 Mt PS, 4.01 ± 0.96 Mt PVC). This is substantially higher than the recently modelled macro/microplastic mass of buoyant plastic in the surface layer amounting to 0.31 Mt in the 'gyre' and to 0.05 Mt in the remaining temperate to subtropical North Atlantic³.

The first observation of ocean nanoplastics was published in 2017 (ref. 25). Concerns about nanoplastics have grown strongly ever since, as nanoplastics have adverse effects on marine life^{6,22,23}. While mechanisms contributing to the creation of nanoplastics from parent ocean macro/microplastics were shown^{20,21,38,39}, only two further studies were able to detect these compounds in the ocean water column^{26,27}. In this study, we provide evidence for the ubiquitous presence of nanoplastics from the surface layer to deep sea bottom waters across the temperate to subtropical North Atlantic. Spatially extrapolated, our measurements strongly suggest that nanoplastics are the largest fraction of the marine plastic mass budget. This implies that the total mass of plastic in the ocean is higher than previously thought^{1,3,4}. Our finding underscores the need to determine the origin, formation and transport of nanoplastics, as well as their further fate in the ocean."

RR.1.5.: Similar to my comment above, reference to the lower size boundary of the previously reported polymer specific

plastics will strengthen the discussion. In addition, reference to Lebreton et al. (2017) and Pabortsava and Lampitt (2020) who advocated the need to address plastic contamination issue on polymer-specific basis will strengthen the discussion and the value of the dataset presented. No method to date can measure all types of plastics, so the concept of bulk concentration is intrinsically/methodologically biased. Similar to Pabortsava and Lampitt (2020), this work shows that the limited number of polymer-specific nanoplastics exceed the mass of bulk macro/microplastics and highlights the critical importance of addressing the smallest category of these contaminants.

'Macro/microplastics' can be referred to as 'larger plastics'.

Discussion in lines 228-238 can be strengthened and include the implications of the most of plastic contamination being in small/nano-size category. This can be linked to behavior and properties of nano-sized contaminants (e.g. ability to pass through biological barriers (cell walls, tissues), translocation, bioaccumulation and speed of chemical interactions (leaching, sorption). We know very little of quantities, properties and behavior of nanoplastics, but they can be the size-fraction that causes the most damage short and long-term. See Gigault et al. 2021 for reference.

Line 230: the authors should emphasize that all studies on harms from nanoplastics were all based on concentrations not supported by real measurements in the environment (they are very scarce and mostly qualitative). Having such measurements (as in this study) is thus critical to reliably assess and understand the real risk from these contaminants now and in the future.

The overall description of sample collection, handling and analysis seem to be appropriate, thus other scientists in the field should be able to replicate this type of nanoplastics measurements. The authors should elaborate on the laboratory conditions where the sample processing was performed, e.g. were the samples handled in the clean laboratory?

R.1.6. At the start of section "TD-PTR-MS analysis" we added the sentence: "The water samples were processed in the PTR-MS lab at IMAU in Utrecht that during the time of analysis was dedicated to nanoplastic research and cleaned on a weekly basis." to clarify this. The laboratory is not a certified clean laboratory, but we did run multiple sampling and process blanks to account for potential contamination during all steps of sample handling and measurement.

RR.1.6.: Thank you for clarifying this. In lines 270-271, what does 'dedicated to nanoplastic research' mean? 'Cleaning on weakly basis' does not really resolve the issue of potential airborne contamination. In their methods section, the authors should recommend future work to be carried out in the certified clean laboratory, which will further minimize plastic contamination of blanks.

I am however not an expert in Thermal Desorption – Proton Transfer Reaction – Mass Spectrometry technique, and specifically in the fingerprinting algorithm used to differentiate synthetic polymer types from natural material. This is interesting given that the authors refer to a potential masking of PE by organic background (line 183).

R.1.7. Matrix effects masking target signals is a common phenomenon in mass spectrometry and could, at least partly explain the lack of observed PE nanoplastics. Please note that for freshwater samples (<https://iopscience.iop.org/article/10.1088/1748-9326/ac68f7/meta>), air samples (<https://doi.org/10.1016/j.chemosphere.2024.141410>) and even marine biota samples (<https://doi.org/10.1038/s43247-024-01300-2>) we could clearly observe PE nanoplastic, and it was the dominant polymer in these samples. It is still unclear to us why we could not identify PE in our ocean water samples. There might be some other unknown processes involved, but this remains speculative. We would also like to stress that, though not quantitatively, a few other studies were able to measure nanoplastics in the ocean (ter Halle et al., 2017, Moon et al. 2024) and these authors could also not find clear signals for PE. In the revised Extended Data we show now a Monte Carlo assessment to demonstrate the effect of the presence of organic matter on the nanoplastic fingerprinting. PE indeed seems sensitive to changes in the ion matrix, with its recognition being affected more dramatically than for other polymers.

To clarify all this, we made the following modifications in the manuscript:

"We cannot fully explain this at present as our method is in principle suitable to measure PE and PP, at least in its pristine configuration^{10,29,31}. Possible explanations are consequently the following: (i) the concentration of PE and PP nanoplastics were below our detection limit, (ii) the chemical configuration of the nanoplastic is masked by the organic background, or (iii) the nanoplastics are chemically different/modified when compared to the standard polymer."

was modified to

"We cannot fully explain this at present as our method has proved suitable to measure PE and PP - at least in its pristine configuration - in freshwater, air and marine biota samples^{32,34,54,55} where it was the dominant polymer. Possible explanations are consequently the following: (i) the nanoplastics are chemically modified in seawater compared to the pristine polymer so that mass spectrometric fingerprinting cannot detect the modified PE/PP, (ii) the concentration of PE and PP nanoplastics were below our detection limit, or (iii) the chemical configuration of PE or PP is masked by the organic background in ocean water. We cannot rule out any of these explanations. However, through a Monte Carlo analysis (Extended Data Fig. 8), we could indeed show that PE identification was most sensitive to the effect of randomly added organic matter. It also seems very likely that photodegradation not only leads to the production of secondary nanoplastics from parent macro/microplastics^{16,21}, but that the secondary PE and PP nanoplastics have also undergone some chemical alteration^{20,25} (e.g., photooxidation introduces carbonyl groups¹⁶). This might result in a disparity with the diagnostic fingerprint and would explain why the ions typically associated with PE or PP are not detected."

Moreover, to emphasize the exclusion of most organic matter through thermal desorption:

"For data reduction the mass spectra were averaged over a time period of 5 minutes once the thermal desorption reached a temperature of 200 °C, hence only the time window from 200°C to 360°C was considered. This resulted in one average value for each mass."

was changed into:

“For data reduction, the mass spectra were averaged over a time period of 5 minutes once the thermal desorption unit reached a temperature of 200 °C, i.e., we only considered the time window from 200°C to 360°C, during which the majority of the plastic thermally desorbed. Hence, the majority of the organic matter matrix was excluded from analysis, as these compounds typically desorb before 200°C. Data integration for oven temperatures above 200°C to 360°C not only excludes volatile compounds, but also avoids pyrolysis and extensive thermolysis. Consequently, our method measures collectively free nanoplastics and nanoplastics that are loosely associated to other OM or that are aggregated (as long as the aggregate size is <1 µm).”

RR.1.7. It is interesting that the only other work published by the lead author on nanoplastics in seawater (the Dutch Wadden Sea, Materic et al. 2022) also reports PS and PET but no PE, PP, and other polymers, unlike other works by the same author on samples of ice and snow, freshwater, and biota. In agreement with Reviewer 2, could the seawater matrix with CDOM and nano/picoplankton (e.g. *Synechococcus*, *E. hax*) impact the detection, by masking PE and PP, and/or by contributing ethylbenzene and benzene groups (i.e. constituents of polystyrene; see Rocco et al. 2021, <https://doi.org/10.1038/s43247-021-00253-0>)? As I mentioned in the first round of reviews, I am not an expert in PTR-MS, but with a limited application of this technique on marine samples with no natural particulates removed/reduced (even at <1 micron size) makes me think that the method favours detection of PS and PET.

The chemical composition of organic matter will also differ between that of biota (zoo-) and phytoplankton – was that a consideration in Monte Carlo assessment? Did the authors know the organic matter load and content of their samples to carry out such assessment, i.e. which compounds were considered?

Line 186 – replace ‘configuration’ to ‘chemical composition’.

Line 294: this requires a reference

Lines 296-298: nanoplastics can also (potentially) aggregate with OM larger than 1 micron in size, so this study probably lost nanoplastics associated with particles >1 micron (e.g. marine snow, faecal pellets). This needs to be included into the discussion. Also see Gigault et al 2021 who proposes that the fate/distribution of nanoplastics might be driven by the association/aggregation with organic matter, e.g. downward transport on nanoplastic to depth).

Overall, the description of data processing needs to be more detailed and some essential aspects of it should be included in the methodology. For example, it is unclear whether a higher matching score means a better hit and if a matching score has an upper value.

R.1.8. Additional info was added to the methods accordingly:

“A matching score of 2σ (z-score = 2, $p < 0.02275$, one tail distribution) was considered a positive fingerprint. Matching scores are indicated with * (z-score > 2), ** (z-score > 3) and *** (z-score > 4), where a higher matching score indicates a better fit with the library mass spectra.”

RR1.8: Lines 331-332: what was the dominant z-score for detected PS and PET in the actual samples?

Number of polymers included in the mass spectra library should be included. Based on reference 29, the mass spectral library used in this study consists of 8 polymers (3 of which are variations of PE). I do not see it as a problem, but a limited number of spectra in the library should be acknowledged.

R.1.9. This is now acknowledged with the following sentence in the methods section:

“The fingerprint algorithm compares the spectra against a library comprising the seven most prevalent polymers: PE, PET, PS, PP, PPC, PVC, and tire wear.”

The authors report their values for PS and PVC as underestimates (14 times and higher) based on the recovery of polystyrene (PS), and partly overlapping spectra of PVC and PS. They also report relatively high values of PS in the lab blanks, which can particularly be an issue for samples where PS concentrations were relatively low (< 5 ng/L). The authors should clarify how the blank corrections were performed and include it into quality assurance and control section. The legend of the Extended Data Figure 3 does refer to blank correction but it is unclear whether the correction was polymer-specific or an average value of 0.90 ng/L was used.

RR. 1.9. Addressed

R.1.10. This was indeed not entirely clear in the preceding version of the manuscript. The correction is mass-specific/ion-specific. i.e., before any fingerprinting is conducted, we firstly process the mass spectra by subtracting the mean of the lab blanks from the sample in the corresponding batch for every m/z. We processed the data in three separate batches (hence the three different figures in Extended data Fig. 3), matching with the three separate periods of measurement. Typically, per batch, we included over 15 lab blanks, from which the lab blank mean was subtracted from the samples. After this subtraction, a 3σ Limit of Detection filter was applied. This means that if the remaining mass signal was not greater than 3 times the standard deviation of the lab blanks from that batch, the value was set to 0 for that m/z. To clarify this procedure, we altered the original text:

“The average of the lab blanks from the corresponding batch was subtracted from these values and a 3σ detection limit was applied.”

as follows:

“To account for background contamination, the mass-specific average of the lab blanks from the corresponding batch was subtracted from the averaged mass of the samples. After subtraction, a 3σ limit of detection filter was applied, whereby the

mass-specific signal was set to zero when it did not exceed 3 times the standard deviation of the lab blanks.”

RR. 1.10. Addressed. In legend of Extended data figure 4 report the units as in the figure (mg m⁻³).

The group that station 9 belongs to is inconsistent throughout the manuscript. In lines 71, 94-95, station 9 is a ‘shelf break/coastal’ station; in lines 113-114, station 9 is pulled into ‘open ocean’ station, but not in line 193, when ‘open ocean’ stations are 1-8. This is also inconsistent throughout figure 2, where ‘outside gyre’ are stations 1-9 and ‘shelf break’ stations are 9-12.

R.1.11. This has now been unified throughout the manuscript and the statistics and figures have been updated accordingly. Based on our improved gyre extent calculation, stations 1-5 are now categorized as ‘gyre’, stations 6-9 as ‘outside gyre’ and stations 10-12 that are on the continental shelf as ‘coastal’.

RR. 1.11. Addressed.

The authors need to clarify what criteria were used to determine the boundary/extent of the gyre?

R.1.12. The ocean gyre is defined based on Dynamic Height Anomaly (DHA) contours, which is a good proxy for streamlines. Ultimately, these calculations are based on a thermal wind balance with a reference velocity. The reference velocity is obtained from an ARGO-float based product, while thermal wind is applied to an observational-based climatology. Hence, we can determine these DHA streamlines and the associated gyre, purely based on observations. When added up, this gives DHA streamlines that correspond well to other (numerical) calculations of the North Atlantic subtropical gyre in other papers. Based on the DHA streamlines we defined the core of the gyre as the part of the gyre that is within the “last” contour that actually loops around the Atlantic Ocean (from the US, towards Europe and back). Exact details of this calculation are now provided in the method section “Calculation of the North Atlantic surface layer volume”.

RR. 1.12. Addressed.

The choice of the sampling depth for the deep stations need clarification. Why 30 m above the seabed was chosen for the open station and not 5-10 m as for the shelf break stations to make them consistent throughout? How comparable are the data given these differences? For example, the effect of resuspension on the amount of nanoplastics observed at seabed might be more pronounced at shelf stations.

R.1.13. 30 meters from the seabed was chosen as a safety margin. The boat’s pitch and roll movement, slight bathymetry changes and drift of the vessel can lead to an unwanted hit of the CTD instrument when sampling >3 km depth. While this can damage the instrument, it, more importantly, can lead to contamination of the bottom sample with sediment particles. This situation is different on the shelf break where the water depth was only <100 meters. We agree that one should be very careful comparing the stations on the shelf break (S10, S11, S12) with the ones in the open ocean. This is also why we made the conscious decision to exclude the three stations on the shelf break from the subsequent ANOVA analysis (Fig. 2f), that only includes S1 to S9.

RR. 1.13. Nepheloid layer created by the resuspension/transport of the seabed sediment by the bottom currents can reach up to 1000 m above the seabed across the Atlantic. This needs to be acknowledged as a potential source of nanoplastics at 30 m depth above the seabed.

In the current form and until the point of concern are clarified, the data needs to be treated with caution.

In addition to clarifications suggested in the sections above, the authors should rework the presentation of their Figures.

In Figure 1, grouping of stations will be more obvious if the dots on the map are numbered instead of having a colour code.

R.1.14. We appreciate the feedback. The figure was updated including station numbering and a new color coding for the different categories used in the statistical analysis.

RR.1.14. add a grey bar in the legend to indicate the coastal/shelf break stations

References to Figure 2 should be improved throughout the text and indicate the subfigure ID.

R.1.15. This was indeed lacking. All references to Fig. 2 now also include the fitting subfigure ID.

Figure 2 layout and presentation can also be improved by indicating a group name (inside gyre /outside gyre/shelf) in subfigures a, b, c. It will make a comparison with subfigures d-f easier. All the box plots in Fig. 2 should indicate number of samples pulled per group.

RR.1.15. Resolved

R.1.16. This feedback has all been incorporated into a new version of Fig. 2. Additionally, we updated the colors to match with the new color coding in Fig. 1:

RR.1.16. Resolved

I suggest to present the mass concentration data in units of mass per m³ to be consistent with the units used for budget calculations.

R.1.17. All our reported units in the text and in the figures have been converted from ng/ml to mg/m³ for better consistency with the budget calculation.

RR.1.17. Resolved

Although appropriate credit to previous work has been given, on number of occasions, references to previous studies are required (see the specific comments).

Statistical analyses have been performed adequately and my only question is whether the authors tested their data for being normally distributed before applying ANOVA test.

R.1.18. To improve the statistical analysis all data points - instead of using a station average - have now been used in the ANOVA analysis (hence the increased n in Fig. 2). The normal distribution has been tested using a Shapiro-Wilk test in JMP. The surface total NP data is normally distributed ($p = 0.040$), the intermediate total NP data is on the verge of being normally distributed ($p = 0.067$) and the bottom PET NP data is also normally distributed ($p = 0.048$).

R.1.18. Resolved

The abstract needs significant improvements and be free of references and abbreviations. In the current form it is too detailed, and with some additional information given in parentheses (lines 23-24, and 26). In line 15, 'globally' needs to be removed; the use of 'this domain' is unclear (line 17). Reference to analytical method is not essential (line 20).

R.1.19. The suggestions to improve readability are greatly valued. We kept a few references since we consider them crucial for our message (a referenced abstract is typical for Nature). We also kept the abbreviations of the polymers as these plastics are commonly often better known by their abbreviated names. We have rewritten the abstract as follows to incorporate the feedback:

"Plastic pollution of the marine realm is widespread, with most scientific attention given to macro and microplastics. Ocean nanoplastics (<1 μm), in contrast, are understudied and the mass budget of this plastic size class thus remains unknown. Here, we measured nanoplastic concentrations on an ocean basin scale along a transect crossing the North Atlantic from the subtropical gyre to the northern European shelf. Our findings revealed substantial amounts of polyethylene terephthalate (PET), polystyrene (PS) and polyvinyl chloride (PVC) nanoplastics ($\sim 2 - 32 \text{ mg m}^{-3}$) throughout the entire water column. We observed higher concentrations of nanoplastics in the surface layer, specifically near the European continent, while nanoplastic accumulation in the subtropical gyre was only weakly visible at intermediate depth. Lowest nanoplastic concentrations, predominantly composed of PET, were found in bottom waters. For the surface layer of the temperate to subtropical North Atlantic, we estimated that the mass of nanoplastic amounts to 30 million tonnes (Mt); 12.0 Mt PET, 9.4 Mt PS and 8.4 Mt PVC. This is in the same range or exceeds previous budget estimates of macro/microplastic for the entire Atlantic^{1,2} or global ocean^{3,4}. Our findings suggest that nanoplastics are the most important part of ocean plastic pollution."

The article does not have a summary paragraph, and it needs to be added to place the results and their significance into wide context.

R.1.20. This is a good suggestion, in addition to synthesizing our data to a North Atlantic nanoplastic budget, we have highlighted now the broader context in the last paragraph of the MS, placing the significance of our findings into a global context (see R.1.5. for the updated discussion text).

RR. 1.19 and 1.20. I still believe that the authors can make both the abstract and the summary paragraph stronger. Nanoplastics are severely understudied (only a few studies exist to date and most of them are qualitative) in the marine environment and this study is the first one to report mass concentrations and estimated budgets of polymer-specific nanoplastics on a basin-wide scale and with depth. The last sentence of the abstract needs to be rephrased with more implications added – marine plastic contamination is now shown to be predominantly in the smallest, nano-size category. This very small size of plastic contaminants may drive completely different physical and chemical interactions and pathways of these contaminants in the marine environment and their impacts therein. In lines 228-230 it should be highlighted that very limited data on the magnitude and characteristics of marine nanoplastic contamination exists to support the laboratory-based studies investigating the effects on nanoplastics on the health of biota/ecosystem functioning.

Introduction highlights major knowledge gaps but the language needs to be improved. There is a contradiction between lines 36 ('accumulating' plastics) and lines 38-39 where the fate of plastic is referred to as 'manifold' (also confusing word).

R.1.21. "is manifold" has been removed resulting in the sentence: "The further fate of plastic in the ocean depends on multiple factors, including the density of the plastic items and their transport at the ocean surface¹⁶."

RR.1.21. I suggest replacing 'accumulating' with 'present'. The introduction paragraph (lines 36-53) needs further revision. I think the introduction will read stronger if it includes the following aspects: i) nanoplastics are the smallest and potentially most pervasive and harmful of plastic contaminants as they can pass through and accumulate in biological cell and tissues; ii) because of their small size, chemical and physical changes and interactions within and on the surfaces on nanoplastics can be faster and completely different to that of their larger precursors (macro/micropastics); authors only mention Brownian motion but it is not sufficient; iii) their small size also makes nanoplastics very challenging to measure/separate from the natural particular matter, especially of a similar size fraction (e.g. colloidal organic and inorganic matter); the authors partly address this in lines 58-59. See Gigault et al. 2021 and Mitrano et al. 2021 for references. We need reliable data on the abundance, characteristics and distribution of nanoplastics in the marine environment to fully understand the extent of plastic contamination, its fate and effects in the environment (partly addressed in lines 60-61).

In line 40, 'accumulating hotspots' need to be rephrased. Subtropical gyres are also zones of convergence (line 41).

R.1.22. This was indeed a mistake in phrasing. The sentence has been rewritten:

"Accumulation hotspots of floating plastics include bays and convergence zones, such as the subtropical ocean gyres^{3,4},..."

RR.1.22. How about 'Floating plastics tend to accumulate in coastal water bays and in the convergence zones of the open ocean, such as subtropical gyres'. Give reference for bays as it is missing in the current version of the manuscript.

References are required in lines 49-51.

R.1.23. We added the following reference to support the statement in these lines:

Sun, H., Jiao, R., & Wang, D. (2021). The difference of aggregation mechanism between microplastics and nanoplastics: Role of Brownian motion and structural layer force. *Environmental Pollution*, 268, 115942.

RR.1.23. Resolved

The authors also need to include sentence about why measuring the nanoplastic concentrations is important.

R.1.24. To emphasize the importance the following sentence was added to the introduction:

"Hence, nanoplastics are not included in any ocean plastic budget estimates^{1,3,4}. This hinders our comprehensive understanding of the environmental impact and potential health hazards of marine plastic pollution."

RR.1.24. I agree; also see comments in RR1.22

Line(s) 72 What does well-mixed surface layer mean as opposed to surface layer? I suggest to drop 'well-mixed' to avoid confusion.

R.1.25. We removed 'well-mixed' from the manuscript to avoid confusion as 'surface layer' indeed already implies the well-mixed layer.

RR.1.25. This terminology is still confusing in the revised manuscript. The authors define 'surface layer' based on the mixed-layer depth. See my earlier comment RR1.3.

Line(s) 72 Start a new paragraph for clarity

R.1.26. A new paragraph was started as follows:

"Samples for nanoplastic analysis were recovered from 12 hydrocast stations of which stations 1-5 were located in the North Atlantic Subtropical Gyre - NASG ('gyre'), stations 6-9 were in the open ocean but outside of the gyre ('outside gyre') and stations 10-12 were on the European shelf ('coastal') (Fig. 1).

Nanoplastics in the surface layer at all 12 hydrocast stations comprised PVC, PET, and PS in the mg m⁻³ range (Fig. 2a),..."

RR.1.26. Resolved

Line(s) 131-133 Provide a reference to the 'less polluted' offshore waters – it is unclear whether the authors refer to their nanoplastic observations or earlier studies reporting decreasing gradient in microplastics and macroplastics abundance away from the shore.

R.1.27. We are referring to our own nanoplastic observations. Therefore, we added a reference to Fig. 2d to clarify this: "Shelf surface waters with comparably high nanoplastic concentrations²⁶ are then entrained with less polluted offshore waters (Fig. 2d),..."

RR.1.27. Resolved

Line(s) 142 What is ocean surface boundary level and how is it related to 10 m depth where the samples were collected?

R.1.28. We aim to refer just to the surface layer, as mentioned before in the manuscript. Hence “ocean” and “boundary” has been removed to improve consistency resulting in: “The moderate difference in nanoplastic concentrations between ‘gyre’ and ‘outside gyre’ stations (Fig. 2d) thus indicates that nanoplastic concentrations in the surface layer might be horizontally homogenized as a result of shear dispersion and wind-induced turbulent mixing^{40,41}.”

RR.1.28. See comments RR. 1.3 and 1.25 when referring to ‘surface’ layer.

Line(s) 136 Large variations in the inter-gyre abundances of macro and microplastics reported in previous studies should be acknowledged.

R.1.29. Nuance was added to line 136 by rephrasing it as follows: “However, floating macro/microplastic generally accumulates in the subtropical gyres^{2,3,4,37}...”

RR.1.29. Change to ‘...macro/microplastics generally accumulate...’; I don’t see the authors mentioning the variability in plastic concentrations within the gyres reported previously.

Line(s) 154 Where is the depth of winter mixing at all the stations/across the transect? Exchange can happen when mixed layer depth is the highest.

R.1.30. The average surface layer depth in the temperate to subtropical ocean seldom exceeds 100-150 m (see for example the globally and annually resolved overview on mixed layer depth presented in de Boyer Montégut et al., JGR, 2004). It thus seems unlikely that the water layer sampled at 1000 m water depth substantially mixes with surface waters.

RR.1.30. Resolved

Line(s) 157 Reference needed

R.1.31. We added the following reference: Boyd, P. W., Claustre, H., Levy, M., Siegel, D. A., & Weber, T. (2019). Multi-faceted particle pumps drive carbon sequestration in the ocean. *Nature*, 568(7752), 327-335.

RR.1.31. Resolved

Line(s) 161-167 References needed

R.1.32. We added the following references to support the statements in these lines:

Gewert, B., Plassmann, M. M., & MacLeod, M. (2015). Pathways for degradation of plastic polymers floating in the marine environment. *Environmental science: processes & impacts*, 17(9), 1513-1521.

Enfrin, M., Lee, J., Gibert, Y., Basheer, F., Kong, L., & Dumée, L. F. (2020). Release of hazardous nanoplastic contaminants due to microplastics fragmentation under shear stress forces. *Journal of hazardous materials*, 384, 121393.

Lv, S., Cui, K., Zhao, S., Li, Y., Liu, R., Hu, R., ... & Shao, Z. (2024). Continuous generation and release of microplastics and nanoplastics from polystyrene by plastic-degrading marine bacteria. *Journal of Hazardous Materials*, 465, 133339.

Dawson, A. L., Kawaguchi, S., King, C. K., Townsend, K. A., King, R., Huston, W. M., & Bengtson Nash, S. M. (2018). Turning microplastics into nanoplastics through digestive fragmentation by Antarctic krill. *Nature communications*, 9(1), 1001.

Zhao, J., Lan, R., Wang, Z., Su, W., Song, D., Xue, R., ... & Xing, B. (2024). Microplastic fragmentation by rotifers in aquatic ecosystems contributes to global nanoplastic pollution. *Nature Nanotechnology*, 19(3), 406-414.

RR.1.32. Resolved

170-171 Degradation, fragmentation and resuspension of plastics at the seabed to be discussed/acknowledged

R.1.33. Since we didn’t measure nanoplastic in the sediments and the initial source of nanoplastic has to be from surface waters we deemed the discussion sufficient as it is. Given the limited word count we have, we prioritized keeping the discussion close to our own measurements.

RR.1.33. I disagree – see the comment on the nepheloid layer RR1.13. Not many words will be required to acknowledge that it can be a source of plastics detected at 30 m above the seabed.

Line(s) 185 Reference missing

R.1.34. We have slightly modified this section, to account for the newly added results from our Monte Carlo analysis. We believe that the further part of this paragraph where we elaborate on the chemical alteration of pristine plastics through photodegradation is sufficiently referenced.

Line(s) 279 Is there an estimate of thermal desorption efficiency for the polymers analysed?

R.1.35. Yes, we performed PS spike-and-recovery experiments of which the results are in Extended Data Tab. 1.

Furthermore, calibration curves for PS and PET have been constructed in a previous study:

<https://doi.org/10.1016/j.scitotenv.2022.157371>. There are no quantitative standards available to test for all plastics, complicating these types of thermal desorption efficiency assessments.

The following sentences have been added to the methods: "Spike-and-recovery experiments were carried out for PS.

Homogenized suspensions of 100 or 200 ng of PS was loaded into a vial along with 1.5 ml of seawater sample.

Fingerprinting these spiked samples consistently yielded positive matches for PS with z-scores of 3 or higher. Spiking experiments were performed in triplicate to obtain a reliable recovery rate (Extended Data Tab. 1). The spiking experiment revealed a recovery/ionization efficiency rate of $\sim 7\% \pm 2.2$, which agrees with our previous works^{26,32,34}. This entails that the actual PS concentrations in the samples might be 14 times higher. Because of the challenge to load precise amounts of plastic in the nanogram range, spike-and-recovery experiments have not yet been performed for PVC or PET. In a previous study, a linear correction factor of 5.28 ± 1.48 for PS and a non-linear correction factor between 15.05 ± 0.9 for 59 ng PET load to 26.06 ± 6.8 for 177 ng PET load have been reported²⁶."

RR.1.35. The reported spike recovery seems very low, compared to that of volatile organic compounds (e.g. Salvador et al. 2016). This needs an expert's opinion (I am not).

Line(s) 301 Uncertainty/error/standard deviation missing here

R.1.36. Standard deviation has been added: "While we found a low background signal of nanoplastics in the lab blanks (0.90 ± 1.45 mg m⁻³ averaged over all polymers and all lab blanks)".

RR.1.36. Resolved

Line(s) 301 and 309 Provide standard deviation

R.1.37. Standard deviation for the blanks and standard errors for the transect averages (as reported in the Main part of the manuscript) have been added: "The average nanoplastic background concentration of 0.90 ± 1.45 mg m⁻³ is low compared with the transect averages of 19.9 ± 2.0 mg m⁻³ for the surface layer, 14.4 ± 1.9 mg m⁻³ for 1000 m depth and 6.2 ± 0.6 mg m⁻³ for the bottom layer."

RR.1.37. Resolved

Referee #3

(Remarks to the Author)

Dear Authors,

this is important work and pushing the boundaries of our abilities to measure plastics to smaller sizes. Being able to robustly quantify nanoplastics is critical to our understanding of the transport, fate, and impact of plastics in the natural environment and also for health science.

Previous reviews have provided excellent summaries and other suggestions that you, the authors, have responded to. The manuscript reads well and all conclusions are robust if the analytical method is measuring nanoplastics. 1 reviewer suggested rejection based upon uncertainty about the analytical method. I share their reservations based on data attained via Py-GC/MS in the North Atlantic Subtropical Gyre.

The data I am describing is unpublished as we are trying to verify the presence of plastic vs other organics via other methods. As the authors describe, we also see PVC and aromatic signatures (PS and PET) in the Py-GC/MS data. The region sampled is the NASG which is also known as the Sargasso Sea due to the prevalence of the Sargassum (floating seaweed). Sargassum is a source of organic matter and found in higher abundance in surface waters of the NASG than plastics. The dissolved organic matter produced from Sargassum is rich in polyphenolics, including halogenated phlorotannins (e.g., Powers et al., 2019; <https://doi.org/10.1029/2019GB006225>). DOM samples in Powers et al. were filtered through 0.2 um filters, so could include polyphenolic-rich nanoparticles from Sargassum. Sargassum-derived particles in the 0.2 to 1 micron range would also presumably contained polyphenolics.

Based on the above, we generated Sargassum Py-GC/MS data, including Sargassum + plastic standards to look for Sargassum-derived pyrolytic products and co-products that could be mistaken for plastics. We are still grappling with the results. I mention this as Sargassum-derived polyphenols and halogenated phlorotannins could potentially produce TD-PTR-MS signatures that overlap with those of other aromatic and halogenated polymers, such as PS, PET and PVC. Thus, I recommend the authors acquire a sample of Sargassum biomass and analyze it to confirm it does not. If not, then the rest of the data and conclusions seem robust.

Without this piece of information, the conclusions seem unsupported and I would recommend rejection if the authors do not commit to test Sargassum signatures.

Version 2:

Reviewer comments:

Referee #1

(Remarks to the Author)

The unique work by Hietbrink et al. presents a step-change in our ability to quantify plastic pollution in the ocean down to the smallest particle size category – nanoplastics – which, as this study now demonstrates, is the most pervasive. The quality of the manuscript improved significantly from the previous submissions and the authors diligently addressed the comments and suggestions raised in the previous rounds of reviews. From methods perspective, the manuscript was strengthened by performing the additional analysis of Sargassum biomass samples to identify whether this type of marine algae could interfere with the detection of PE, PET and PS (per suggestion of Reviewer 2). I would like to point out that the Sargasso Sea (20-35°N and 40-70°W) is outside of the subtropical oligotrophic gyre sampled in this study (stations 1-5), where the community structure (and thus the source of CDOM) is likely dominated by nano- and picoplankton (Organelli et al 2019). However, studies suggest (e.g. Johns et al. 2020) that wind and surface currents disperse Sargassum and thus associated CDOM over the long distances in the Atlantic, including the N-E Atlantic where the samples were collected for this study. I therefore agree that the demonstrated lack of interference from the Sargassum-derived organic matter can support the robustness of the nanoplastic measurements.

I thus recommend Hietbrink et al.'s work for publication in Nature. There just some very minor points that need to be addressed prior to publishing this research:

Line 52 and line 63: As authors correctly suggest, 'most studies assessing the impacts and toxicity of nanoplastics use baseline nanoplastic concentrations unsupported by robust environmental measurements' (lines 247-248). I therefore suggest to refer to the effects from nanoplastics to the environment as 'potential' throughout the manuscript.

Line 59: I suggest to use the word 'spatially' instead of 'geographically'.

Line 197: the term 'pristine configuration' is confusing. I suggest to use, for example, 'unaltered chemical composition' or 'plastic that has not been weathered yet'... or something like that.

Line 290: please give a full name to IMAU

Line 398: please refer to Sargassum biomass, not 'tissue'. Also please give the justification to using Sargassum as the readers may not be aware of this choice for testing the robustness of the method.

Referee #3

(Remarks to the Author)

Main comments based on earlier review:

Thank you for the further explanation of the TD-PTR-MS method and inclusion of a Sargassum sample to assess the potential for Sargassum OM interference. This added check increases confidence in the method and addresses my prior concerns. I now recommend publication.

I re-read the rest of the document, including the response to other reviews. I don't have any material edits to suggest, so I will not comment further on the manuscript.

This is an important, surprising, and concerning result. If NPs are this widespread and at such high concentrations, we need to confirm their quantification (e.g., by applying complimentary methods that provide different views of organic carbon chemistry) and assess their potential impacts on ocean life as well as people (I assume if they are this abundant in the ocean they will also be more prevalent than recognized in media through which people can be exposed – e.g., air, food, drink).

I commend the authors on their work and their positive responses to the critique provided by reviewers.

Regarding Nature's review criteria: The manuscript presents highly original and significant results. The data and the quality of presentation are excellent. Statistics and uncertainties are dealt with appropriately. The work is clearly presented with the response of the authors to other reviews having further improved the manuscript in this regard.

Point to point reply for manuscript **2023-09-17026** entitled
'*High nanoplastic concentrations across the North Atlantic*'

We thank two reviewers for their insightful comments and their thorough review. For better readability we reproduce the reviewer comments and provide our labeled answers (**Rx.xx**) in *cursive* letters.

Referee #1 (Remarks to the Author):

Summary of the key results:

Materic et al. quantify and identify chemical composition of nanoplastic contaminants in the marine samples collected at 2-3 depths (surface, 1000 m and 5-30 m above the seabed) on a transect crossing the North Atlantic (12 stations in total) from the European shelf to the centre of the North Atlantic gyre. The authors report presence of nanoplastics (<1 µm) at all sampled locations and depths, composed of mainly PS, PVC and PET. They measured elevated concentrations of these polymers in the surface water of the European shelf and inside the gyre. The authors report an overall decreasing vertical gradient of the combined load of these three polymers, with significantly smaller abundance observed close to the seabed. PET dominated the nanoplastics load at all the deep stations. Assuming average concentrations measured in the surface layer, the authors estimated that 33-50 Mt of PET+PVC+PS nanoplastics are present in the top 80 of the North Atlantic basin. Their estimates significantly exceed the load of larger plastic contaminants reported in the ocean. The authors conclude that nanoplastics are the largest contributor to the marine plastic contamination load.

The presence and fate of plastic litter in the marine environment is of major concern with high societal relevance. The measurements of very small, nano-sized plastics (nanoplastics) in the ocean are extremely difficult and thus rare. To date, the studies of nanoplastics in the ocean were focused on measurements in the surface water (e.g., Ter Halle et 2017). The work by Materic et al. is the first assessment of nanoplastic contamination on a basin-wide transect covering different geographic and oceanographic regions, and at distinct depth horizons. This study is extremely timely and sheds new light on the extent and fate of plastic contaminants in the marine environment all the way to nano-scale. Although limited to three polymer types and 12 stations, the observations are very important and useful for constraining further the marine plastic budgets and improving the models of dispersion and fate of plastic contaminants in the environment.

Overall, this study might rock field of plastics research, since it puts the attention to the importance of very small (nano-sized) plastic contaminants, which are not captured common with methods as manta trawls, and which are currently recommended to use for monitoring of surface sea water (e.g. GESAMP (2019) Guidelines or the monitoring and assessment of plastic litter and microplastics in the ocean) and are still used to predict the global mass of marine plastics (e.g. Kaandorp et al. 2023). The authors also show that nanoplastics can be of a major importance in the overall context of plastic weight and fate in the ocean compared to small microplastics (e.g., Pabortsava and Lampitt 2020) and larger microplastics captured with nets (> 200-300 microns in size; Kaandorp et al. 2023). These find can have significant implications plastic contaminants in smaller size range (<1 micron) are predicted to be the most pervasive and harmful because of their ability to pass through biological cells and tissues.

The dataset that the authors produce is first-of-its-kind and in that sense very valuable. Especially the depth-resolved stations are unique. However, I have several major points for consideration which preclude me from recommending this manuscript for publication in current form. This study requires significant revisions before being accepted for publication in Nature or Nature family journals.

R.1.1. *We greatly appreciate the reviewer's time and effort in suggesting improvements to our manuscript, as well as their positive assessment of the novelty and impact of our work.*

The authors report three polymer types sampled at 12 stations, they should discuss further how representative their samples/observations are for the chosen region and also discuss them in broader context (including ocean transport and physical properties of the water column which will be different in the gyral, non-gyral and shelf regions).

R.1.2. *Water In the North Atlantic subtropical gyre circles several times around the center before it moves downward and/or northward (Berglund et al 2022). There is a convergence of water towards the middle and subsequent downwelling. Hence, water parcels "trapped" in the gyre will temporarily be subject to different dynamical pathways than water outside of the gyre. Because of this circulatory movement of the upper water mass and our transect crossing these flowlines, our 'gyre' stations are representative for the 'gyre' region. We have further elaborated this in the revised manuscript (see Methods: "Calculation of the North Atlantic surface layer volume"). In the original manuscript, the extrapolation of the 'outside gyre' region was comprising the remainder of the North Atlantic. Upon reviewing oceanographic circulation patterns, we agree that this was not fully justified because of the circulation pattern in the subarctic/arctic North Atlantic featuring its own subpolar gyre. In the revised version of the manuscript, we have now confined the northern limit for extrapolation to 55 °N, as this separates the sub-polar area from the temperate to subtropical region in which we sampled. We did not perform an extrapolation for the 'coastal' region, as nanoplastic concentrations are likely much more heterogeneous there due to the vicinity of contamination (point) sources. Considering the differences in oceanographical properties as well as coastal dynamics, we are confident that the three categories 'gyre', 'outside gyre' and 'coastal' most representative to be used in our statistical analysis.*

The extrapolations of the mass-budgets of the three polymer types should be clarified and justified. It is currently unclear, for example, why the authors picked 80 m as a depth interval for the mass budget calculations.

R.1.3. *The motivation and explanation for the extrapolation volume has been improved by the addition of section "Calculation of the North Atlantic surface layer volume" in the new method section. Previously, we based the surface layer depth on our CTD measurements. Now, we improved our extrapolation by including multi-year Argo Float data to better constrain gyre boundaries. Instead of only using our own CTD measurements, we have now calculated the surface layer depth based on the World Ocean Atlas November mean data. Nevertheless, our CTD measurements agree generally well with the multi-year Ocean Atlas data set, so we are confident that we sampled a*

representative surface layer. This provides a good estimate for the nanoplastic mass budget calculation in the subtropical to temperate North Atlantic Ocean.

The findings of this study should also be evaluated in the context of plastic sources and current understanding of fate of plastics pollutants in general. I am surprised not to see a reference to the conclusions of Kaandorp et al. 2023 which are so contrasting to the results of this study.

R.1.4. *The authors fully agree that the paper of Kaandorp et al. 2023 should be discussed. Our nanoplastic mass budget findings are now explicitly and in a quantitative matter discussed and compared to the buoyant microplastic concentrations reported in Kaandorp et al., 2023 (see R.1.5. last paragraph of “A nanoplastic mass budget for the surface layer of the North Atlantic”). The Kaandorp data was downloaded and overlaid with the same regions (‘gyre’ and ‘outside gyre’) as used in our nanoplastic mass budget calculations. In this manner, we were able to compare the buoyant microplastic budget for the North Atlantic with that of nanoplastic, which supports our claim that the nanoplastic budget likely exceeds that of microplastic. Additionally, we would like to emphasize that the paper of Kaandorp et al. disregards nanoplastics and fitted their model to buoyant microplastic observations, not taking into account for example PVC and PET. The conclusion in Kaandorp et al. that their modeling approach provides a well-constrained ocean plastic budget may thus be valid for microplastic, but not for nanoplastics. Nevertheless, Kaandorp et al. suggested that about ~15% of the annual input of plastic to the ocean is “lost” due to fragmentation (see Fig. 2 in Kaandorp et al.). Fragmentation is basically the grinding of larger plastic pieces to fine microplastics and ultimately nanoplastics. The formation of nanoplastics due to fragmentation has been shown in several studies, with pioneering works by for example in Lambert & Wagner (Chemosphere, 2016), which we have cited already in the original manuscript.*

The authors compare their observations with the studies reporting bulk concentrations of plastics. They need to refine their discussion towards polymer-specific data comparison with earlier studies to strengthen their evidence/conclusion. This will also provide a better context for the ‘missing PE and PP nanoplastics’ reported in this study.

R.1.5. *We appreciate this valuable comment. The authors fully agree that it is more sensible to include a polymer-specific discussion, especially since we could not find PE and PP nanoplastics. The new discussion was updated as follows:*

“In the surface layer within the ‘gyre’ (stations 1-5), we measured average nanoplastic concentrations of 16.3 mg m^{-3} ($6.67 \pm 1.12 \text{ mg m}^{-3}$ PET, $5.32 \pm 1.21 \text{ mg m}^{-3}$ PS, $4.32 \pm 1.27 \text{ mg m}^{-3}$ PVC). These contrast previous reports of directly measured macro/microplastic concentrations. At the same stations as measured here, the microplastic mass (consisting primarily of PE and PP) was found to amount to $\sim 0.11 \text{ mg m}^{-3}$ at the sea surface and to $< 0.02 \text{ mg m}^{-3}$ (consisting primarily of PET) at depth $> 5 \text{ m}$ in the surface layer². Higher macro/microplastic mass concentrations of $\sim 1.25 \text{ mg m}^{-3}$ (consisting primarily of PP and PE) at the sea surface and 0.62 mg m^{-3} (consisting primarily of PE, PP and PS) at depth $> 10 \text{ m}$ in the surface layer were found at two other stations in the NASG¹. Also recently modelled concentrations of up to 3.4 mg m^{-3} buoyant (primarily PE, PP and PS) macro/microplastics at the sea surface of the NASG³ are lower than our measured nanoplastic concentrations.

To estimate a surface layer nanoplastic mass budget, we considered an average surface layer depth for November and the region from the temperate to subtropical North Atlantic. This is bounded by the subpolar gyre north of 55 °N and by the southern extend of the NASG at 8.5 °N (Extended Data Fig. 5 and 7). The volume of the surface layer in the NASG was determined to be $10.1 \times 10^{14} \text{ m}^3$, and $7.01 \times 10^{14} \text{ m}^3$ for the remaining temperate to subtropical North Atlantic (Extended Data Fig. 5c). With respect to our measurements in the surface layer in the ‘gyre’ (stations 1-5), the total nanoplastic mass amounts to 16.47 million tonnes (Mt) ($6.74 \pm 1.13 \text{ Mt PET}$, $5.37 \pm 1.22 \text{ Mt PS}$, $4.36 \pm 1.28 \text{ Mt PVC}$). For the surface layer of the ‘outside gyre’ region (stations 6-9), our extrapolation yielded a total nanoplastic mass of 13.22 Mt ($5.21 \pm 0.84 \text{ Mt PET}$, $4.00 \pm 0.91 \text{ Mt PS}$, $4.01 \pm 0.96 \text{ Mt PVC}$). This is substantially higher than the recently modelled macro/microplastic mass of buoyant plastic in the surface layer amounting to 0.31 Mt in the ‘gyre’ and to 0.05 Mt in the remaining temperate to subtropical North Atlantic³.

The first observation of ocean nanoplastics was published in 2017 (ref. 25). Concerns about nanoplastics have grown strongly ever since, as nanoplastics have adverse effects on marine life^{6,22,23}. While mechanisms contributing to the creation of nanoplastics from parent ocean macro/microplastics were shown^{20,21,38,39}, only two further studies were able to detect these compounds in the ocean water column^{26,27}. In this study, we provide evidence for the ubiquitous presence of nanoplastics from the surface layer to deep sea bottom waters across the temperate to subtropical North Atlantic. Spatially extrapolated, our measurements strongly suggest that nanoplastics are the largest fraction of the marine plastic mass budget. This implies that the total mass of plastic in the ocean is higher than previously thought^{1,3,4}. Our finding underscores the need to determine the origin, formation and transport of nanoplastics, as well as their further fate in the ocean.”

The overall description of sample collection, handling and analysis seem to be appropriate, thus other scientists in the field should be able to replicate this type of nanoplastics measurements. The authors should elaborate on the laboratory conditions where the sample processing was performed, e.g. were the samples handled in the clean laboratory?

R.1.6. At the start of section “TD-PTR-MS analysis” we added the sentence: “The water samples were processed in the PTR-MS lab at IMAU in Utrecht that during the time of analysis was dedicated to nanoplastic research and cleaned on a weekly basis.” to clarify this. The laboratory is not a certified clean laboratory, but we did run multiple sampling and process blanks to account for potential contamination during all steps of sample handling and measurement.

I am however not an expert in Thermal Desorption – Proton Transfer Reaction – Mass Spectrometry technique, and specifically in the fingerprinting algorithm used to differentiate synthetic polymer types from natural material. This is interesting given that the authors refer to a potential masking of PE by organic background (line 183).

R.1.7. Matrix effects masking target signals is a common phenomenon in mass spectrometry and could, at least partly explain the lack of observed PE nanoplastics. Please note that for freshwater samples (<https://iopscience.iop.org/article/10.1088/1748-9326/ac68f7/meta>), air samples (<https://doi.org/10.1016/j.chemosphere.2024.141410>) and even marine biota samples (<https://doi.org/10.1038/s43247-024-01300-2>) we could clearly observe PE nanoplastic, and it was

the dominant polymer in these samples. It is still unclear to us why we could not identify PE in our ocean water samples. There might be some other unknown processes involved, but this remains speculative. We would also like to stress that, though not quantitatively, a few other studies were able to measure nanoplastics in the ocean (ter Halle et al., 2017, Moon et al. 2024) and these authors could also not find clear signals for PE. In the revised Extended Data we show now a Monte Carlo assessment to demonstrate the effect of the presence of organic matter on the nanoplastic fingerprinting. PE indeed seems sensitive to changes in the ion matrix, with its recognition being affected more dramatically than for other polymers.

To clarify all this, we made the following modifications in the manuscript:

“We cannot fully explain this at present as our method is in principle suitable to measure PE and PP, at least in its pristine configuration^{10,29,31}. Possible explanations are consequently the following: (i) the concentration of PE and PP nanoplastics were below our detection limit, (ii) the chemical configuration of the nanoplastic is masked by the organic background, or (iii) the nanoplastics are chemically different/modified when compared to the standard polymer.”

was modified to

“We cannot fully explain this at present as our method has proved suitable to measure PE and PP - at least in its pristine configuration - in freshwater, air and marine biota samples^{32,34,54,55} where it was the dominant polymer. Possible explanations are consequently the following: (i) the nanoplastics are chemically modified in seawater compared to the pristine polymer so that mass spectrometric fingerprinting cannot detect the modified PE/PP, (ii) the concentration of PE and PP nanoplastics were below our detection limit, or (iii) the chemical configuration of PE or PP is masked by the organic background in ocean water. We cannot rule out any of these explanations. However, through a Monte Carlo analysis (Extended Data Fig. 8), we could indeed show that PE identification was most sensitive to the effect of randomly added organic matter. It also seems very likely that photodegradation not only leads to the production of secondary nanoplastics from parent macro/microplastics^{16,21}, but that the secondary PE and PP nanoplastics have also undergone some chemical alteration^{20,25} (e.g., photooxidation introduces carbonyl groups¹⁶). This might result in a disparity with the diagnostic fingerprint and would explain why the ions typically associated with PE or PP are not detected.”

Moreover, to emphasize the exclusion of most organic matter through thermal desorption:

“For data reduction the mass spectra were averaged over a time period of 5 minutes once the thermal desorption reached a temperature of 200 °C, hence only the time window from 200°C to 360°C was considered. This resulted in one average value for each mass.”

was changed into:

“For data reduction, the mass spectra were averaged over a time period of 5 minutes once the thermal desorption unit reached a temperature of 200 °C, i.e., we only considered the time window from 200°C to 360°C, during which the majority of the plastic thermally desorbed. Hence, the majority of the organic matter matrix was excluded from analysis, as these compounds typically desorb before 200°C. Data integration for oven temperatures above 200°C to 360°C not only excludes volatile compounds, but also avoids pyrolysis and extensive thermolysis. Consequently, our

method measures collectively free nanoplastics and nanoplastics that are loosely associated to other OM or that are aggregated (as long as the aggregate size is $<1 \mu\text{m}$)."

Overall, the description of data processing needs to be more detailed and some essential aspects of it should be included in the methodology. For example, it is unclear whether a higher matching score means a better hit and if a matching score has an upper value.

R.1.8. *Additional info was added to the methods accordingly:*

*"A matching score of 2σ (z-score = 2, $p < 0.02275$, one tail distribution) was considered a positive fingerprint. Matching scores are indicated with * (z-score > 2), ** (z-score > 3) and *** (z-score > 4), where a higher matching score indicates a better fit with the library mass spectra."*

Number of polymers included in the mass spectra library should be included. Based on reference 29, the mass spectral library used in this study consists of 8 polymers (3 of which are variations of PE). I do not see it as a problem, but a limited number of spectra in the library should be acknowledged.

R.1.9. *This is now acknowledged with the following sentence in the methods section:*

"The fingerprint algorithm compares the spectra against a library comprising the seven most prevalent polymers: PE, PET, PS, PP, PPC, PVC, and tire wear."

The authors report their values for PS and PVC as underestimates (14 times and higher) based on the recovery of polystyrene (PS), and partly overlapping spectra of PVC and PS. They also report relatively high values of PS in the lab blanks, which can particularly be an issue for samples where PS concentrations were relatively low ($< 5 \text{ ng/L}$). The authors should clarify how the blank corrections were performed and include it into quality assurance and control section. The legend of the Extended Data Figure 3 does refer to blank correction but it is unclear whether the correction was polymer-specific or an average value of 0.90 ng/L was used.

R.1.10. *This was indeed not entirely clear in the preceding version of the manuscript. The correction is mass-specific/ion-specific. I.e., before any fingerprinting is conducted, we firstly process the mass spectra by subtracting the mean of the lab blanks from the sample in the corresponding batch for every m/z . We processed the data in three separate batches (hence the three different figures in Extended data Fig. 3), matching with the three separate periods of measurement. Typically, per batch, we included over 15 lab blanks, from which the lab blank mean was subtracted from the samples. After this subtraction, a 3σ Limit of Detection filter was applied. This means that if the remaining mass signal was not greater than 3 times the standard deviation of the lab blanks from that batch, the value was set to 0 for that m/z . To clarify this procedure, we altered the original text:*

"The average of the lab blanks from the corresponding batch was subtracted from these values and a 3σ detection limit was applied."

as follows:

"To account for background contamination, the mass-specific average of the lab blanks from the corresponding batch was subtracted from the averaged mass of the samples. After subtraction, a 3σ

limit of detection filter was applied, whereby the mass-specific signal was set to zero when it did not exceed 3 times the standard deviation of the lab blanks.”

The group that station 9 belongs to is inconsistent throughout the manuscript. In lines 71, 94-95, station 9 is a ‘shelf break/coastal’ station; in lines 113-114, station 9 is pulled into ‘open ocean’ station, but not in line 193, when ‘open ocean’ stations are 1-8. This is also inconsistent throughout figure 2, where ‘outside gyre’ are stations 1-9 and ‘shelf break’ stations are 9-12.

R.1.11. *This has now been unified throughout the manuscript and the statistics and figures have been updated accordingly. Based on our improved gyre extent calculation, stations 1-5 are now categorized as ‘gyre’, stations 6-9 as ‘outside gyre’ and stations 10-12 that are on the continental shelf as ‘coastal’.*

The authors need to clarify what criteria were used to determine the boundary/extent of the gyre?

R.1.12. *The ocean gyre is defined based on Dynamic Height Anomaly (DHA) contours, which is a good proxy for streamlines. Ultimately, these calculations are based on a thermal wind balance with a reference velocity. The reference velocity is obtained from an ARGO-float based product, while thermal wind is applied to an observational-based climatology. Hence, we can determine these DHA streamlines and the associated gyre, purely based on observations. When added up, this gives DHA streamlines that correspond well to other (numerical) calculations of the North Atlantic subtropical gyre in other papers. Based on the DHA streamlines we defined the core of the gyre as the part of the gyre that is within the “last” contour that actually loops around the Atlantic Ocean (from the US, towards Europe and back). Exact details of this calculation are now provided in the method section “Calculation of the North Atlantic surface layer volume”.*

The choice of the sampling depth for the deep stations need clarification. Why 30 m above the seabed was chosen for the open station and not 5-10 m as for the shelf break stations to make them consistent throughout? How comparable are the data given these differences? For example, the effect of resuspension on the amount of nanoplastics observed at seabed might be more pronounced at shelf stations.

R.1.13. *30 meters from the seabed was chosen as a safety margin. The boat’s pitch and rol movement, slight bathymetry changes and drift of the vessel can lead to an unwanted hit of the CTD instrument when sampling >3 km depth. While this can damage the instrument, it, more importantly, can lead to contamination of the bottom sample with sediment particles. This situation is different on the shelf break where the water depth was only <100 meters. We agree that one should be very careful comparing the stations on the shelf break (S10, S11, S12) with the ones in the open ocean. This is also why we made the conscious decision to exclude the three stations on the shelf break from the subsequent ANOVA analysis (Fig. 2f), that only includes S1 to S9.*

In the current form and until the point of concern are clarified, the data needs to be treated with caution.

In addition to clarifications suggested in the sections above, the authors should rework the presentation of their Figures.

In Figure 1, grouping of stations will be more obvious if the dots on the map are numbered instead of having a colour code.

R.1.14. *We appreciate the feedback. The figure was updated including station numbering and a new color coding for the different categories used in the statistical analysis.*

References to Figure 2 should be improved throughout the text and indicate the subfigure ID.

R.1.15. *This was indeed lacking. All references to Fig. 2 now also include the fitting subfigure ID.*

Figure 2 layout and presentation can also be improved by indicating a group name (inside gyre /outside gyre/shelf) in subfigures a, b, c. It will make a comparison with subfigures d-f easier. All the box plots in Fig. 2 should indicate number of samples pulled per group.

R.1.16. *This feedback has all been incorporated into a new version of Fig. 2. Additionally, we updated the colors to match with the new color coding in Fig. 1:*

I suggest to present the mass concentration data in units of mass per m³ to be consistent with the units used for budget calculations.

R.1.17. All our reported units in the text and in the figures have been converted from ng/ml to mg/m³ for better consistency with the budget calculation.

Although appropriate credit to previous work has been given, on number of occasions, references to previous studies are required (see the specific comments).

Statistical analyses have been performed adequately and my only question is whether the authors tested their data for being normally distributed before applying ANOVA test.

R.1.18. To improve the statistical analysis all data points - instead of using a station average - have now been used in the ANOVA analysis (hence the increased n in Fig. 2). The normal distribution has been tested using a Shapiro-Wilk test in JMP. The surface total NP data is normally distributed ($p = 0.040$), the intermediate total NP data is on the verge of being normally distributed ($p = 0.067$) and the bottom PET NP data is also normally distributed ($p = 0.048$).

The abstract needs significant improvements and be free of references and abbreviations. In the current form it is too detailed, and with some additional information given in parentheses (lines 23-24, and 26). In line 15, 'globally' needs to be removed; the use of 'this domain' is unclear (line 17). Reference to analytical method is not essential (line 20).

R.1.19. The suggestions to improve readability are greatly valued. We kept a few references since we consider them crucial for our message (a referenced abstract is typical for Nature). We also kept the abbreviations of the polymers as these plastics are commonly often better known by their abbreviated names. We have rewritten the abstract as follows to incorporate the feedback:

"Plastic pollution of the marine realm is widespread, with most scientific attention given to macro and microplastics. Ocean nanoplastics ($<1 \mu\text{m}$), in contrast, are understudied and the mass budget of this plastic size class thus remains unknown. Here, we measured nanoplastic concentrations on an ocean basin scale along a transect crossing the North Atlantic from the subtropical gyre to the northern European shelf. Our findings revealed substantial amounts of polyethylene terephthalate (PET), polystyrene (PS) and polyvinyl chloride (PVC) nanoplastics ($\sim 2 - 32 \text{ mg m}^{-3}$) throughout the entire water column. We observed higher concentrations of nanoplastics in the surface layer, specifically near the European continent, while nanoplastic accumulation in the subtropical gyre was only weakly visible at intermediate depth. Lowest nanoplastic concentrations, predominantly composed of PET, were found in bottom waters. For the surface layer of the temperate to subtropical North Atlantic, we estimated that the mass of nanoplastic amounts to 30 million tonnes (Mt); 12.0 Mt PET, 9.4 Mt PS and 8.4 Mt PVC. This is in the same range or exceeds previous budget estimates of macro/microplastic for the entire Atlantic^{1,2} or global ocean^{3,4}. Our findings suggest that nanoplastics are the most important part of ocean plastic pollution."

The article does not have a summary paragraph, and it needs to be added to place the results and their significance into wide context.

R.1.20. This is a good suggestion, in addition to synthesizing our data to a North Atlantic nanoplastic budget, we have highlighted now the broader context in the last paragraph of the MS, placing the significance of our findings into a global context (see **R.1.5.** for the updated discussion text).

Introduction highlights major knowledge gaps but the language needs to be improved. There is a contradiction between lines 36 ('accumulating' plastics) and lines 38-39 where the fate of plastic is referred to as 'manifold' (also confusing word).

R.1.21. "is manifold" has been removed resulting in the sentence: "The further fate of plastic in the ocean depends on multiple factors, including the density of the plastic items and their transport at the ocean surface¹⁶."

In line 40, 'accumulating hotspots' need to be rephrased. Subtropical gyres are also zones of convergence (line 41).

R.1.22. *This was indeed a mistake in phrasing. The sentence has been rewritten:*

"Accumulation hotspots of floating plastics include bays and convergence zones, such as the subtropical ocean gyres^{3,4},..."

References are required in lines 49-51.

R.1.23. *We added the following reference to support the statement in these lines:*

Sun, H., Jiao, R., & Wang, D. (2021). The difference of aggregation mechanism between microplastics and nanoplastics: Role of Brownian motion and structural layer force. Environmental Pollution, 268, 115942.

The authors also need to include sentence about why measuring the nanoplastic concentrations is important.

R.1.24. *To emphasize the importance the following sentence was added to the introduction:*

"Hence, nanoplastics are not included in any ocean plastic budget estimates^{1,3,4}. This hinders our comprehensive understanding of the environmental impact and potential health hazards of marine plastic pollution."

Line(s) 72 What does well-mixed surface layer mean as opposed to surface layer? I suggest to drop 'well-mixed' to avoid confusion.

R.1.25. *We removed 'well-mixed' from the manuscript to avoid confusion as 'surface layer' indeed already implies the well-mixed layer.*

Line(s) 72 Start a new paragraph for clarity

R.1.26. *A new paragraph was started as follows:*

"Samples for nanoplastic analysis were recovered from 12 hydrocast stations of which stations 1-5 were located in the North Atlantic Subtropical Gyre - NASG ('gyre'), stations 6-9 were in the open ocean but outside of the gyre ('outside gyre') and stations 10-12 were on the European shelf ('coastal') (Fig. 1).

Nanoplastics in the surface layer at all 12 hydrocast stations comprised PVC, PET, and PS in the mg m⁻³ range (Fig. 2a),..."

Line(s) 131-133 Provide a reference to the 'less polluted' offshore waters – it is unclear whether the authors refer to their nanoplastic observations or earlier studies reporting decreasing gradient in microplastics and macroplastics abundance away from the shore.

R.1.27. We are referring to our own nanoplastic observations. Therefore, we added a reference to Fig. 2d to clarify this: “Shelf surface waters with comparably high nanoplastic concentrations²⁶ are then entrained with less polluted offshore waters (Fig. 2d), ...”

Line(s) 142 What is ocean surface boundary level and how is it related to 10 m depth where the samples were collected?

R.1.28. We aim to refer just to the surface layer, as mentioned before in the manuscript. Hence “ocean” and “boundary” has been removed to improve consistency resulting in: “The moderate difference in nanoplastic concentrations between ‘gyre’ and ‘outside gyre’ stations (Fig. 2d) thus indicates that nanoplastic concentrations in the surface layer might be horizontally homogenized as a result of shear dispersion and wind-induced turbulent mixing^{40,41}.”

Line(s) 136 Large variations in the inter-gyre abundances of macro and microplastics reported in previous studies should be acknowledged.

R.1.29. Nuance was added to line 136 by rephrasing it as follows: “However, floating macro/microplastic generally accumulates in the subtropical gyres^{2,3,4,37} ...”

Line(s) 154 Where is the depth of winter mixing at all the stations/across the transect? Exchange can happen when mixed layer depth is the highest.

R.1.30. The average surface layer depth in the temperate to subtropical ocean seldom exceeds 100-150 m (see for example the globally and annually resolved overview on mixed layer depth presented in de Boyer Montégut et al., JGR, 2004). It thus seems unlikely that the water layer sampled at 1000 m water depth substantially mixes with surface waters.

Line(s) 157 Reference needed

R.1.31. We added the following reference: Boyd, P. W., Claustre, H., Levy, M., Siegel, D. A., & Weber, T. (2019). Multi-faceted particle pumps drive carbon sequestration in the ocean. *Nature*, 568(7752), 327-335.

Line(s) 161-167 References needed

R.1.32. We added the following references to support the statements in these lines:

Gewert, B., Plassmann, M. M., & MacLeod, M. (2015). Pathways for degradation of plastic polymers floating in the marine environment. *Environmental science: processes & impacts*, 17(9), 1513-1521.

Enfrin, M., Lee, J., Gibert, Y., Basheer, F., Kong, L., & Dumée, L. F. (2020). Release of hazardous nanoplastic contaminants due to microplastics fragmentation under shear stress forces. *Journal of hazardous materials*, 384, 121393.

Lv, S., Cui, K., Zhao, S., Li, Y., Liu, R., Hu, R., ... & Shao, Z. (2024). Continuous generation and release of microplastics and nanoplastics from polystyrene by plastic-degrading marine bacteria. *Journal of Hazardous Materials*, 465, 133339.

Dawson, A. L., Kawaguchi, S., King, C. K., Townsend, K. A., King, R., Huston, W. M., & Bengtson Nash, S. M. (2018). Turning microplastics into nanoplastics through digestive fragmentation by Antarctic krill. *Nature communications*, 9(1), 1001.

Zhao, J., Lan, R., Wang, Z., Su, W., Song, D., Xue, R., ... & Xing, B. (2024). Microplastic fragmentation by rotifers in aquatic ecosystems contributes to global nanoplastic pollution. *Nature Nanotechnology*, 19(3), 406-414.

170-171 Degradation, fragmentation and resuspension of plastics at the seabed to be discussed/acknowledged

R.1.33. *Since we didn't measure nanoplastic in the sediments and the initial source of nanoplastic has to be from surface waters we deemed the discussion sufficient as it is. Given the limited word count we have, we prioritized keeping the discussion close to our own measurements.*

Line(s) 185 Reference missing

R.1.34. *We have slightly modified this section, to account for the newly added results from our Monte Carlo analysis. We believe that the further part of this paragraph where we elaborate on the chemical alteration of pristine plastics through photodegradation is sufficiently referenced.*

Line(s) 279 Is there an estimate of thermal desorption efficiency for the polymers analysed?

R.1.35. *Yes, we performed PS spike-and-recovery experiments of which the results are in Extended Data Tab. 1. Furthermore, calibration curves for PS and PET have been constructed in a previous study: <https://doi.org/10.1016/j.scitotenv.2022.157371>. There are no quantitative standards available to test for all plastics, complicating these types of thermal desorption efficiency assessments.*

The following sentences have been added to the methods: "Spike-and-recovery experiments were carried out for PS. Homogenized suspensions of 100 or 200 ng of PS was loaded into a vial along with 1.5 ml of seawater sample. Fingerprinting these spiked samples consistently yielded positive matches for PS with z-scores of 3 or higher. Spiking experiments were performed in triplicate to obtain a reliable recovery rate (Extended Data Tab. 1). The spiking experiment revealed a recovery/ionization efficiency rate of $\sim 7\% \pm 2.2$, which agrees with our previous works^{26,32,34}. This entails that the actual PS concentrations in the samples might be 14 times higher. Because of the challenge to load precise amounts of plastic in the nanogram range, spike-and-recovery experiments have not yet been performed for PVC or PET. In a previous study, a linear correction factor of 5.28 ± 1.48 for PS and a non-linear correction factor between 15.05 ± 0.9 for 59 ng PET load to 26.06 ± 6.8 for 177 ng PET load have been reported²⁶."

Line(s) 301 Uncertainty/error/standard deviation missing here

R.1.36. Standard deviation has been added: “While we found a low background signal of nanoplastics in the lab blanks ($0.90 \pm 1.45 \text{ mg m}^{-3}$ averaged over all polymers and all lab blanks)”.

Line(s) 301 and 309 Provide standard deviation

R.1.37. Standard deviation for the blanks and standard errors for the transect averages (as reported in the Main part of the manuscript) have been added: “The average nanoplastic background concentration of $0.90 \pm 1.45 \text{ mg m}^{-3}$ is low compared with the transect averages of $19.9 \pm 2.0 \text{ mg m}^{-3}$ for the surface layer, $14.4 \pm 1.9 \text{ mg m}^{-3}$ for 1000 m depth and $6.2 \pm 0.6 \text{ mg m}^{-3}$ for the bottom layer.”

Referee #2 (Remarks to the Author):

The article presents the distribution and quantification of plastic nanoparticles in the North Atlantic Ocean. For this, the authors rely on a technique called Thermal Desorption – Proton Transfer Reaction – Mass Spectrometry. For several reasons mentioned below, I recommend rejecting this paper and validating the analytical method using other complementary techniques, taking into account the colloidal marine organic matter present in the oceans, which is strongly associated with nanoplastics.

To my knowledge, the technique of Thermal Desorption coupled with Proton Transfer Reaction Mass Spectrometry (PTR-MS) is primarily employed for the analysis of organic volatile compounds rather than for the specific detection of nanoplastics. However, the application of this technique for nanoplastics can be considered in certain cases, which are empty such as samples without natural organic matrices. Thermal desorption is a method used to release volatile compounds from a material by heating the sample. PTR-MS, on the other hand, is a mass spectrometry technique that utilizes protonated ions to analyze volatile compounds in real-time. By combining these, it is possible to potentially detect and quantify volatile compounds emitted by nanoplastics when they are heated.

R.2.1. TD-PTR-MS is indeed a novel technique for measuring nanoplastic identity and quantity. However, we would like to refer the reviewer to earlier articles where TD-PTR-MS was successfully used to quantitatively measure nanoplastics in snow (Materić et al., EST, 2020), freshwater (Materić et al., Environ. Res. Lett., 2022), coastal marine waters (Materić et al., STOTEN, 2022) and marine organisms (Fraissinet et al., Nature’s COMMSENV, 2024). In these publications, spike and recovery with standards were conducted, validating the suitability of TD-PTR-MS for measuring nanoplastics in environmental samples.

We kindly disagree with the reviewer that TD-PTR-MS has been developed for volatile OM, only. Indeed, we have used this for volatile OM (e.g. Park et al., Science; 2013; Kramshøj et al., Nat Com, 2018), but the community has also been using TD-PTR-MS for semi-volatiles compounds (Materić et al., Sci Reports, 2018) and now for nanoplastics, too (see references provided above). Moreover, please note that all thermoanalytical methods, including PyGC-MS and TED-GC-MS also use semivolatile organic markers to detect plastics, so TD-PTR-MS in that sense uses the same principle.

As described in the methods section, water samples were first filtered through a $1 \mu\text{m}$ filter, removing any particles and aggregates larger than $1 \mu\text{m}$. The analysis was then conducted on the organic matter that passed through this filter, i.e., free/colloidal matter as well as small hetero aggregates ($<1 \mu\text{m}$). Our analytical technique primarily discriminates against particle sizes larger than $1 \mu\text{m}$ and

hence targets nanoplastics. The technique does, however, not distinguish free/colloidal nanoplastics from those that may have undergone heteroaggregation into aggregates smaller than 1 μm (note that pyrolysis GC-MS, another method used for nanoplastic identification and quantification, is associated with the same limitation). While it would be interesting to resolve this, it does not preclude us from quantitatively informing about nanoplastic concentrations in sea water, which is the focus of the manuscript.

Several points raise questions for me: (i) the significant quantities that may exceed those of organic matter; (ii) the vertical distribution, closely resembling the distribution of organic matter (CDOM/CPOM) found in major oceans; (iii) possible interferences in plastic detection.

R.2.2. (i) We make sure that the nanoplastic quantities exceed those of organic matter using filtering, thermal desorption optimization and fingerprinting based on library plastic spectra. The majority of the particulate organic matter is excluded in the first step where we filter 10 ml of the sample through a 1.0 μm PTFE syringe filter. The thermal desorption protocol was optimized for ocean water samples before we did our routine measurements, where the temperature ramping was adjusted in such a way that the compounds associated with the organic matter matrix and with plastics desorbed at different times. The library spectra we use for fingerprinting are based on 40 ions associated with pristine plastic, allowing us to differentiate between the organic matter matrix and the nanoplastic. See also **R.2.3.** for an in-depth explanation of these principles. (ii) since we are looking at extremely small particles, it is very well possible that similar dispersion processes govern the transport of nanoscopic POC as well as that of nanoplastic, resulting in comparable vertical profiles. We would like to emphasize here though that in our results we could clearly detect different vertical distribution profiles for PET, PVC and PS, supporting that we can distinguish between these compounds. (iii) We acknowledge there is to some extent possible interference in plastic detection. This has now been investigated in more detail through a Monte Carlo analysis (added in the Extended Data). In short, we found that overestimation due to interference does not exceed 31% before the fingerprint match fails.

However, the authors do not address the potential interferences of colloidal organic matter here. In the literature, it is now proven that nanoplastics strongly form hetero-aggregates with various natural macromolecules.

R.2.3. The reviewer is correct, TD-PTR-MS measures collectively free and aggregated nanoplastics. However, firstly, also hetero aggregated nanoplastics and those associated to other OM can be considered as nanoplastics (Gigault et al., *Env Pol*, 2018) as long as the aggregate is $<1 \mu\text{m}$. Secondly, the principle and advantages of TD-PTR-MS applied to nanoplastics detection are that we can reduce the organic matrix influence in multiple ways:

- 1) we discard highly volatile compounds by applying slow TD and integrating the data when the oven temperature was above 200°C. Thus, all OM that has a boiling point below 200°C is not considered.
- 2) we avoid pyrolysis and extensive thermolysis. We don't heat above 360°C to avoid mobilization of non-volatile/colloidal OM. This way, we avoid all OM that has a boiling point above this value. So, these practices result in a minimal influence of the organic matrix.
- 3) We use 40 ions for fingerprinting (polymer recognition) and quantification, which is very conservative and prevents overestimation.

4) our PTR-MS is TOF with a reasonably high resolution, which means we can differentiate multiple ions at integer mass units. For example, we can differentiate three signals close to styren molecule, which improves our selectivity.

5) As quality control, we routinely spike our samples with a known amount of nanoplastics (e.g., PS spheres) and evaluate if the matrix has a negative effect. In conclusion, our findings are representative of free nanoplastics as well as those loosely aggregated as long as the aggregate is <1 μm and alterations in the chemical nature are not severe (our fingerprint algorithm would exclude strongly modified matter). We have added some explanations in the material and methods section.

The authors also do not propose hypotheses regarding the potential presence of these nanoparticles.

R.2.4. *We kindly disagree with the reviewer. We provided explanations regarding the origin of nanoplastics (i.e., transported from land, atmospheric deposition, ongoing macro/microplastic degradation in the ocean). However, our data do not allow us to go further than this, as our measurements do not enable us to determine the specific source of the nanoplastics (e.g., whether a PET nanoparticle originates from a drinking bottle or from clothing). Additionally, since it is unclear to what degree nanoplastics are deposited from the atmosphere versus drifted to the location with currents, we cannot determine backtrack trajectories.*

Plastic nanoparticles dispersed alone in the marine environment will quickly aggregate due to the high ionic strength and the presence of strongly complexing divalent cations. The only possibility of having them dispersed is if these particles are associated with marine organic matter capable of maintaining anthropogenic nanoparticles stably in solution.

R.2.5. *Any nanoplastics aggregating to the point of forming particles larger than 1 μm are excluded by our technique. However, we also measure nanoplastics loosely associated with other organic matter (see our comments above). Furthermore, nanoparticles in seawater are unlikely to be composed of fully pristine plastic. Weathering through photodegradation, for example, will likely alter the particle surfaces. UV-aged nanoplastics exhibit higher colloid stability in seawater, probably due to the oxygen-containing functional groups induced into the polymer during photodegradation (Liu et al., Water Res., 2019). Our findings of high amounts of nanoparticles thus call for more research on the origin, stability, and further fate of nanoplastics in the ocean.*

When using thermal desorption, heating the sample releases volatile compounds present in the material. However, if volatile organic compounds from organic matter are present in the sample, they can also be released and detected by PTR-MS. This situation can lead to signal overlap and make it challenging to distinguish between compounds originating from nanoplastics and those from organic matter. To minimize these interferences, it may be necessary to selectively adjust the ions to specifically target molecular masses associated with nanoplastics in presence of not of organic matter.

R.2.6. *Yes, we totally agree that selectivity is the key. We had that in mind when developing our method. For the matrix influence, we applied strict quality control to prevent false positives and overestimation (see comments R.2.3 and R.2.7). For instance, we cleared the PTR-MS signal from*

unwanted OM by focusing our analysis on the volatility range of plastics using a slow TD process (e.g., we integrate the signal when TD reached 200°C, thus ignoring volatile organic matrix interference and stopped ~5min after the temperature was kept at 360°C, avoiding non-volatile matrix). We use reasonably high-resolution mass spectrometry, and our data analysis is focused on key 40 ions emitted from each plastic type (unlike other methods that usually use 1-4 ions), and if any of the ions do have an overlap with OM, we can perform the correction if the overall polymer recognition is positive.

Does the authors have applied selective digestion protocol to remove natural organic matter? Appropriate sample preparation methods can be used to eliminate as much organic matter as possible that could interfere with the detection of nanoplastics.

R.2.7. For these relatively low OM samples, after the optimisation of the TD ramp, we decided it was not necessary to involve preprocessing with a digestion protocol, as we could clearly see the fingerprint. The fingerprint recognition in the presence of a high OM sample would have resulted in a false negative (we would not be able to detect any plastics). We also applied our method successfully in coastal waters with a much higher OM background (Materić et al., STOTEN, 2022). For another project, we used a digestion protocol, e.g., for marine organisms (Nature's COMMSENV: <https://www.nature.com/articles/s43247-024-01300-2>).

Furthermore, in the revised Extended Data we now show a Monte Carlo assessment to demonstrate the effect of the presence of organic matter on the nanoplastic fingerprinting. Thus, there is an entire new section devoted to this:

"Sensitivity analysis of the fingerprinting algorithm

To evaluate the uncertainty in potential overestimation of our plastic identification approach (e.g. due to the presence of natural organic matter), we performed a Monte Carlo assessment¹. We simulated the addition of organic matter to the mass spectra of our plastic library and assessed identification and quantification performance. We systematically added 50-350% (increment of 50%) of signal randomly spread over up to 5, 10, and 40 ions of our library used for the identification of nanoplastics. Each sequence of the run was done in 1000 replicas.

Our Monte Carlo analysis showed that the identification of PET and PS was least affected by the simulated addition of OM. We could add 200% of the organic matter in relation to the polymer signal without compromising identification of these two plastics. PVC plastic identification was affected more strongly; addition of more than 100% progressively reduced the plastic identification of the fingerprinting algorithms. PE identification was mostly affected by organic matter presence, where the recognition of the polymer was dramatically affected already when ~50% OM was added.

On the other hand, the Monte Carlo analysis showed that the overestimation in all scenarios (different levels of OM impurity spread over different numbers of ions) for all plastic polymers did not exceed 31%. For PET, for example, increasing the OM background by 100%, 150%, 200% or 250% of the polymer signal, the overestimation was only ~20%, 27%, ~31% (peak) and ~10% respectively (Extended Data Fig. 7). In other words, if a sample contains a high amount of natural organic matter, the plastic recognition (fingerprint match) is likely to fail before the nanoplastic amount is

overestimated by >31%. Thus, we consider our results conservative, with a possible overestimation of ~30% due to the organic matrix effects.

1. Thompson, K. M., Burmaster, D. E. & Crouch, E. A. C. Monte Carlo Techniques for Quantitative Uncertainty Analysis in Public Health Risk Assessments. *Risk Analysis* 12, 53–63 (1992).“

Regarding the third point, the authors claim that certain plastics interfere, and the quantification is semi-quantitative and likely significantly underestimated. Given the high quantities of polystyrene (which are underestimated) and the absence of polyethylene/polypropylene, the possible interference of organic matter is also mentioned.

The stability of plastics is not addressed either. For example, PET is rapidly hydrolyzed into micro and nanoparticulate forms under marine conditions.

R.2.8. *We agree with the reviewer. This might be the very reason why we (and others) observed higher PET nanoplastics in the water column. The presence of PET nanoplastics in the deepsea is also recently confirmed by others using Raman (Moon et al., Science Advances, 2004; doi.org/10.1126/sciadv.adh1675), as mentioned in our manuscript. This supports our finding.*

The high measured quantities imply particle concentrations far from those produced on a global scale.

R.2.9. *According to the Plastic Europe report global PET production for 2021 was 24.2 MT, and this figure excludes the production of PET for fibers for the clothing industry (which is estimated to be more than 50% of the total PET market). Our upscaled estimate of a stock of ~20 Mt of PET nanoplastics in the top 80m of the North Atlantic basin is thus not unrealistic, considering the continued high production of PET in the last 50 years. Note that PET is the common polymer found as fibers or floating debris in marine surface water (and due to the UV and abrasion is likely a continuous and high source of PET nanoplastics).*

Although the authors addressed the validation of their method with laboratory blanks, demonstrating the presence of nanoparticles requires providing details about size, size distribution, or colloidal behavior. Environmental nanometrology involves determining several physico-chemical parameters specific to nanoparticles to attribute the colloidal nature of the material. Given the concentrations stated in the article, the authors could employ electron microscopy or other techniques to directly or indirectly detect and characterize the presence of plastic nanoparticles.

R.2.10. *We kindly disagree with the reviewer. Though details on size and shape are interesting parameters, methods that can determine these are not quantitative. Other publications found nanoplastics using optical methods (e.g., Ter Halle et al., 2017, Moon et al., 2024), it thus seems not unlikely that we would find nanoparticles with such methods. Our TD-PTR-MS method has been validated in previous publications (see R.2.1). The microscopy methods suggested by the reviewer provide a ‘real picture’ of nanoplastics, which is interesting in itself. However, conducting these methods requires a totally different sampling and processing approach than we have done. Furthermore, they are not useful to ‘cross calibrate’ our method as they are not quantitative. Using*

other methods would also involve other challenges; for instance, with electron microscopy small particles can be detected, but it is not possible to confirm that the particle is indeed a nanoplastic particle. In conclusion, while a method comparison study would be interesting, it would deviate the focus of this manuscript, which is to quantitatively measure nanoplastics across the North Atlantic Ocean.

There is a crucial lack of information on sample preparation, preventing the validation of the authors' method. From what I understand, the authors filtered the 2 liters of water through a 1-micron PTFE filter, which is also a plastic. Unfortunately, there are no specific details provided.

R.2.11. *We sampled about two liters of water, which were stored in glass flasks. This was then subsampled and the subsample was indeed filtered through a 1 μm PTFE filter. The reviewer is correct that PTFE is indeed plastic, but the molecular structure is different than PET, PS and PVC. Potential leaching from the filter would be detected with our mass spectrometry method and the signal would be distinct from the plastic types discussed here. We have assessed contamination from labware, sampling and handling by assessing procedural blanks and field blanks. i.e., our blanks were exposed to all the labware and all the impurities in the same way as the samples (this procedure is also carried out by other labs working on nanoplastics). These results are reported in the quality control section. In fact, all blanks showed that contamination was minor in comparison to the nanoplastic levels in the samples. The relevant sections “Quality assurance and control” and “Data processing and nanoplastic quantification” respectively read as follows:*

“...Field blanks were analysed in the same batches as regular samples. While we found a low background signal of nanoplastics in the lab blanks ($0.90 \pm 1.45 \text{ mg m}^{-3}$ averaged over all polymers and all lab blanks), the field blanks did not contain significant additional nanoplastic contamination (Extended Data Fig. 3 and 4); hence, we conclude the low concentrations of background nanoplastics originated from the preparation and procedures in our laboratory and not from the sampling procedure. The average nanoplastic background concentration of $0.90 \pm 1.45 \text{ mg m}^{-3}$ is low compared with the transect averages of $19.9 \pm 2.0 \text{ mg m}^{-3}$ for the surface layer, $14.4 \pm 1.9 \text{ mg m}^{-3}$ for 1000 m depth and $6.2 \pm 0.6 \text{ mg m}^{-3}$ for the bottom layer.”

“The lab blanks consisting of HPLC water (VWR, filtered with $0.2 \mu\text{m}$, CAS 7732-18-5) were subjected to similar preparation and analysis as performed for the regular samples. In this manner, we corrected the samples for background noise and possible procedural contamination.”

Even though detecting nanoplastics at different stations is crucial for documenting the distribution of these emerging contaminants, the authors take considerable shortcuts in estimating the total mass of nanoplastics in the global ocean (33-350 million tons). This approximation, while appealing, is purely hypothetical and requires validation through additional sampling and studies. This is especially true since the authors themselves admit that the method is semi-quantitative.

R.2.12. *The reviewer is correct, budget estimates suffer from extrapolation biases: the higher the data density, the lower the extrapolation bias. It needs to be stressed, however, that extrapolation is a common approach to determine budgets, such as microplastic budgets determined by e.g., Pabortsava and Lampitt, Nat Com, 2020 Or Kaandorp et al., Nat Geosci, 2024. For example,*

Pabortsava and Lampitt extrapolated their microplastic budget for the entire North Atlantic from 6 individual stations. We use 9 stations. We have furthermore also limited our extrapolation to the temperate to subtropical North Atlantic, which is covered by our sampling strategy. We have excluded the shelf region from our extrapolation as our sampling scheme cannot account for the dynamics of the shelf. Likewise, we have not integrated nanoplastic concentrations over depth (i.e., beyond the surface layer depth) as the different deeper water masses are also not well resolved with our sampling approach. We have now added further discussion and modeling to constrain the oceanographic setting (extent of the gyre, surface layer depth), to constrain extent and depth of the surface layer.

Regarding the semi-quantitative nature of our measurements, it should be understood as the low threshold of the real value, which extensively explained in our previous works, e.g., in Materic et al 2022 we wrote: “Such calculated concentrations are considered semi-quantitative, because they represent the amount of organic matter that is actually ionised in the system. As the PTR ionization efficiency of organics is not 100 % (there are losses in neutral molecular fragments and CO₂ – not detected by the PTR-MS) the semi-quantitative values can be considered as the minimum amount – lower threshold of the actual ion concentrations (Materić et al., 2020).” In other words, our approximation, is rather conservative from the analytical point of view. In the present manuscript, we have made that clearer in the material and method section, this reads now:

“Presented nanoplastic concentrations are semi-quantitative as not all of the plastic material is eventually converted into detectable ions. This is due to i) thermal desorption not being perfectly efficient and ii) fractions of the analyte ending up as non-analysable ions. Hence, the reported concentrations represent the lower limit of nanoplastic concentrations.”

Similar to the reviewer, we are looking forward to future research that will provide a comparison with our results.

In summary, while Thermal Desorption coupled with PTR-MS can be useful for studying volatile compounds associated with nanoplastics, it may not provide the necessary specificity for the direct identification of nanoplastics themselves. For publication in Nature, it is crucial to validate the results obtained with other analytical techniques to confirm the presence of nanoplastics.

R.2.13. *So far, TD-PTR-MS has been applied to many environmental samples resulting in numerous publications (see also R.2.1). With each, the TD-PTR-MS method was improved. Nanoplastics were quantitatively measured in low and high OM matrices such as:*

- *ice and snow from polar regions (Materic et al., Env Res, 2022; <https://doi.org/10.1016/j.envres.2022.112741>) and Alpine regions (Materic et al., Env Pollution, 2021 <https://doi.org/10.1016/j.envpol.2021.117697>),*
- *rain depositions (Allen et al., Journal Haz Mat, 2022, <https://doi.org/10.1016/j.hazadv.2022.100104>),*
- *rural and remote surface waters (Materic et al., Env Res, 2022, <https://iopscience.iop.org/article/10.1088/1748-9326/ac68f7/meta>),*
- *air samples (Kau et al., Chemosphere, 2024, <https://doi.org/10.1016/j.chemosphere.2024.141410>),*

- sea water samples (Materic et al., STOTEN, 2022, <https://doi.org/10.1016/j.scitotenv.2022.157371>), and
- marine biota muscle samples (Fraissinet et al., Nature's COMMSENV, 2024, <https://doi.org/10.1038/s43247-024-01300-2>).

All TD-PTR-MS based nanoplastic measurements quoted above were published in good/high-ranking journals. The TD-PTR-MS data for these NPs papers (raw files and all stages of data processing) were published together with the manuscript, which makes all of our findings transparent.

We furthermore kindly disagree with the reviewer that TD-PTR-MS has been developed only for volatile OM (see **R.2.1**); the community has been using TD-PTR-MS for semi-volatile compounds and now for nanoplastics too, for several years already.

If OM is a confounding factor to certain samples, case-to-case basis, we apply a digestion protocol. That was not necessary for these relatively low OM seawater samples. Please look at the comments above regarding specificity and OM matrix influence as TD-PTR-MS, if applied correctly, does provide the necessary specificity.

The response regarding the reviewer's point of the necessity of using other analytical techniques to publish in Nature is below.

For the specific analysis of nanoplastics, other techniques such as infrared spectroscopy, scanning electron microscopy (SEM), tandem mass spectrometry (MS/MS), or gas chromatography-mass spectrometry (GC-MS) may be useful to cross-validate the results, especially at these levels. These methods allow for a more precise characterization of nanoplastics in terms of chemical composition, size, shape, and other properties.

R.2.14. The reviewer is correct that nanoplastics can be analysed with electron microscopy-based methods as well as mass spectrometry-based methods. However, each of these methods comes with their own bias:

- Infrared spectroscopy is classically limited for particles >10 µm because of diffraction limitations.
- SEM is useful to 'picture' nanoparticles (but not to quantify them). Moreover, it is not possible to determine the chemical identity of the nanoparticle.
- Tandem mass spectrometry (MS/MS): to our knowledge, no paper has shown that this technique has been successfully used for quantifying NPs in environmental samples.
- Gas chromatography-mass spectrometry (GC-MS) (we assume that the reviewer refers to pyrolysis or TED-GC-MS): these methods can be used to detect and identify nanoplastics. In fact, they share several characteristics with our TD-PTR-MS method (see also **R.2.1**). However, their limit of detection is much higher than TD-PTR-MS. Py-GC-MS or TED-GC-MS are furthermore less selective for nanoplastics than TD-PTR-MS as these methods use only a few ion markers for each plastic type, making them prone to produce false-positive results.

Finally, also other studies could find nanoplastics in the ocean, including deeper water layers (Moon et al., Science Advances, 2024), see also **R.2.8**.

We thank two reviewers for their insightful comments and their thorough review. For better readability we reproduce the reviewer comments and provide our labeled answers (**Rx.xx**) in *cursive* letters.

#Reviewer 1

RR. 1.2.: The authors provided a good description of the physical properties of the gyres (here, the circulation patterns). The gyres also have distinct biogeochemical conditions (e.g. strongly and/or permanently stratified water column, low productivity, etc.) compared to the non-gyral regions, which will impact on the vertical distribution of nanoplastics. This needs to be acknowledged.

This is a good suggestion. The following sentence has been added to the paragraph ‘Vertical distribution of nanoplastics’ to acknowledge this: “Hence, in addition to varying circulation patterns and stratification, differences in productivity across ocean provinces may also influence the distribution of nanoplastics. However, the 1 μm filtration threshold excludes marine snow, preventing us from accounting for most aggregated nanoplastics.”

RR. 1.3.: It still remains confusing to understand the meaning of the ‘surface layer’ in this manuscript. In oceanographic context, surface layer could have a number of definitions (e.g. top 5-10 m, photosynthetically active layer, etc.). The authors define the vertical boundary of the surface layer to the mixed layer depth. To avoid confusion and misinterpretation, I suggest they refer to ‘the mixed layer’ instead of the ‘surface layer’. Also, provide the ‘surface’/‘mixed layer’ depth range for clarity (in Ex. Fig. 5c red numbers are difficult to read).

The reviewer is right, we thus have replaced ‘surface layer’ with ‘mixed layer’ (the commonly used term in oceanography) throughout the manuscript to improve clarity. We refer now more clearly to the two types of mixed layer depths (the one calculated from CTD data for every station, and the climatological one used for the calculation of the North Atlantic mixed layer volume) with the following adaptations:

“The ocean’s mixed layer was sampled at 10 m water depth (see Extended Data Fig. 5c for mixed layer depth ranges of the stations).”

“To estimate a mixed layer nanoplastic mass budget, we considered an average climatological mixed layer depth for November (indicated by the contours in Ext. Data. Fig. 5c)...”.

“The climatological mixed layer depth was calculated⁶⁴ using World Ocean Atlas November mean data (Extended Data Fig. 5c). The station mixed layer depths were calculated from the CTD measurements from this study (Extended Data Fig. 5c). Although the CTD occasionally measured deeper instantaneous mixed layer depths than the climatological mean, they are within expectations.”

Extended Data Fig. 5 has been improved for readability and the mixed layer depth definitions have been clarified in the caption:

Extended Data Fig. 5: **Dynamic Height Anomaly Contours (m^2s^{-2}) of Ψ_{ref} at 1000 m depth (a), the depth weighted average Ψ_{rel} over 410 m depth (b) and the November climatological and station mixed layer depths (c).** Note the different scale of the colormaps for panel a and b. Panel c shows the climatological mixed layer depth from November gridded climatology (purple colours) that was used for the North Atlantic mass budget calculations, which are in good agreement with the station mixed layer depths derived from CTD measurements (white boxes) at the stations (thick black dots). The small black dots indicate the grid points “inside the gyre”, while the red crosses indicate the grid points “outside the gyre”, both bounded by the latitude domain. The thin black contour is the coastline, and the grey contour marks the 200 m isobath.

RR. 1.4.: I agree. Yet, it is not only the case of not considering PVC or PET in the work by Kaandorp et al. 2023. Their model only considers plastics >100 microns in size and accounts neither for smallmicroplastics nor nanoplastics, which are more abundant than larger plastics – this need be emphasized further and will add more value to the data presented in this manuscript. In lines of 201-212 it is important to include the lowest size category macro/microplastics reported in studies referenced for the comparison, which will accentuate the importance of the very small nanosized category of plastic contaminants reported in this study.

We completely agree that this point could be emphasized more effectively. In the conclusive part we extended the following sentence to emphasize this more strongly:

“This implies that the total mass of plastic in the ocean is higher than previously thought, because nanoplastics were not accounted for in marine plastic budget assessments^{1,3,4}.”

We have also now expanded the discussion comparing our nanoplastic concentrations with values from the literature by specifying the size fractions considered in each study:

“At the same stations as measured here, the macro/microplastic (>500 μm) mass (consisting primarily of PE and PP) was found to amount to $\sim 0.11 mg m^{-3}$ at the sea surface, and to $< 0.02 mg m^{-3}$ (consisting primarily of PET) at depth >5 m in the mixed layer². Higher microplastic (32–651 μm) mass concentrations of $\sim 1.25 mg m^{-3}$ (consisting primarily of PP and PE) at the sea surface and $0.62 mg m^{-3}$ (consisting primarily of PE, PP and PS) at depth

>10 m were found at two other stations in the mixed layer of the NASG¹. Furthermore, also recently modelled concentrations of up to 3.4 mg m⁻³ of buoyant macro/microplastics (0.1–1,600.0 mm, primarily PE, PP and PS) at the sea surface of the NASG³ are lower than our measured nanoplastic concentrations.”

RR.1.5.: Similar to my comment above, reference to the lower size boundary of the previously reported polymer specific plastics will strengthen the discussion. In addition, reference to Lebreton et al. (2017) and Pabortsava and Lampitt (2020) who advocated the need to address plastic contamination issue on polymer-specific basis will strengthen the discussion and the value of the dataset presented. No method to date can measure all types of plastics, so the concept of bulk concentration is intrinsically/methodologically biased.

This comment has partially been addressed in RR1.4. Moreover, we have now added the following sentence to the discussion:

“As bulk plastic concentration measurements are inherently prone to methodological bias^{1,12}, the following provides a polymer-specific budget assessment.”

Similar to Pabortsava and Lampitt (2020), this work shows that the limited number of polymer-specific nanoplastics exceed the mass of bulk macro/microplastics and highlights the critical importance of addressing the smallest category of these contaminants.

Pabortsava and Lampitt (2020) indeed found that the smaller microplastic fraction was more important for the bulk plastic budget, however they did not measure plastics in the nano size fraction. However, to acknowledge that Pabortsava and Lampitt (2020) mentioned the importance of addressing small microplastic and nanoplastic, we have added their reference to:

“A skewed ocean plastic size distribution towards smaller particle diameters^{30,31}, however, suggests that nanoplastics could be a globally important contaminant¹.”

‘Macro/microplastics’ can be referred to as ‘larger plastics.

We agree that the reviewer’s suggestion sounds better, but to avoid confusion and for the sake of consistency throughout the manuscript and in alignment with the literature, we would prefer to continue using the terms macro- and microplastics (for the size fractions >1µm – 5mm and >5mm) and nanoplastic (<1µm).

Discussion in lines 228-238 can be strengthened and include the implications of the most of plastic contamination being in small/nano-size category. This can be linked to behavior and properties of nano-sized contaminants (e.g. ability to pass through biological barriers (cell walls, tissues), translocation, bioaccumulation and speed of chemical interactions (leaching, sorption). We know very little of quantities, properties and behavior of nanoplastics, but they can be the size-fraction that causes the most damage short and long-term. See Gigault et al. 2021 for reference.

The discussion was strengthened by the addition of the sentence highlighted in green:

“The first observation of ocean nanoplastics was published in 2017 (ref. 25). Concerns about nanoplastics have grown strongly ever since, as nanoplastics have adverse effects on marine life^{6,23}. Due to their ability to traverse biological barriers⁶⁶, translocate⁶⁵, bioaccumulate²², and interact chemically at rapid rates⁶⁷, nanoplastics may represent the most problematic plastic size fraction for ocean life.”

Line 230: the authors should emphasize that all studies on harms from nanoplastics were all based on concentrations not supported by real measurements in the environment (they are very scarce and mostly qualitative). Having such measurements (as in this study) is thus critical to reliably assess and understand the real risk from these contaminants now and in the future.

The following sentence was added to emphasize this:

“Due to their ability to traverse biological barriers⁶⁶, translocate⁶⁵, bioaccumulate²², and interact chemically at rapid rates⁶⁷, nanoplastics may represent the most problematic plastic size fraction for ocean life. Notably, most studies assessing the impacts and toxicity of nanoplastics use baseline nanoplastic concentrations unsupported by robust environmental measurements.”

In the final discussion paragraph it was now also emphasized that we provide quantitative nanoplastic measurements:

“While mechanisms contributing to the creation of secondary nanoplastics from parent ocean macro/micropastics were shown^{20,21,38,39}, only two studies were able to detect these compounds in the ocean water column^{26,27}. This study provides the first quantitative evidence of the ubiquitous presence of PET, PVC and PS nanoplastics from the mixed layer to deep sea bottom waters across the temperate to subtropical North Atlantic.”

RR.1.6.: Thank you for clarifying this. In lines 270-271, what does ‘dedicated to nanoplastic research’ mean? ‘Cleaning on weakly basis’ does not really resolve the issue of potential airborne contamination. In their methods section, the authors should recommend future work to be carried out in the certified clean laboratory, which will further minimize plastic contamination of blanks.

We added the recommendation to conduct future work in clean labs and have further clarified upon the lab conditions:

“The water samples were processed in the PTR-MS lab at IMAU in Utrecht. During the time of analysis, this was thoroughly cleaned and de-dusted on a weekly basis. Typically, only one person was present in the lab during analysis to minimize potential contamination. Blanks were included with every sample batch to account for the risk of airborne contamination. For future work, processing samples in a clean room should be considered, although the effectiveness of clean labs in eliminating plastic contamination at the nanoscale is at present uncertain.”

RR.1.7. It is interesting that the only other work published by the lead author on nanoplastics in seawater (the Dutch Wadden Sea, Materic et al. 2022) also reports PS and PET but no PE, PP, and other polymers, unlike other works by the same author on

samples of ice and snow, freshwater, and biota. In agreement with Reviewer 2, could the seawater matrix with CDOM and nano/picoplankton (e.g. Synechococcus, E. hax) impact the detection, by masking PE and PP, and/or by contributing ethylbenzene and benzene groups (i.e. constituents of polystyrene; see Rocco et al. 2021, <https://doi.org/10.1038/s43247-021-00253-0>)? As I mentioned in the first round of reviews, I am not an expert in PTR-MS, but with a limited application of this technique on marine samples with no natural particulates removed/reduced (even at <1 micron size) makes me think that the method favours detection of PS and PET.

As with all analytical methods, some analytes are easier to detect than others. For nanoplastics (and microplastics), PET and PS indeed have more unique features in their mass spectra (compared to PE and PP), which facilitates their identification and results in a lower limit of detection (LoD). This applies not only to TD-PTR-MS, but to all methods used in micro- and nanoplastics analysis, including Py-GC-MS, Raman, and FTIR.

In our case, we observe only some of the PE-associated ions in our mass spectra, but this is not sufficient for strict analytical confirmation. Many of the ion markers typically present in the mass spectra of PE are entirely absent in our sample (see the figure below, red arrows point to such ions). Therefore, we cannot unambiguously conclude that background organic matter is masking our PE signal and/or that the absence of the diagnostic ions for a positive match indicate that the original PE matrix has been altered (e.g., through photooxidation) – we have added some discussion to the main text addressing this. In any case, we must conclude that PE (at least in its pristine form) is below our detection limit. Just as a side note, we did indeed detect these ions in our previous works on surface waters, snow and biota, as reviewer is well informed – our method as such is hence suitable to measure PE nanoplastics (at least in the pristine configuration of PE). The following figure has been added to the Extended Data:

*Extended Data Fig. 9: **The absence of certain ions typically associated with PE in the seawater samples.** Many ion markers typically observed in the mass spectra of PE are completely absent from our samples (indicated by red arrows). As a result, we cannot definitively determine whether background organic matter is obscuring the PE signal or whether the absence of diagnostic ions indicates that the original PE matrix has been altered (e.g., through photooxidation). Regardless, we must conclude that pristine PE, if present, remains below our detection limit in seawater samples.*

The chemical composition of organic matter will also differ between that of biota (zoo-) and phytoplankton – was that a consideration in Monte Carlo assessment? Did the authors know the organic matter load and content of their samples to carry out such assessment, i.e. which compounds were considered?

Here, to preserve strictness and cover for the broad range of possible OM sources, we used an unsupervised Monte Carlo simulation, and increased the load in each iteration, spreading

over 5, 10 and 40 ions (with exact m/z and volatility range as plastics in questions). Note that we wrote:

“We simulated the addition of organic matter to the mass spectra of our plastic library and assessed identification and quantification performance. We systematically added 50-350% (increment of 50%) of signal randomly spread over up to 5, 10, and 40 ions of our library used for the identification of nanoplastics. Each sequence of the run was done in 1000 replicas.”

Because of the random spread of the signal, different possible OM sources are accounted for. We also performed an additional experiment to rule out a possible false positive from OM. We wrote in the QC section the following:

“To assess potential false positives from organic matter, we analysed Sargassum tissue samples, using Sargassum as a proxy for complex organic material that may be present in seawater. Approximately 0.5 mm³ of Sargassum tissue – harvested during our previous campaign and stored frozen – was dried in an oven at 50°C for two hours before TD-PTR-MS analysis. The Sargassum tissue samples (no digestion applied) showed no positive matches for PE, PP, PET, PVC, or TWP and only a negligible match for PS, characterized by a low final PS quantity and a low algorithm matching score (see Extended Data Tab. 1). To maintain a conservative approach, we considered this PS match as a potential false positive in our water samples and accordingly increased the PS matching threshold to eliminate such false positives across all samples.”

Line 186 – replace ‘configuration’ to ‘chemical composition’.

Configuration has been replaced with composition, resulting in “(iii) the chemical composition...”

Line 294: this requires a reference

References were added and the sentence was modified as follows: “Hence, much of the organic matter matrix was excluded from analysis, as many monomers and most volatile compounds typically desorb at temperatures below 200°C^{26,32,56}.”

Lines 296-298: nanoplastics can also (potentially) aggregate with OM larger than 1 micron in size, so this study probably lost nanoplastics associated with particles >1 micron (e.g. marine snow, faecal pellets). This needs to be included into the discussion. Also see Gigault et al 2021 who proposes that the fate/distribution of nanoplastics might be driven by the association/aggregation with organic matter, e.g. downward transport on nanoplastic to depth).

The reviewer is absolutely right and the exclusion of >1 µm NP aggregates is now more clearly acknowledged in the Method and Discussion sections:

“Consequently, our method measures collectively free nanoplastics and nanoplastics that are loosely associated to OM or that are aggregated, provided that the aggregates pass filter pores ($\leq 1 \mu\text{m}$) during prefiltration.”

“However, sinking particles and aggregates (e.g. marine snow) can cross the pycnocline⁴⁴. Hence, in addition to varying circulation patterns and stratification, differences in productivity across ocean provinces may also influence the distribution of nanoplastics. However, the 1 µm filtration threshold excludes marine snow, preventing us from accounting for most aggregated nanoplastics.”

“Nanoplastic production from sinking micro and macro-particles is hence the least parsimonious explanation for the presence of nanoplastics in bottom waters, in addition to sinking of nanoplastic aggregates.”

RR1.8: Lines 331-332: what was the dominant z-score for detected PS and PET in the actual samples?

Since spike experiments were conducted exclusively for PS and all z-scores will be included in the supplementary data (allowing readers to review z-scores for PET/PVC if desired), the sentence has been revised as follows:

“Fingerprinting these spiked samples consistently yielded positive matches for PS with z-scores of 4 or higher. In contrast, only 29.4% of the unspiked mixed layer samples with PS showed z-scores of 4 or above.”

RR. 1.10. Addressed. In legend of Extended data figure 4 report the units as in the figure (mg m⁻³).

All the units in the caption are now consistent with the figure:

*Extended Data Fig. 4: **Plastic contamination detected in the lab blanks.** All lab blank batches were found to have consistently low average PE, PET, PPC, PP, PS and PE nanoplastic concentrations <3 mg m⁻³. After background subtraction, composed of the mean of the lab blanks of the corresponding batch, still considerable amounts of nanoplastic could be detected in the ocean water samples. We acknowledge nonetheless that the presence of background nanoplastic, although in low amounts, results in additional uncertainty of nanoplastic concentrations. Negligible amounts of PET were detected in the lab blanks performed during the measurements of the bottom water samples (see “batch 3”), implying that the considerable amounts of PET nanoplastic detected at kilometers depth are not a result of procedural contamination. However, up to 4 mg m⁻³ of PS has been observed in some of the lab blanks.*

RR. 1.13. Nepheloid layer created by the resuspension/transport of the seabed sediment by the bottom currents can reach up to 1000 m above the seabed across the Atlantic. This needs to be acknowledged as a potential source of nanoplastics at 30 m depth above the seabed.

The following sentence was added to the discussion of nanoplastics in bottom waters:

“Accumulation of nanoplastics in a nepheloid layer – which, in some areas in the North Atlantic, can extend up to 800 m above the seabed⁶⁸ – as well as resuspension of sediments and the remobilisation of potentially deposited nanoplastics, may further contribute to elevated nanoplastic concentrations in bottom waters.”

*Gardner, W. D., Richardson, M. J., Mishonov, A. V., & Biscaye, P. E. (2018). Global comparison of benthic nepheloid layers based on 52 years of nephelometer and transmissometer measurements. *Progress in Oceanography*, 168, 100-111.*

RR.1.14. add a grey bar in the legend to indicate the coastal/shelf break stations

This has been done as suggested, the new Fig. 1 now provides this information:

RR. 1.19 and 1.20. I still believe that the authors can make both the abstract and the summary paragraph stronger. Nanoplastics are severely understudied (only a few studies exist to date and most of them are qualitative) in the marine environment and this study is the first one to report mass concentrations and estimated budgets of polymer-specific nanoplastics on a basin-wide scale and with depth. The last sentence of the abstract needs to be rephrased with more implications added – marine plastic contamination is now shown to be predominantly in the smallest, nano-size category. This very small size of plastic contaminants may drive completely different physical and chemical interactions and pathways of these contaminants in the marine environment and their impacts therein.

We agree with the reviewer and have now added context in this regard making the summary paragraph and abstract stronger:

“The first observation of ocean nanoplastics was published in 2017 (ref. 25). Concerns about nanoplastics have grown strongly ever since, as nanoplastics have adverse effects on marine life^{6,23}. Due to their ability to traverse biological barriers⁶⁶, translocate⁶⁵, bioaccumulate²², and interact chemically at rapid rates⁶⁷, nanoplastics may represent the most problematic plastic size fraction for ocean life. Notably, most studies assessing the impacts and toxicity of nanoplastics use baseline nanoplastic concentrations unsupported by robust environmental measurements. While mechanisms contributing to the creation of secondary nanoplastics from parent ocean macro/micropastics were shown^{20,21,38,39}, only two studies were able to detect these compounds in the ocean water column^{26,27}. This study provides the first quantitative evidence of the ubiquitous presence of PET, PVC and PS nanoplastics from the mixed layer to deep sea bottom waters across the temperate to subtropical North Atlantic. Spatially extrapolated, our measurements strongly suggest that nanoplastics are the largest fraction of the marine plastic mass budget. This implies that the total mass of plastic in the ocean is higher than previously thought, because nanoplastics were not accounted for in marine plastic budget

assessments^{1,3,4}. Our finding underscores the need to determine the origin, formation and transport of nanoplastics, as well as their further fate in the ocean.”

The abstract was improved as follows:

“understudied” was substituted with “remain unquantified” resulting in:

“Plastic pollution of the marine realm is widespread, with most scientific attention given to macro and microplastics. Ocean nanoplastics (<1 μm), in contrast, remain unquantified, leaving the mass budget of this plastic size class unknown.”

The final two sentences of the abstract now read as follows:

“Our findings suggest that nanoplastics comprise the dominant fraction of marine plastic pollution, which has far-reaching implications for plastic-related impacts to the marine environment and the fate of plastic waste in the ocean.”

In lines 228-230 it should be highlighted that very limited data on the magnitude and characteristics of marine nanoplastic contamination exists to support the laboratory-based studies investigating the effects on nanoplastics on the health of biota/ecosystem functioning.

We hope this is now sufficiently highlighted with the previous modification:

“Due to their ability to traverse biological barriers⁶⁶, translocate⁶⁵, bioaccumulate²², and interact chemically at rapid rates⁶⁷, nanoplastics may represent the most problematic plastic size fraction for ocean life. Notably, most studies assessing the impacts and toxicity of nanoplastics use baseline nanoplastic concentrations unsupported by robust environmental measurements.”

Referee #3 (Remarks to the Author):

Dear Authors,

this is important work and pushing the boundaries of our abilities to measure plastics to smaller sizes. Being able to robustly quantify nanoplastics is critical to our understanding of the transport, fate, and impact of plastics in the natural environment and also for health science.

Previous reviews have provided excellent summaries and other suggestions that you, the authors, have responded to. The manuscript reads well and all conclusions are robust if the analytical method is measuring nanoplastics. 1 reviewer suggested rejection based upon uncertainty about the analytical method. I share their reservations based on data attained via Py-GC/MS in the North Atlantic Subtropical Gyre.

The data I am describing is unpublished as we are trying to verify the presence of plastic vs other organics via other methods. As the authors describe, we also see PVC and aromatic signatures (PS and PET) in the Py-GC/MS data. The region sampled is the NASG which is also known as the Sargasso Sea due to the prevalence of the

Sargassum (floating seaweed). Sargassum is a source of organic matter and found in higher abundance in surface waters of the NASG than plastics. The dissolved organic matter produced from Sargassum is rich in polyphenolics, including halogenated phlorotannins (e.g., Powers et al., 2019; <https://doi.org/10.1029/2019GB006225>). DOM samples in Powers et al. were filtered through 0.2 μm filters, so could include polyphenolic-rich nanoparticles from Sargassum. Sargassum-derived particles in the 0.2 to 1 micron range would also presumably contained polyphenolics.

Based on the above, we generated Sargassum Py-GC/MS data, including Sargassum + plastic standards to look for Sargassum-derived pyrolytic products and co-products that could be mistaken for plastics. We are still grappling with the results. I mention this as Sargassum-derived polyphenols and halogenated phlorotannins could potentially produce TD-PTR-MS signatures that overlap with those of other aromatic and halogenated polymers, such as PS, PET and PVC. Thus, I recommend the authors acquire a sample of Sargassum biomass and analyze it to confirm it does not. If not, then the rest of the data and conclusions seem robust.

Without this piece of information, the conclusions seem unsupported and I would recommend rejection if the authors do not commit to test Sargassum signatures.

We very much appreciate the reviewer's suggestion for the interference test. Previously we have done this type of test for humic acids and cellulose and published it within our surface water paper DOI 10.1088/1748-9326/ac68f7. We were happy to carry out the Sargassum experiment as suggested. Sargassum tissue (undigested) was analysed using TD-PTR-MS. With the same parametrisation as used for the nanoplastic measurements carried out in this paper. We could not find a positive match for PVC or PET (or PP, PE and TWP). Mass spectra of those plastics massively differ from ions that are produced by the Sargassum organic matter. The figures below highlight many ions/ion clusters that are present in plastics mass spectra, but not in Sargassum. This gives a sufficient analytical confidence that there is no likely false positive scenario for these plastics using our strict 40-ion fingerprint method (note the arrows pointing to the key features).

However, we found a, though very low, positive match for PS in undigested Sargassum tissues yielding an algorithm matching z-score of 2 or 3. Despite the positive fingerprint, the overall amounts of PS were very low compared to the high amount of Sargassum tissue used in the experiment (note that it is virtually impossible for a sargassum piece $>1\mu\text{m}$ to pass through the filter membrane – for the experiment we used a piece of 0.5 mm^3 and exposed this directly to thermal desorption). However, to remain conservative, we considered a z-score of 3 or lower for PS in Sargassum tissue as a potential false positive for our water samples.

To ensure the complete elimination of possible false positives caused by potential algal interference, we thus raised the 40-ion identification threshold for PS, from a z-score of >2 to a z-score of >4 .

To the Methods the following sentence has been added together with more information in Extended Data:

“To assess potential false positives from organic matter, we analysed Sargassum tissue samples, using Sargassum as a proxy for complex organic material that may be present in seawater. Approximately 0.5 mm³ of Sargassum tissue – harvested during our previous campaign and stored frozen – was dried in an oven at 50°C for two hours before TD-PTR-MS analysis. The Sargassum tissue samples (no digestion applied) showed no positive matches for PE, PP, PET, PVC, or TWP and only a negligible match for PS, characterized by a low final PS quantity and a low algorithm matching score (see Extended Data Tab. 1). To maintain a conservative approach, we considered this PS match as a potential false positive in our water samples and accordingly increased the PS matching threshold to eliminate such false positives across all samples.”

“The fingerprint algorithm compares the spectra against a library comprising the seven most prevalent polymers: PE, PET, PS, PP, PPC, PVC, and tire wear. A matching score of 2σ (z -score = 2, $p < 0.02275$, one tail distribution) was considered a positive fingerprint. Algal organic matter may slightly increase false positive PS detection (see QC section and Sargassum experiment in Extended Data Tab. 1). To minimise this risk of false positive annotations, we only considered a z -score of 4 or higher as a positive fingerprint match for PS. Matching scores are indicated with * (z -score > 2), ** (z -score > 3) and *** (z -score > 4), where a higher matching score indicates a better fit with the library mass spectra.”

“

Extended Data Tab. 1: **Results of the Sargassum fingerprinting experiment.** Only for PS, Sargassum induced a false positive fingerprint exclusively with Algorithm (ALG) 3. For all other polymers, no positive fingerprint could be generated using Sargassum tissue alone.

Sample	ALG1	ALG3	Mass
PS_Spike_1-4	***	***	93.99 ng
PS_Spike_2-5	**	***	28.01 ng
PS_Spike_3-6	**	***	89.83 ng
Sargassum Tissue_1-7	-	**	2.41 ng
Sargassum Tissue_2-11	-	*	8.84 ng
Sargassum Tissue_3-15	-	*	2.82 ng

“

All results and figures have been updated, with PS concentrations in samples that did not meet the new PS positive-fingerprint threshold requirements set to 0 mg/m³. We believe this approach is sufficiently conservative.

We would like to highlight that our TD-PTR-MS method utilizes 40 different ions, with the distribution and ratios among these ions being used to quantify nanoplastics based on the library spectra. This significantly reduces the likelihood of interference from organic matter, as it is improbable for such interference to consistently affect all 40 ions in the same way. In this regard, TD-PTR-MS distinguishes itself from methods like pyGCMS, which rely on far fewer ions. Additionally, interference from Sargassum does not align with the observed patterns of nanoplastic concentrations in our study. Specifically, we observe a gradient of

high nanoplastic concentrations near the European coast that decreases toward the North Atlantic gyre region, where Sargassum is most abundant. This makes it highly unlikely that Sargassum interference significantly impacts our results.

I am still of an opinion that the manuscript by Materic et al. presents the work of high scientific importance and is worthy of publication in a Nature group journal, provided that the methodology is solid and a second round of my reviews is addressed. The former requires an expert opinion on PTR-MS, especially for detecting plastics, as I have a limited understanding of this technique. My current concern with this technique is i) low recovery rate for PS, ii) unexplained inability to detect PE in marine samples (also in Materic et al. 2022), iii) potential interference of organic matter (e.g. CDOM, phytoplankton), also raised by Reviewer 2. My new set of comments is a point by point reply to the authors labelled as **RRx.xx** for clarity.

Point to point reply for manuscript **2023-09-17026** entitled
'*High nanoplastic concentrations across the North Atlantic*'

We thank two reviewers for their insightful comments and their thorough review. For better readability we reproduce the reviewer comments and provide our labeled answers (**Rx.xx**) in *cursive* letters.

Referee #1 (Remarks to the Author):

Summary of the key results:

Materic et al. quantify and identify chemical composition of nanoplastic contaminants in the marine samples collected at 2-3 depths (surface, 1000 m and 5-30 m above the seabed) on a transect crossing the North Atlantic (12 stations in total) from the European shelf to the centre of the North Atlantic gyre. The authors report presence of nanoplastics (<1 µm) at all sampled locations and depths, composed of mainly PS, PVC and PET. They measured elevated concentrations of these polymers in the surface water of the European shelf and inside the gyre. The authors report an overall decreasing vertical gradient of the combined load of these three polymers, with significantly smaller abundance observed close to the seabed. PET dominated the nanoplastics load at all the deep stations. Assuming average concentrations measured in the surface layer, the authors estimated that 33-50 Mt of PET+PVC+PS nanoplastics are present in the top 80 of the North Atlantic basin. Their estimates significantly exceed the load of larger plastic contaminants reported in the ocean. The authors conclude that nanoplastics are the largest contributor to the marine plastic contamination load.

The presence and fate of plastic litter in the marine environment is of major concern with high societal relevance. The measurements of very small, nano-sized plastics (nanoplastics) in the ocean are extremely difficult and thus rare. To date, the studies of nanoplastics in the ocean were focused on measurements in the surface water (e.g., Ter Halle et 2017). The work by Materic et al. is the first assessment of nanoplastic contamination on a basin-wide transect covering different geographic and oceanographic regions, and at distinct depth horizons. This study is extremely timely and sheds new light on the extent and fate of plastic contaminants in the marine environment all the way to nano-scale. Although limited to three polymer types and 12 stations, the observations are very important and useful for constraining further the marine plastic budgets and improving the models of dispersion and fate of plastic contaminants in the environment.

Overall, this study might rock field of plastics research, since it puts the attention to the importance of very small (nano-sized) plastic contaminants, which are not captured common with methods as

manta trawls, and which are currently recommended to use for monitoring of surface sea water (e.g. GESAMP (2019) Guidelines or the monitoring and assessment of plastic litter and microplastics in the ocean) and are still used to predict the global mass of marine plastics (e.g. Kaandorp et al. 2023). The authors also show that nanoplastics can be of a major importance in the overall context of plastic weight and fate in the ocean compared to small microplastics (e.g., Pabortsava and Lampitt 2020) and larger microplastics captured with nets (> 200-300 microns in size; Kaandorp et al. 2023). These find can have significant implications plastic contaminants in smaller size range (<1 micron) are predicted to be the most pervasive and harmful because of their ability to pass through biological cells and tissues.

The dataset that the authors produce is first-of-its-kind and in that sense very valuable. Especially the depth-resolved stations are unique. However, I have several major points for consideration which preclude me from recommending this manuscript for publication in current form. This study requires significant revisions before being accepted for publication in Nature or Nature family journals.

R.1.1. *We greatly appreciate the reviewer's time and effort in suggesting improvements to our manuscript, as well as their positive assessment of the novelty and impact of our work.*

RR. 1.1. Thank you.

The authors report three polymer types sampled at 12 stations, they should discuss further how representative their samples/observations are for the chosen region and also discuss them in broader context (including ocean transport and physical properties of the water column which will be different in the gyral, non-gyral and shelf regions).

R.1.2. *Water In the North Atlantic subtropical gyre circles several times around the center before it moves downward and/or northward (Berglund et al 2022). There is a convergence of water towards the middle and subsequent downwelling. Hence, water parcels "trapped" in the gyre will temporarily be subject to different dynamical pathways than water outside of the gyre. Because of this circulatory movement of the upper water mass and our transect crossing these flowlines, our 'gyre' stations are representative for the 'gyre' region. We have further elaborated this in the revised manuscript (see Methods: "Calculation of the North Atlantic surface layer volume"). In the original manuscript, the extrapolation of the 'outside gyre' region was comprising the remainder of the North Atlantic. Upon reviewing oceanographic circulation patterns, we agree that this was not fully justified because of the circulation pattern in the subarctic/arctic North Atlantic featuring its own subpolar gyre. In the revised version of the manuscript, we have now confined the northern limit for extrapolation to 55 °N, as this separates the sub-polar area from the temperate to subtropical region in which we sampled. We did not perform an extrapolation for the 'coastal' region, as nanoplastic concentrations are likely much more heterogeneous there due to the vicinity of contamination (point) sources. Considering the differences in oceanographical properties as well as coastal dynamics, we are confident that the three categories 'gyre', 'outside gyre' and 'coastal' most representative to be used in our statistical analysis.*

RR. 1.2.: *The authors provided a good description of the physical properties of the gyres (here, the circulation patterns). The gyres also have distinct biogeochemical conditions (e.g. strongly and/or*

permanently stratified water column, low productivity, etc.) compared to the non-gyral regions, which will impact on the vertical distribution of nanoplastics. This needs to be acknowledged.

The extrapolations of the mass-budgets of the three polymer types should be clarified and justified. It is currently unclear, for example, why the authors picked 80 m as a depth interval for the mass budget calculations.

R.1.3. *The motivation and explanation for the extrapolation volume has been improved by the addition of section “Calculation of the North Atlantic surface layer volume” in the new method section. Previously, we based the surface layer depth on our CTD measurements. Now, we improved our extrapolation by including multi-year Argo Float data to better constrain gyre boundaries. Instead of only using our own CTD measurements, we have now calculated the surface layer depth based on the World Ocean Atlas November mean data. Nevertheless, our CTD measurements agree generally well with the multi-year Ocean Atlas data set, so we are confident that we sampled a representative surface layer. This provides a good estimate for the nanoplastic mass budget calculation in the subtropical to temperate North Atlantic Ocean.*

RR. 1.3.: *It still remains confusing to understand the meaning of the ‘surface layer’ in this manuscript. In oceanographic context, surface layer could have a number of definitions (e.g. top 5-10 m, photosynthetically active layer, etc.). The authors define the vertical boundary of the surface layer to the mixed layer depth. To avoid confusion and misinterpretation, I suggest they refer to ‘the mixed layer’ instead of the ‘surface layer’. Also, provide the ‘surface’/‘mixed layer’ depth range for clarity (in Ex. Fig. 5c red numbers are difficult to read).*

The findings of this study should also be evaluated in the context of plastic sources and current understanding of fate of plastics pollutants in general. I am surprised not to see a reference to the conclusions of Kaandorp et al. 2023 which are so contrasting to the results of this study.

R.1.4. *The authors fully agree that the paper of Kaandorp et al. 2023 should be discussed. Our nanoplastic mass budget findings are now explicitly and in a quantitative matter discussed and compared to the buoyant microplastic concentrations reported in Kaandorp et al., 2023 (see **R.1.5.** last paragraph of “A nanoplastic mass budget for the surface layer of the North Atlantic”). The Kaandorp data was downloaded and overlaid with the same regions (‘gyre’ and ‘outside gyre’) as used in our nanoplastic mass budget calculations. In this manner, we were able to compare the buoyant microplastic budget for the North Atlantic with that of nanoplastic, which supports our claim that the nanoplastic budget likely exceeds that of microplastic. Additionally, we would like to emphasize that the paper of Kaandorp et al. disregards nanoplastics and fitted their model to buoyant microplastic observations, not taking into account for example PVC and PET. The conclusion in Kaandorp et al. that their modeling approach provides a well-constrained ocean plastic budget may thus be valid for microplastic, but not for nanoplastics. Nevertheless, Kaandorp et al. suggested that about ~15% of the annual input of plastic to the ocean is “lost” due to fragmentation (see Fig. 2 in Kaandorp et al.). Fragmentation is basically the grinding of larger plastic pieces to fine microplastics and ultimately nanoplastics. The formation of nanoplastics due to fragmentation has been shown in several studies,*

with pioneering works by for example in Lambert & Wagner (Chemosphere, 2016), which we have cited already in the original manuscript.

RR. 1.4.: I agree. Yet, it is not only the case of not considering PVC or PET in the work by Kaandorp et al. 2023. Their model only considers plastics >100 microns in size and accounts neither for small microplastics nor nanoplastics, which are more abundant than larger plastics – this need be emphasized further and will add more value to the data presented in this manuscript. In lines of 201-212 it is important to include the lowest size category macro/microplastics reported in studies referenced for the comparison, which will accentuate the importance of the very small nanosized category of plastic contaminants reported in this study.

The authors compare their observations with the studies reporting bulk concentrations of plastics. They need to refine their discussion towards polymer-specific data comparison with earlier studies to strengthen their evidence/conclusion. This will also provide a better context for the ‘missing PE and PP nanoplastics’ reported in this study.

R.1.5. We appreciate this valuable comment. The authors fully agree that it is more sensible to include a polymer-specific discussion, especially since we could not find PE and PP nanoplastics. The new discussion was updated as follows:

“In the surface layer within the ‘gyre’ (stations 1-5), we measured average nanoplastic concentrations of 16.3 mg m^{-3} ($6.67 \pm 1.12 \text{ mg m}^{-3}$ PET, $5.32 \pm 1.21 \text{ mg m}^{-3}$ PS, $4.32 \pm 1.27 \text{ mg m}^{-3}$ PVC). These contrast previous reports of directly measured macro/microplastic concentrations. At the same stations as measured here, the microplastic mass (consisting primarily of PE and PP) was found to amount to $\sim 0.11 \text{ mg m}^{-3}$ at the sea surface and to $< 0.02 \text{ mg m}^{-3}$ (consisting primarily of PET) at depth $> 5 \text{ m}$ in the surface layer². Higher macro/microplastic mass concentrations of $\sim 1.25 \text{ mg m}^{-3}$ (consisting primarily of PP and PE) at the sea surface and 0.62 mg m^{-3} (consisting primarily of PE, PP and PS) at depth $> 10 \text{ m}$ in the surface layer were found at two other stations in the NASG¹. Also recently modelled concentrations of up to 3.4 mg m^{-3} buoyant (primarily PE, PP and PS) macro/microplastics at the sea surface of the NASG³ are lower than our measured nanoplastic concentrations.

To estimate a surface layer nanoplastic mass budget, we considered an average surface layer depth for November and the region from the temperate to subtropical North Atlantic. This is bounded by the subpolar gyre north of 55°N and by the southern extend of the NASG at 8.5°N (Extended Data Fig. 5 and 7). The volume of the surface layer in the NASG was determined to be $10.1 \times 10^{14} \text{ m}^3$, and $7.01 \times 10^{14} \text{ m}^3$ for the remaining temperate to subtropical North Atlantic (Extended Data Fig. 5c). With respect to our measurements in the surface layer in the ‘gyre’ (stations 1-5), the total nanoplastic mass amounts to 16.47 million tonnes (Mt) ($6.74 \pm 1.13 \text{ Mt}$ PET, $5.37 \pm 1.22 \text{ Mt}$ PS, $4.36 \pm 1.28 \text{ Mt}$ PVC). For the surface layer of the ‘outside gyre’ region (stations 6-9), our extrapolation yielded a total nanoplastic mass of 13.22 Mt ($5.21 \pm 0.84 \text{ Mt}$ PET, $4.00 \pm 0.91 \text{ Mt}$ PS, $4.01 \pm 0.96 \text{ Mt}$ PVC). This is substantially higher than the recently modelled macro/microplastic mass of buoyant plastic in the surface layer amounting to 0.31 Mt in the ‘gyre’ and to 0.05 Mt in the remaining temperate to subtropical North Atlantic³.

The first observation of ocean nanoplastics was published in 2017 (ref. 25). Concerns about nanoplastics have grown strongly ever since, as nanoplastics have adverse effects on marine life^{6,22,23}. While mechanisms contributing to the creation of nanoplastics from parent ocean macro/microplastics were shown^{20,21,38,39}, only two further studies were able to detect these compounds in the ocean water column^{26,27}. In this study, we provide evidence for the ubiquitous presence of nanoplastics from the surface layer to deep sea bottom waters across the temperate to subtropical North Atlantic. Spatially extrapolated, our measurements strongly suggest that nanoplastics are the largest fraction of the marine plastic mass budget. This implies that the total mass of plastic in the ocean is higher than previously thought^{1,3,4}. Our finding underscores the need to determine the origin, formation and transport of nanoplastics, as well as their further fate in the ocean.”

RR.1.5.: Similar to my comment above, reference to the lower size boundary of the previously reported polymer specific plastics will strengthen the discussion. In addition, reference to Lebreton et al. (2017) and Pabortsava and Lampitt (2020) who advocated the need to address plastic contamination issue on polymer-specific basis will strengthen the discussion and the value of the dataset presented. No method to date can measure all types of plastics, so the concept of bulk concentration is intrinsically/methodologically biased. Similar to Pabortsava and Lampitt (2020), this work shows that the limited number of polymer-specific nanoplastics exceed the mass of bulk macro/microplastics and highlights the critical importance of addressing the smallest category of these contaminants.

‘Macro/microplastics’ can be referred to as ‘larger plastics.’

Discussion in lines 228-238 can be strengthened and include the implications of the most of plastic contamination being in small/nano-size category. This can be linked to behavior and properties of nano-sized contaminants (e.g. ability to pass through biological barriers (cell walls, tissues), translocation, bioaccumulation and speed of chemical interactions (leaching, sorption). We know very little of quantities, properties and behavior of nanoplastics, but they can be the size-fraction that causes the most damage short and long-term. See Gigault et al. 2021 for reference.

Line 230: the authors should emphasize that all studies on harms from nanoplastics were all based on concentrations not supported by real measurements in the environment (they are very scarce and mostly qualitative). Having such measurements (as in this study) is thus critical to reliably assess and understand the real risk from these contaminants now and in the future.

The overall description of sample collection, handling and analysis seem to be appropriate, thus other scientists in the field should be able to replicate this type of nanoplastics measurements. The authors should elaborate on the laboratory conditions where the sample processing was performed, e.g. were the samples handled in the clean laboratory?

R.1.6. At the start of section “TD-PTR-MS analysis” we added the sentence: “The water samples were processed in the PTR-MS lab at IMAU in Utrecht that during the time of analysis was dedicated to nanoplastic research and cleaned on a weekly basis.” to clarify this. The laboratory is not a certified clean laboratory, but we did run multiple sampling and process blanks to account for potential contamination during all steps of sample handling and measurement.

RR.1.6.: Thank you for clarifying this. In lines 270-271, what does ‘dedicated to nanoplastic research’ mean? ‘Cleaning on weakly basis’ does not really resolve the issue of potential airborne contamination. In their methods section, the authors should recommend future work to be carried out in the certified clean laboratory, which will further minimize plastic contamination of blanks.

I am however not an expert in Thermal Desorption – Proton Transfer Reaction – Mass Spectrometry technique, and specifically in the fingerprinting algorithm used to differentiate synthetic polymer types from natural material. This is interesting given that the authors refer to a potential masking of PE by organic background (line 183).

R.1.7. *Matrix effects masking target signals is a common phenomenon in mass spectrometry and could, at least partly explain the lack of observed PE nanoplastics. Please note that for freshwater samples (<https://iopscience.iop.org/article/10.1088/1748-9326/ac68f7/meta>), air samples (<https://doi.org/10.1016/j.chemosphere.2024.141410>) and even marine biota samples (<https://doi.org/10.1038/s43247-024-01300-2>) we could clearly observe PE nanoplastic, and it was the dominant polymer in these samples. It is still unclear to us why we could not identify PE in our ocean water samples. There might be some other unknown processes involved, but this remains speculative. We would also like to stress that, though not quantitatively, a few other studies were able to measure nanoplastics in the ocean (ter Halle et al., 2017, Moon et al. 2024) and these authors could also not find clear signals for PE. In the revised Extended Data we show now a Monte Carlo assessment to demonstrate the effect of the presence of organic matter on the nanoplastic fingerprinting. PE indeed seems sensitive to changes in the ion matrix, with its recognition being affected more dramatically than for other polymers.*

To clarify all this, we made the following modifications in the manuscript:

“We cannot fully explain this at present as our method is in principle suitable to measure PE and PP, at least in its pristine configuration^{10,29,31}. Possible explanations are consequently the following: (i) the concentration of PE and PP nanoplastics were below our detection limit, (ii) the chemical configuration of the nanoplastic is masked by the organic background, or (iii) the nanoplastics are chemically different/modified when compared to the standard polymer.”

was modified to

“We cannot fully explain this at present as our method has proved suitable to measure PE and PP - at least in its pristine configuration - in freshwater, air and marine biota samples^{32,34,54,55} where it was the dominant polymer. Possible explanations are consequently the following: (i) the nanoplastics are chemically modified in seawater compared to the pristine polymer so that mass spectrometric fingerprinting cannot detect the modified PE/PP, (ii) the concentration of PE and PP nanoplastics were below our detection limit, or (iii) the chemical configuration of PE or PP is masked by the organic background in ocean water. We cannot rule out any of these explanations. However, through a Monte Carlo analysis (Extended Data Fig. 8), we could indeed show that PE identification was most sensitive to the effect of randomly added organic matter. It also seems very likely that photodegradation not only leads to the production of secondary nanoplastics from parent macro/microplastics^{16,21}, but that the secondary PE and PP nanoplastics have also undergone some chemical alteration^{20,25} (e.g.,

photooxidation introduces carbonyl groups¹⁶). This might result in a disparity with the diagnostic fingerprint and would explain why the ions typically associated with PE or PP are not detected.”

Moreover, to emphasize the exclusion of most organic matter through thermal desorption:

“For data reduction the mass spectra were averaged over a time period of 5 minutes once the thermal desorption reached a temperature of 200 °C, hence only the time window from 200°C to 360°C was considered. This resulted in one average value for each mass.”

was changed into:

“For data reduction, the mass spectra were averaged over a time period of 5 minutes once the thermal desorption unit reached a temperature of 200 °C, i.e., we only considered the time window from 200°C to 360°C, during which the majority of the plastic thermally desorbed. Hence, the majority of the organic matter matrix was excluded from analysis, as these compounds typically desorb before 200°C. Data integration for oven temperatures above 200°C to 360°C not only excludes volatile compounds, but also avoids pyrolysis and extensive thermolysis. Consequently, our method measures collectively free nanoplastics and nanoplastics that are loosely associated to other OM or that are aggregated (as long as the aggregate size is <1 µm).”

RR.1.7. It is interesting that the only other work published by the lead author on nanoplastics in seawater (the Dutch Wadden Sea, Materic et al. 2022) also reports PS and PET but no PE, PP, and other polymers, unlike other works by the same author on samples of ice and snow, freshwater, and biota. In agreement with Reviewer 2, could the seawater matrix with CDOM and nano/picoplankton (e.g. Synechococcus, E. hax) impact the detection, by masking PE and PP, and/or by contributing ethylbenzene and benzene groups (i.e. constituents of polystyrene; see Rocco et al. 2021, <https://doi.org/10.1038/s43247-021-00253-0>)? As I mentioned in the first round of reviews, I am not an expert in PTR-MS, but with a limited application of this technique on marine samples with no natural particulates removed/reduced (even at <1 micron size) makes me think that the method favours detection of PS and PET.

The chemical composition of organic matter will also differ between that of biota (zoo-) and phytoplankton – was that a consideration in Monte Carlo assessment? Did the authors know the organic matter load and content of their samples to carry out such assessment, i.e. which compounds were considered?

Line 186 – replace ‘configuration’ to ‘chemical composition’.

Line 294: this requires a reference

Lines 296-298: nanoplastics can also (potentially) aggregate with OM larger than 1 micron in size, so this study probably lost nanoplastics associated with particles >1 micron (e.g. marine snow, faecal pellets). This needs to be included into the discussion. Also see Gigault et al 2021 who proposes that the fate/distribution of nanoplastics might be driven by the association/aggregation with organic matter, e.g. downward transport on nanoplastic to depth).

Overall, the description of data processing needs to be more detailed and some essential aspects of it should be included in the methodology. For example, it is unclear whether a higher matching score means a better hit and if a matching score has an upper value.

R.1.8. *Additional info was added to the methods accordingly:*

*“A matching score of 2σ (z-score = 2, $p < 0.02275$, one tail distribution) was considered a positive fingerprint. Matching scores are indicated with * (z-score > 2), ** (z-score > 3) and *** (z-score > 4), where a higher matching score indicates a better fit with the library mass spectra.”*

RR1.8: Lines 331-332: what was the dominant z-score for detected PS and PET in the actual samples?

Number of polymers included in the mass spectra library should be included. Based on reference 29, the mass spectral library used in this study consists of 8 polymers (3 of which are variations of PE). I do not see it as a problem, but a limited number of spectra in the library should be acknowledged.

R.1.9. *This is now acknowledged with the following sentence in the methods section:*

“The fingerprint algorithm compares the spectra against a library comprising the seven most prevalent polymers: PE, PET, PS, PP, PPC, PVC, and tire wear.”

The authors report their values for PS and PVC as underestimates (14 times and higher) based on the recovery of polystyrene (PS), and partly overlapping spectra of PVC and PS. They also report relatively high values of PS in the lab blanks, which can particularly be an issue for samples where PS concentrations were relatively low (< 5 ng/L). The authors should clarify how the blank corrections were performed and include it into quality assurance and control section. The legend of the Extended Data Figure 3 does refer to blank correction but it is unclear whether the correction was polymer-specific or an average value of 0.90 ng/L was used.

RR. 1.9. Addressed

R.1.10. *This was indeed not entirely clear in the preceding version of the manuscript. The correction is mass-specific/ion-specific. I.e., before any fingerprinting is conducted, we firstly process the mass spectra by subtracting the mean of the lab blanks from the sample in the corresponding batch for every m/z . We processed the data in three separate batches (hence the three different figures in Extended data Fig. 3), matching with the three separate periods of measurement. Typically, per batch, we included over 15 lab blanks, from which the lab blank mean was subtracted from the samples. After this subtraction, a 3σ Limit of Detection filter was applied. This means that if the remaining mass signal was not greater than 3 times the standard deviation of the lab blanks from that batch, the value was set to 0 for that m/z . To clarify this procedure, we altered the original text:*

“The average of the lab blanks from the corresponding batch was subtracted from these values and a 3σ detection limit was applied.”

as follows:

“To account for background contamination, the mass-specific average of the lab blanks from the corresponding batch was subtracted from the averaged mass of the samples. After subtraction, a 3σ limit of detection filter was applied, whereby the mass-specific signal was set to zero when it did not exceed 3 times the standard deviation of the lab blanks.”

RR. 1.10. Addressed. In legend of Extended data figure 4 report the units as in the figure (mg m⁻³).

The group that station 9 belongs to is inconsistent throughout the manuscript. In lines 71, 94-95, station 9 is a ‘shelf break/coastal’ station; in lines 113-114, station 9 is pulled into ‘open ocean’ station, but not in line 193, when ‘open ocean’ stations are 1-8. This is also inconsistent throughout figure 2, where ‘outside gyre’ are stations 1-9 and ‘shelf break’ stations are 9-12.

R.1.11. *This has now been unified throughout the manuscript and the statistics and figures have been updated accordingly. Based on our improved gyre extent calculation, stations 1-5 are now categorized as ‘gyre’, stations 6-9 as ‘outside gyre’ and stations 10-12 that are on the continental shelf as ‘coastal’.*

RR. 1.11. Addressed.

The authors need to clarify what criteria were used to determine the boundary/extent of the gyre?

R.1.12. *The ocean gyre is defined based on Dynamic Height Anomaly (DHA) contours, which is a good proxy for streamlines. Ultimately, these calculations are based on a thermal wind balance with a reference velocity. The reference velocity is obtained from an ARGO-float based product, while thermal wind is applied to an observational-based climatology. Hence, we can determine these DHA streamlines and the associated gyre, purely based on observations. When added up, this gives DHA streamlines that correspond well to other (numerical) calculations of the North Atlantic subtropical gyre in other papers. Based on the DHA streamlines we defined the core of the gyre as the part of the gyre that is within the “last” contour that actually loops around the Atlantic Ocean (from the US, towards Europe and back). Exact details of this calculation are now provided in the method section “Calculation of the North Atlantic surface layer volume”.*

RR. 1.12. Addressed.

The choice of the sampling depth for the deep stations need clarification. Why 30 m above the seabed was chosen for the open station and not 5-10 m as for the shelf break stations to make them consistent throughout? How comparable are the data given these differences? For example, the effect of resuspension on the amount of nanoplastics observed at seabed might be more pronounced at shelf stations.

R.1.13. *30 meters from the seabed was chosen as a safety margin. The boat’s pitch and roll movement, slight bathymetry changes and drift of the vessel can lead to an unwanted hit of the CTD instrument when sampling >3 km depth. While this can damage the instrument, it, more importantly, can lead to contamination of the bottom sample with sediment particles. This situation is different on the shelf break where the water depth was only <100 meters. We agree that one should be very careful comparing the stations on the shelf break (S10, S11, S12) with the ones in the open ocean. This is also*

why we made the conscious decision to exclude the three stations on the shelf break from the subsequent ANOVA analysis (Fig. 2f), that only includes S1 to S9.

RR. 1.13. Nepheloid layer created by the resuspension/transport of the seabed sediment by the bottom currents can reach up to 1000 m above the seabed across the Atlantic. This needs to be acknowledged as a potential source of nanoplastics at 30 m depth above the seabed.

In the current form and until the point of concern are clarified, the data needs to be treated with caution.

In addition to clarifications suggested in the sections above, the authors should rework the presentation of their Figures.

In Figure 1, grouping of stations will be more obvious if the dots on the map are numbered instead of having a colour code.

R.1.14. We appreciate the feedback. The figure was updated including station numbering and a new color coding for the different categories used in the statistical analysis.

RR.1.14. add a grey bar in the legend to indicate the coastal/shelf break stations

References to Figure 2 should be improved throughout the text and indicate the subfigure ID.

R.1.15. This was indeed lacking. All references to Fig. 2 now also include the fitting subfigure ID.

Figure 2 layout and presentation can also be improved by indicating a group name (inside gyre /outside gyre/shelf) in subfigures a, b, c. It will make a comparison with subfigures d-f easier. All the box plots in Fig. 2 should indicate number of samples pulled per group.

RR.1.15. Resolved

R.1.16. This feedback has all been incorporated into a new version of Fig. 2. Additionally, we updated the colors to match with the new color coding in Fig. 1:

RR.1.16. Resolved

I suggest to present the mass concentration data in units of mass per m³ to be consistent with the units used for budget calculations.

R.1.17. All our reported units in the text and in the figures have been converted from ng/ml to mg/m³ for better consistency with the budget calculation.

RR.1.17. Resolved

Although appropriate credit to previous work has been given, on number of occasions, references to previous studies are required (see the specific comments).

Statistical analyses have been performed adequately and my only question is whether the authors tested their data for being normally distributed before applying ANOVA test.

R.1.18. To improve the statistical analysis all data points - instead of using a station average - have now been used in the ANOVA analysis (hence the increased n in Fig. 2). The normal distribution has been tested using a Shapiro-Wilk test in JMP. The surface total NP data is normally distributed ($p = 0.040$), the intermediate total NP data is on the verge of being normally distributed ($p = 0.067$) and the bottom PET NP data is also normally distributed ($p = 0.048$).

R.1.18. Resolved

The abstract needs significant improvements and be free of references and abbreviations. In the current form it is too detailed, and with some additional information given in parentheses (lines 23-24, and 26). In line 15, 'globally' needs to be removed; the use of 'this domain' is unclear (line 17). Reference to analytical method is not essential (line 20).

R.1.19. The suggestions to improve readability are greatly valued. We kept a few references since we consider them crucial for our message (a referenced abstract is typical for Nature). We also kept the abbreviations of the polymers as these plastics are commonly often better known by their abbreviated names. We have rewritten the abstract as follows to incorporate the feedback:

“Plastic pollution of the marine realm is widespread, with most scientific attention given to macro and microplastics. Ocean nanoplastics (<1 μm), in contrast, are understudied and the mass budget of this plastic size class thus remains unknown. Here, we measured nanoplastic concentrations on an ocean basin scale along a transect crossing the North Atlantic from the subtropical gyre to the northern European shelf. Our findings revealed substantial amounts of polyethylene terephthalate (PET), polystyrene (PS) and polyvinyl chloride (PVC) nanoplastics ($\sim 2 - 32 \text{ mg m}^{-3}$) throughout the entire water column. We observed higher concentrations of nanoplastics in the surface layer, specifically near the European continent, while nanoplastic accumulation in the subtropical gyre was only weakly visible at intermediate depth. Lowest nanoplastic concentrations, predominantly composed of PET, were found in bottom waters. For the surface layer of the temperate to subtropical North Atlantic, we estimated that the mass of nanoplastic amounts to 30 million tonnes (Mt); 12.0 Mt PET, 9.4 Mt PS and 8.4 Mt PVC. This is in the same range or exceeds previous budget estimates of macro/microplastic for the entire Atlantic^{1,2} or global ocean^{3,4}. Our findings suggest that nanoplastics are the most important part of ocean plastic pollution.”

The article does not have a summary paragraph, and it needs to be added to place the results and their significance into wide context.

R.1.20. *This is a good suggestion, in addition to synthesizing our data to a North Atlantic nanoplastic budget, we have highlighted now the broader context in the last paragraph of the MS, placing the significance of our findings into a global context (see R.1.5. for the updated discussion text).*

RR. 1.19 and 1.20. *I still believe that the authors can make both the abstract and the summary paragraph stronger. Nanoplastics are severely understudied (only a few studies exist to date and most of them are qualitative) in the marine environment and this study is the first one to report mass concentrations and estimated budgets of polymer-specific nanoplastics on a basin-wide scale and with depth. The last sentence of the abstract needs to be rephrased with more implications added – marine plastic contamination is now shown to be predominantly in the smallest, nano-size category. This very small size of plastic contaminants may drive completely different physical and chemical interactions and pathways of these contaminants in the marine environment and their impacts therein.*

In lines 228-230 it should be highlighted that very limited data on the magnitude and characteristics of marine nanoplastic contamination exists to support the laboratory-based studies investigating the effects on nanoplastics on the health of biota/ecosystem functioning.

Introduction highlights major knowledge gaps but the language needs to be improved. There is a contradiction between lines 36 ('accumulating' plastics) and lines 38-39 where the fate of plastic is referred to as 'manifold' (also confusing word).

R.1.21. *"is manifold" has been removed resulting in the sentence: "The further fate of plastic in the ocean depends on multiple factors, including the density of the plastic items and their transport at the ocean surface¹⁶."*

RR.1.21. *I suggest replacing 'accumulating' with 'present'. The introduction paragraph (lines 36-53) needs further revision. I think the introduction will read stronger if it includes the following aspects: i) nanoplastics are the smallest and potentially most pervasive and harmful of plastic contaminants as they can pass through and accumulate in biological cell and tissues; ii) because of their small size, chemical and physical changes and interactions within and on the surfaces on nanoplastics can be faster and completely different to that of their larger precursors (macro/microplastics); authors only mention Brownian motion but it is not sufficient; iii) their small size also makes nanoplastics very challenging to measure/separate from the natural particular matter, especially of a similar size fraction (e.g. colloidal organic and inorganic matter); the authors partly address this in lines 58-59. See Gigault et al. 2021 and Mitrano et al. 2021 for references. We need reliable data on the abundance, characteristics and distribution of nanoplastics in the marine environment to fully understand the extent of plastic contamination, its fate and effects in the environment (partly addressed in lines 60-61).*

In line 40, 'accumulating hotspots' need to be rephrased. Subtropical gyres are also zones of convergence (line 41).

R.1.22. *This was indeed a mistake in phrasing. The sentence has been rewritten:*

“Accumulation hotspots of floating plastics include bays and convergence zones, such as the subtropical ocean gyres^{3,4},...”

RR.1.22. *How about ‘Floating plastics tend to accumulate in coastal water bays and in the convergence zones of the open ocean, such as subtropical gyres’. Give reference for bays as it is missing in the current version of the manuscript.*

References are required in lines 49-51.

R.1.23. *We added the following reference to support the statement in these lines:*

Sun, H., Jiao, R., & Wang, D. (2021). The difference of aggregation mechanism between microplastics and nanoplastics: Role of Brownian motion and structural layer force. Environmental Pollution, 268, 115942.

RR.1.23. Resolved

The authors also need to include sentence about why measuring the nanoplastic concentrations is important.

R.1.24. *To emphasize the importance the following sentence was added to the introduction:*

“Hence, nanoplastics are not included in any ocean plastic budget estimates^{1,3,4}. This hinders our comprehensive understanding of the environmental impact and potential health hazards of marine plastic pollution.”

RR.1.24. I agree; also see comments in RR1.22

Line(s) 72 What does well-mixed surface layer mean as opposed to surface layer? I suggest to drop ‘well-mixed’ to avoid confusion.

R.1.25. *We removed ‘well-mixed’ from the manuscript to avoid confusion as ‘surface layer’ indeed already implies the well-mixed layer.*

RR.1.25. This terminology is still confusing in the revised manuscript. The authors define ‘surface layer’ based on the mixed-layer depth. See my earlier comment RR1.3.

Line(s) 72 Start a new paragraph for clarity

R.1.26. *A new paragraph was started as follows:*

“Samples for nanoplastic analysis were recovered from 12 hydrocast stations of which stations 1-5 were located in the North Atlantic Subtropical Gyre - NASG (‘gyre’), stations 6-9 were in the open ocean but outside of the gyre (‘outside gyre’) and stations 10-12 were on the European shelf (‘coastal’) (Fig. 1).

Nanoplastics in the surface layer at all 12 hydrocast stations comprised PVC, PET, and PS in the mg m⁻³ range (Fig. 2a),...

RR.1.26. Resolved

Line(s) 131-133 Provide a reference to the 'less polluted' offshore waters – it is unclear whether the authors refer to their nanoplastic observations or earlier studies reporting decreasing gradient in microplastics and macroplastics abundance away from the shore.

R.1.27. We are referring to our own nanoplastic observations. Therefore, we added a reference to Fig. 2d to clarify this: "Shelf surface waters with comparably high nanoplastic concentrations²⁶ are then entrained with less polluted offshore waters (Fig. 2d),..."

RR.1.27. Resolved

Line(s) 142 What is ocean surface boundary level and how is it related to 10 m depth where the samples were collected?

R.1.28. We aim to refer just to the surface layer, as mentioned before in the manuscript. Hence "ocean" and "boundary" has been removed to improve consistency resulting in: "The moderate difference in nanoplastic concentrations between 'gyre' and 'outside gyre' stations (Fig. 2d) thus indicates that nanoplastic concentrations in the surface layer might be horizontally homogenized as a result of shear dispersion and wind-induced turbulent mixing^{40,41}."

RR.1.28. See comments RR. 1.3 and 1.25 when referring to 'surface' layer.

Line(s) 136 Large variations in the inter-gyre abundances of macro and microplastics reported in previous studies should be acknowledged.

R.1.29. Nuance was added to line 136 by rephrasing it as follows: "However, floating macro/microplastic generally accumulates in the subtropical gyres^{2,3,4,37}..."

RR.1.29. Change to '...macro/microplastics generally accumulate...'; I don't see the authors mentioning the variability in plastic concentrations within the gyres reported previously.

Line(s) 154 Where is the depth of winter mixing at all the stations/across the transect? Exchange can happen when mixed layer depth is the highest.

R.1.30. The average surface layer depth in the temperate to subtropical ocean seldom exceeds 100-150 m (see for example the globally and annually resolved overview on mixed layer depth presented in de Boyer Montégut et al., JGR, 2004). It thus seems unlikely that the water layer sampled at 1000 m water depth substantially mixes with surface waters.

RR.1.30. Resolved

Line(s) 157 Reference needed

R.1.31. We added the following reference: Boyd, P. W., Claustre, H., Levy, M., Siegel, D. A., & Weber, T. (2019). Multi-faceted particle pumps drive carbon sequestration in the ocean. *Nature*, 568(7752), 327-335.

RR.1.31. Resolved

Line(s) 161-167 References needed

R.1.32. We added the following references to support the statements in these lines:

Gewert, B., Plassmann, M. M., & MacLeod, M. (2015). Pathways for degradation of plastic polymers floating in the marine environment. *Environmental science: processes & impacts*, 17(9), 1513-1521.

Enfrin, M., Lee, J., Gibert, Y., Basheer, F., Kong, L., & Dumée, L. F. (2020). Release of hazardous nanoplastic contaminants due to microplastics fragmentation under shear stress forces. *Journal of hazardous materials*, 384, 121393.

Lv, S., Cui, K., Zhao, S., Li, Y., Liu, R., Hu, R., ... & Shao, Z. (2024). Continuous generation and release of microplastics and nanoplastics from polystyrene by plastic-degrading marine bacteria. *Journal of Hazardous Materials*, 465, 133339.

Dawson, A. L., Kawaguchi, S., King, C. K., Townsend, K. A., King, R., Huston, W. M., & Bengtson Nash, S. M. (2018). Turning microplastics into nanoplastics through digestive fragmentation by Antarctic krill. *Nature communications*, 9(1), 1001.

Zhao, J., Lan, R., Wang, Z., Su, W., Song, D., Xue, R., ... & Xing, B. (2024). Microplastic fragmentation by rotifers in aquatic ecosystems contributes to global nanoplastic pollution. *Nature Nanotechnology*, 19(3), 406-414.

RR.1.32. Resolved

170-171 Degradation, fragmentation and resuspension of plastics at the seabed to be discussed/acknowledged

R.1.33. Since we didn't measure nanoplastic in the sediments and the initial source of nanoplastic has to be from surface waters we deemed the discussion sufficient as it is. Given the limited word count we have, we prioritized keeping the discussion close to our own measurements.

RR.1.33. I disagree – see the comment on the nepheloid layer RR1.13. Not many words will be required to acknowledge that it can be a source of plastics detected at 30 m above the seabed.

Line(s) 185 Reference missing

R.1.34. We have slightly modified this section, to account for the newly added results from our Monte Carlo analysis. We believe that the further part of this paragraph where we elaborate on the chemical alteration of pristine plastics through photodegradation is sufficiently referenced.

Line(s) 279 Is there an estimate of thermal desorption efficiency for the polymers analysed?

R.1.35. Yes, we performed PS spike-and-recovery experiments of which the results are in Extended Data Tab. 1. Furthermore, calibration curves for PS and PET have been constructed in a previous study: <https://doi.org/10.1016/j.scitotenv.2022.157371>. There are no quantitative standards available to test for all plastics, complicating these types of thermal desorption efficiency assessments.

The following sentences have been added to the methods: "Spike-and-recovery experiments were carried out for PS. Homogenized suspensions of 100 or 200 ng of PS was loaded into a vial along with 1.5 ml of seawater sample. Fingerprinting these spiked samples consistently yielded positive matches for PS with z-scores of 3 or higher. Spiking experiments were performed in triplicate to obtain a reliable recovery rate (Extended Data Tab. 1). The spiking experiment revealed a recovery/ionization efficiency rate of $\sim 7\% \pm 2.2$, which agrees with our previous works^{26,32,34}. This entails that the actual PS concentrations in the samples might be 14 times higher. Because of the challenge to load precise amounts of plastic in the nanogram range, spike-and-recovery experiments have not yet been performed for PVC or PET. In a previous study, a linear correction factor of 5.28 ± 1.48 for PS and a non-linear correction factor between 15.05 ± 0.9 for 59 ng PET load to 26.06 ± 6.8 for 177 ng PET load have been reported²⁶."

RR.1.35. The reported spike recovery seems very low, compared to that of volatile organic compounds (e.g. Salvador et al. 2016). This needs an expert's opinion (I am not).

Line(s) 301 Uncertainty/error/standard deviation missing here

R.1.36. Standard deviation has been added: "While we found a low background signal of nanoplastics in the lab blanks ($0.90 \pm 1.45 \text{ mg m}^{-3}$ averaged over all polymers and all lab blanks)".

RR.1.36. Resolved

Line(s) 301 and 309 Provide standard deviation

R.1.37. Standard deviation for the blanks and standard errors for the transect averages (as reported in the Main part of the manuscript) have been added: "The average nanoplastic background concentration of $0.90 \pm 1.45 \text{ mg m}^{-3}$ is low compared with the transect averages of $19.9 \pm 2.0 \text{ mg m}^{-3}$ for the surface layer, $14.4 \pm 1.9 \text{ mg m}^{-3}$ for 1000 m depth and $6.2 \pm 0.6 \text{ mg m}^{-3}$ for the bottom layer."

RR.1.37. Resolved

Dear Editor,

Please find below a point-by-point reply to the referee's comments and criticism. Our replies are provided in italic type.

Referees' comments:

Referee #1 (Remarks to the Author):

The unique work by Hietbrink et al. presents a step-change in our ability to quantify plastic pollution in the ocean down to the smallest particle size category – nanoplastics – which, as this study now demonstrates, is the most pervasive. The quality of the manuscript improved significantly from the previous submissions and the authors diligently addressed the comments and suggestions raised in the previous rounds of reviews. From methods perspective, the manuscript was strengthened by performing the additional analysis of Sargassum biomass samples to identify whether this type of marine algae could interfere with the detection of PE, PET and PS (per suggestion of Reviewer 2). I would like to point out that the Sargasso Sea (20-35°N and 40-70°W) is outside of the subtropical oligotrophic gyre sampled in this study (stations 1-5), where the community structure (and thus the source of CDOM) is likely dominated by nano- and picoplankton (Organelli et al 2019). However, studies suggest (e.g. Johns et al. 2020) that wind and surface currents disperse Sargassum and thus associated CDOM over the long distances in the Atlantic, including the N-E Atlantic where the samples were collected for this study. I therefore agree that the demonstrated lack of interference from the Sargassum-derived organic matter can support the robustness of the nanoplastic measurements.

I thus recommend Hietbrink et al.'s work for publication in Nature. There just some very minor points that need to be addressed prior to publishing this research:

We are grateful for the referee's appraisal of our manuscript.

Line 52 and line 63: As authors correctly suggest, 'most studies assessing the impacts and toxicity of nanoplastics use baseline nanoplastic concentrations unsupported by robust environmental measurements' (lines 247-248). I therefore suggest to refer to the effects from nanoplastics to the environment as 'potential' throughout the manuscript.

We agree and have amended the following sentences (changes highlighted in green):

*"Especially photodegradation has been discussed as a key process in the breakdown of floating plastic litter at the sea surface, likely providing a constant source of nanoplastic particles to the ocean^{3,23,24}, where they **potentially** have negative effects on marine life^{10,25,26}."*

*"This hinders our comprehensive understanding of the **potential** environmental impact and health hazards of marine plastic pollution."*

Line 59: I suggest to use the word 'spatially' instead of 'geographically'.

We intend to communicate that nanoplastic concentration differences between regions or ocean basins are unknown, and are worried that the word 'spatially' would be associated

with smaller spatial scales, e.g., the nanoplastic distribution within an ocean basin. Moreover, 'spatially' technically would also mean over depth, and we would like to clearly distinguish here between vertical distributions over depth and distributions between ocean basins. Considering these concerns, we ultimately chose to retain "geographically."

Line 197: the term 'pristine configuration' is confusing. I suggest to use, for example, 'unaltered chemical composition' or 'plastic that has not been weathered yet' ... or something like that.

We agree that would contribute to clearer communication. Accordingly, the following sentences have been modified:

*"We cannot fully explain this at present as our method has proved suitable to measure PE and PP – **provided the chemical composition remains unaltered** – in freshwater, air and marine biota samples^{33,35,62,63} where it was the dominant polymer."*

*"(i) the nanoplastics are chemically modified in seawater compared to **unaltered polymers** so that mass spectrometric fingerprinting cannot detect the modified PE/PP,..."*

*"The pre-processed data was subsequently used for nanoplastic fingerprinting against **chemically unaltered** plastics (the library mass spectra)..."*

*Extended Data Figure 9 caption: "Regardless, we must conclude that **chemically unaltered** PE, if present, remains below our detection limit in seawater samples."*

Line 290: please give a full name to IMAU

*Has been implemented: "The water samples were processed in the PTR-MS lab at **the Institute for Marine and Atmospheric research** in Utrecht."*

Line 398: please refer to Sargassum biomass, not 'tissue'. Also please give the justification to using Sargassum as the readers may not be aware of this choice for testing the robustness of the method.

"Tissue" has been replaced with "biomass" throughout the manuscript.

*To clarify the justification the following sentence: "To assess potential false positives from organic matter, we analysed Sargassum **biomass** samples, using Sargassum as a proxy for complex organic material that may be present in seawater."*

Was changed into:

*"To assess potential false positives from organic matter, we analysed Sargassum **biomass** samples as a proxy for complex organic material. Sargassum is abundant in the Sargasso Sea and disperses to other parts of the Atlantic, including the northeast⁶⁰."*

Referee #3 (Remarks to the Author):

Main comments based on earlier review:

Thank you for the further explanation of the TD-PTR-MS method and inclusion of a Sargassum sample to assess the potential for Sargassum OM interference. This added check increases confidence in the method and addresses my prior concerns. I now recommend publication.

I re-read the rest of the document, including the response to other reviews. I don't have any material edits to suggest, so I will not comment further on the manuscript.

This is an important, surprising, and concerning result. If NPs are this widespread and at such high concentrations, we need to confirm their quantification (e.g., by applying complimentary methods that provide different views of organic carbon chemistry) and assess their potential impacts on ocean life as well as people (I assume if they are this abundant in the ocean they will also be more prevalent than recognized in media through which people can be exposed – e.g., air, food, drink).

I commend the authors on their work and their positive responses to the critique provided by reviewers.

Regarding Nature's review criteria: The manuscript presents highly original and significant results. The data and the quality of presentation are excellent. Statistics and uncertainties are dealt with appropriately. The work is clearly presented with the response of the authors to other reviews having further improved the manuscript in this regard.